# Snow sensitivity to temperature and precipitation change during compound cold-hot and wet-dry seasons in the Pyrenees.

Josep Bonsoms [1], Juan I. López-Moreno [2], Esteban Alonso-González [3]

[1] Department of Geography, Universitat de Barcelona, Barcelona, Spain.

[2] Instituto Pirenaico de Ecología (IPE-CSIC), Campus de Aula Dei, Zaragoza, Spain.

[3] Centre d'Etudes Spatiales de la Biosphère (CESBIO), Université de Toulouse, CNES/CNRS/IRD/UPS, Toulouse, France.

Corresponding author: Juan I. López-Moreno (nlopez@ipe.csic.es)

**Abstract.** The Mediterranean basin has experienced one of the highest warming rates on earth
during the last few decades, and climate projections predict water-scarcity in the future. Mid-
latitude Mediterranean mountain areas, such as the Pyrenees, play a key role in the hydrological
resources for the highly populated lowland areas. However, there are still large uncertainties about
the impact of climate change on snowpack in the high mountain ranges of this region. Here, we
perform a snow sensitivity to temperature and precipitation change analysis of the Pyrenean
snowpack (1980 – 2019 period) using five key snow-climatological indicators. We analyzed snow
sensitivity to temperature and precipitation during four different compounds weather conditions
(cold-dry [CD], cold-wet [CW], warm-dry [WD], and warm-wet [WW]) at low elevations (1500
m), mid-elevations (1800 m), and high elevations (2400 m) in the Pyrenees. In particular, we
forced a physically based energy and mass balance snow model (FSM2), with validation by
ground-truth data, and applied this model to the entire range, with forcing of perturbed reanalysis
climate data for the period 1980 to 2019 as the baseline. The FSM2 model results successfully
reproduced the observed snow depth (HS) values ($R^2 > 0.8$), with relative root-mean square error
and mean absolute error values less than 10% of the observed HS values. Overall, the snow
sensitivity to temperature and precipitation change decreased with elevation and increased
towards the eastern Pyrenees. When the temperature increased progressively at 1ºC intervals, the
largest seasonal HS decreases from the baseline were at +1ºC. A 10% increase of precipitation
counterbalanced the temperature increases (≤1ºC) at high elevations during the coldest months,
because temperature was far from the isothermal 0ºC conditions. The maximal seasonal HS and
peak HS max reductions were during WW seasons, and the minimal reductions were during CD
seasons. During WW (CD) seasons, the seasonal HS decline per °C was 37% (28 %) at low
elevations, 34% (30%) at mid elevations, and 27% (22 %) at high elevations. Further, the peak
HS date was on average anticipated 2, 3 and 8 days at low, mid and high elevation, respectively.
Results suggest snow sensitivity to temperature and precipitation change will be similar at other

mid-latitude mountain areas, where snowpack reductions will have major consequences on the nearby ecological and socioeconomic systems.

**Keywords:** Snow, Climate change, Sensitivity, Alpine, Mediterranean Mountains, Mid-latitude, Pyrenees.

## 1   Introduction

Snow is a key element of the Earth's climate system (Armstrong and Brun, 1998) because it cools the planet (Serreze and Barry, 2011) by altering the Surface Energy Balance (SEB), increasing the albedo, and modulating surface and air temperatures (Hall, 2004). Northern-Hemispheric snowpack patterns have changed rapidly during recent decades (Hammond et al., 2018; Hock et al., 2019; Notarnicola et al., 2020). It is crucial to improve our understanding of the timing of snow ablation and snow accumulation due to changing climate conditions because snowpack affects many nearby social and environmental systems. From the hydrological point of view, snow melt controls mountain runoff rate during the spring (Barnett et al., 2005; Adams et al., 2009; Stahl et al., 2010), river flow magnitude and timing (Morán-Tejeda et al., 2014; Sanmiguel-Vallelado et al., 2017), water infiltration and groundwater storage (Gribovszki et al., 2010; Evans et al., 2018), and transpiration rate (Cooper et al., 2020). The presence and duration of snowpack affects terrestrial ecosystem dynamics because snow ablation date affects photosynthesis (Woelber et al., 2018), forest productivity (Barnard et al., 2018), freezing and thawing of the soil (Luetschg et al., 2008; Oliva et al., 2014), and thickness of the active layer in permafrost environments (Hrbáček et al., 2016; Magnin et al., 2017). Snowpack also has remarkable economic impacts. For example, the snowpack at high elevations and surrounding areas determines the economic success of many mountain ski-resorts (Scott et al., 2003; Pons et al., 2015; Gilaberte-Búrdalo et al., 2017). Changes in the snowpack of mountainous regions also influence associated lowland areas because it affects the availability of snow meltwater that is used for water reservoirs, hydropower generation, agriculture, industries, and other applications (e.g., Sturm et al., 2017; Beniston et al., 2018).

Mid-latitude snowpacks have among the highest snow sensitivities worldwide (Brown and Mote, 2009; López-Moreno et al., 2017; 2020b). In regions at high latitudes or high elevations, increasing precipitation can partly counterbalance the effect of increases of temperature on snow cover duration (Brown and Mote, 2009). Climate warming decreases the maximum and seasonal

snow depth (HS), the snow water equivalent (SWE) (Trujillo and Molotch, 2014; Alonso-González et al., 2020a; López-Moreno et al., 2013; 2017), and the fraction total precipitation as snowfall (snowfall ratio; e.g., Mote et al., 2005; Lynn et al., 2020; Jeenings and Molotoch, 2020; Marshall et al., 2019), and also delays the snow onset date (Beniston, 2009; Klein et al., 2016). However, warming can slow the early snow ablation rate on the season (Pomeroy et al., 2015; Rasouli et al., 2015; Jennings and Molotch, 2020; Bonsoms et al., 2022; Sanmiguel-Vallelado et al., 2022) because of the earlier HS and SWE peak dates (Alonso-González et al., 2022), which coincide with periods of low solar radiation (Pomeroy et al., 2015; Musselman et al., 2017a).

The Mediterranean basin is a region that is critically affected by climate change (Giorgi, 2006) being densely populated (>500 million inhabitants) and affected by an intense anthropogenic activity. Warming of the Mediterranean basin will accelerate for the next decades, and temperatures will continue to increase in this region during the warm months (Knutti and Sedlacek, 2013; Lionello and Scarascia 2018; Cramer et al., 2018; Evin et al., 2021; Cos et al., 2022), increasing atmospheric evaporative demands (Vicente-Serrano et al., 2020), drought severity (Tramblay et al., 2020), leading to water-scarcity over most of this region (García-Ruiz et al., 2011). Mediterranean mid-latitude mountains, such as the Pyrenees, where this research focuses, are the main runoff generation zones of the downstream areas (Viviroli and Weingartner, 2004) and provide most of the water used by major cities in the lowlands (Morán-Tejeda et al., 2014).

Snow patterns in the Pyrenees have high spatial diversity (Alonso-González et al., 2019), due to internal climate variability of mid-latitude precipitation (Hawkins and Sutton 2010; Deser et al., 2012), high interannual and decadal variability of precipitation in the Iberian Peninsula (Esteban-Parra et al., 1998; Peña-Angulo et al., 2020) as well as the abrupt topography and the different mountain exposure to the Atlantic air masses (Bonsoms et al., 2021a). Thus, snow accumulation per season is almost twice as much in the northern slopes as in the southern slopes (Navarro-Serrano and López-Moreno, 2017), and there is a high interannual variability of snow in regions at lower elevations (Alonso-González et al., 2020a) and in the southern and eastern regions of the Pyrenees (Salvador-Franch et al., 2014; Salvador-Franch et al., 2016; Bonsoms et al., 2021b). Since the 1980s, the energy available for snow ablation has significantly increased in the Pyrenees (Bonsoms et al., 2022), and winter snow days and snow accumulation have non-statically significantly increased (Buisan et al., 2016; Serrano-Notivoli et al., 2018; López-Moreno et al., 2020a; Bonsoms et al., 2021a) due to the increasing frequency of positive west and south-west advections (Buisan et al., 2016). 21[st] century climate projections for Pyrenees anticipate a

temperature increase of more than 1°C to 4°C (relative to 1986–2005), and an increase (decrease)
of precipitation by about 10% for the eastern (western) regions during winter and spring (Amblar-
Frances et al., 2020). Therefore, changes in snow patterns in regions with high elevations are
uncertain because winter snow accumulation is affected by precipitation (López-Moreno et al.,
2008) and Mediterranean basin winter precipitation projections have uncertainties up to 80% of
the total variance (Evin et al., 2021).

Previous studies in the central Pyrenees (López-Moreno et al., 2013), Iberian Peninsula Mountain
ranges (Alonso-González et al., 2020a), and mountain areas that have Mediterranean climates
(López-Moreno et al., 2017) demonstrated that snowpack sensitivity to changes in climate are
mostly controlled by elevation. Despite the impact of climate warming in mountain hydrological
processes, there is limited understanding of the snow sensitivity to temperature and precipitation
changes and seasonality of mid-latitude Mediterranean mountain snowpacks. Some studies
reported different snowpack sensitivities during wet and dry years (López-Moreno et al., 2017;
Musselman et al., 2017b; Rasouli et al., 2022; Roche et al., 2018). However, the sensitivity of
snow during periods when there are seasonal compound weather (temperature and precipitation)
conditions has not yet been analyzed. The high interannual variability of the Pyrenean snowpack,
which is expected to increase according to climate projections (López-Moreno et al., 2008),
indicates a need to examine snowpack sensitivity to temperature and precipitation change
focusing on the year-to-year variability. Warm seasons in the Mediterranean basin require special
attention because these are likely to increase in the future (e.g., Vogel et al., 2019; De Luca et al.,
2020; Meng et al., 2022). Further, the occurrence of different HS trends at mid- and high-elevation
areas of this range (López-Moreno et al., 2020a) suggest that elevation and spatial factors
contribute to the wide variations of the sensitivity of snow to the climate.

Therefore, the main objective of this research is to quantify snow (accumulation, ablation, and
timing) sensitivity to temperature and precipitation change during compound temperature and
precipitation seasons in the Pyrenees.

**2 Geographical area and climate setting**

The Pyrenees is a mountain range located in the north of the Iberian Peninsula (south Europe;
42ºN-43ºN to 2ºW-3ºE) that is aligned east-to-west between the Atlantic Ocean and the
Mediterranean Sea. The highest elevations are in the central region (Aneto, 3404 m) and
elevations decrease towards the west and east (Figure 1). The Mediterranean basin, including the
Pyrenees, is in a transition area, and is influenced by the continental climate and the subtropical
temperate climate. Precipitation is mostly driven by large-scale circulation patterns (Zappa et al.,
2015; Borgli et al., 2019), the jet-stream oscillation during winter (Hurell, 1995), and land-sea
temperature differences (Tuel and Eltahir, 2020). During the summer, the northward movement
of the Azores high pressure region brings stable weather, and precipitation is mainly convective
at that time (Xercavins, 1985). Precipitation is highly variable depending on mountain exposure
to the main circulation weather types; it ranges from about 1000 mm/year to about 2000 mm/year
(in the mountain summits), with lower levels in the east and south (Cuadrat et al., 2007). There is
a slight disconnection of the general climate circulation towards the eastern Pyrenees, where the
Mediterranean climate and East Atlantic/West Russia (EA-WR) oscillations have greater effects
on snow accumulation (Bonsoms et al., 2021a). In the southern, western, and central massifs of
the range, the Atlantic climate and the negative North Atlantic Oscillation (NAO) phases regulate
snow accumulation (W and SW wet air flows; López-Moreno, 2005; López-Moreno and Vicente-
Serrano, 2007; Buisan et al., 2016; Alonso-González et al., 2020b). In the northern slopes, the
positive phases of the Western Mediterranean Oscillation (WeMO) linked with NW and N
advections trigger the most episodes of snow accumulation (Bonsoms et al., 2021a). The seasonal
snow accumulation in the northern slopes is almost double the amount (about 500 cm more) as in
the southern slopes at an elevation of about 2000 m (Bonsoms et al., 2021a). The
temperature/elevation gradient is about 0.55ºC/100 m (Navarro-Serrano and López-Moreno,
2018) and the annual 0ºC isotherm is at about 2750 to 2950 m (López-Moreno and García-Ruiz,
2004). Net radiation and latent heat flux governs the energy available for snow ablation; the
former heat flux increases at high elevations and the latter towards the east (Bonsoms et al., 2022).


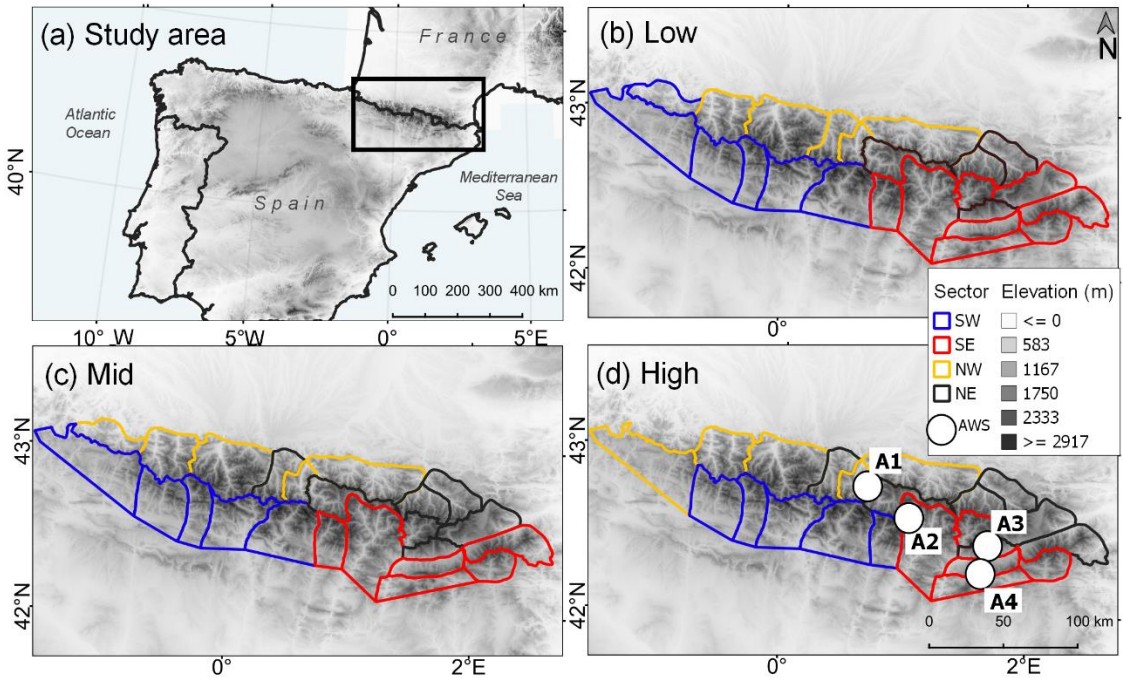


**Figure 1** (a) Study area. Pyrenean massifs grouped by sectors for (b) low, (c) mid and (d) high elevation. The white dots indicate the locations of the automatic weather stations (AWS) shown at Table 1. Massifs delimitation is based on the spatial regionalization of the SAFRAN system, which groups massifs according to topographical and meteorological characteristics (modified from Durand et al., 1999).

**3 Data and methods**

**3.1 Snow model**

Snowpack was modelled using a physical-based snow model, the Flexible Snow Model (FSM2; Essery, 2015). This model resolves the SEB and mass balance to simulate the state of the snowpack. FSM2 is open access and available at https://github.com/RichardEssery/FSM2 (last access 16 December 2022). Previous studies tested the FSM2 (Krinner et al., 2018), and its application in different forest environments (Mazzoti et al., 2021), and hydro-climatological mountain zones such the Andes (Urrutia et al., 2019), Alps (Mazzoti et al., 2020), Colorado (Smyth et al., 2022), Himalayas (Pritchard et al., 2020), Iberian Peninsula Mountains (Alonso-González et al., 2020a; Alonso-González et al., 2022), Lebanese mountains (Alonso-González et al., 2021), providing confidential results. The FSM2 requires forcing data of precipitation, air temperature, relative humidity, surface atmospheric pressure, wind speed, incoming shortwave radiation ($SW_{inc}$), and incoming long wave radiation ($LW_{inc}$). We have evaluated different FSM2 model configurations (not shown) without remarkable differences in the accuracy and

performance metrics. Thus, the FSM2 configuration included in this work estimates snow cover
fraction based on a linear function of HS and albedo based on a prognostic function, with increases
due to snowfall and decreases due to snow age. Atmospheric stability is calculated as function of
the Richardson number. Snow density is calculated as a function of viscous compaction by
overburden and thermal metamorphism. Snow hydrology is estimated by gravitational drainage,
including internal snowpack processes, runoff, refreeze rates, and thermal conductivity. Table S1
summarizes the FSM2 configuration and the FSM2 compile numbers.

**3.2 Snow model validation**

FSM2 configuration was validated by in situ snow records of four automatic weather stations
(AWSs) that were at high elevations in the Pyrenees. Precipitation in mountainous and windy
regions is usually affected by undercatch (Kochendorfer et al., 2020). Thus, the instrumental
records of precipitation were corrected for undercatch by applying an empirical equation validated
for the Pyrenees (Buisan et al., 2019). Precipitation type was classified by a threshold method
(Musselman et al., 2017b; Corripio et al., 2017): snow when the air temperature was below 1°C
and rain when the air temperature was above 1°C, according to previous research in the study area
(Corripio et al., 2017). The $LW_{inc}$ heat flux of the AWSs (Table 1) were estimated as previously
described (Corripio et al., 2017). Due to the wide instrumental data coverage (99.3% of the total
dataset), gap-filling was not performed. The HS records were measured each 30 min using an
ultrasonic snow depth sensor. The meteorological data used in the validation process were
provided and managed by the local meteorological service of Catalonia
(https://www.meteo.cat/wpweb/serveis/formularis/peticio-dinformes-i-dades-
meteorologiques/peticio-de-dades-meteorologiques/; data requested: 14/01/2021). Quality-
checking of the data was performed using an automatic error filtering process in combination with
a climatological, spatial, and internal coherency control defined by the SMC (2011).

**Table 1.** Characteristics of the four AWSs.

| Area | Code | Lat/Lonº | Elevation (m) | Atlantic Ocean, Distance (km) | Mediterranean Sea, Distance (km) | Validation period (years) | Years |
|---|---|---|---|---|---|---|---|
| Central-Pyrenees, Northern slopes | A1 | 42.77/0.73 | 2228 | 200 | 190 | 2004–2020 | 16 |
| | A2 | 42.61/0.98 | 2266 | 225 | 170 | 2001–2020 | 19 |
| Eastern Pyrenees, Southern slopes | A3 | 42.46/1.78 | 2230 | 295 | 115 | 2005–2020 | 15 |

| Eastern Pre-Pyrenees, Northern slopes | A4 | 42.29/1.71 | 2143 | 300 | 110 | 2009–2020 | 11 |


Model accuracy was estimated based on the mean absolute error (MAE) and the root mean square
error (RMSE), and model performance was estimated by the coefficient of determination ($R^2$).
The MAE and the RMSE indicate the mean differences of the modelled and observed values.

**3.3 Atmospheric forcing data**

We forced the FSM2 with the open access climate reanalysis dataset provided by Vernay et al.
(2021), which consists of the modelled values from the SAFRAN meteorological analysis. The
FSM2 was run at an hourly resolution for each massif, each elevation range, and each climate
baseline and perturbation scenario from 1980 to 2019. The SAFRAN system provides data for
homogeneous meteorological and topographical mountain massifs every 300 m, from 0 to 3600
m (Durand et al., 1999; Vernay et al., 2021). We analyzed three elevation bands: low (1500 m),
middle (1800 m), and high (2400 m). Precipitation type was classified using the same threshold
approach used for model validation. Atmospheric emissivity was derived from the SAFRAN
$LW_{inc}$ and air temperature. SAFRAN was forced using numerical weather prediction models
(ERA-40 reanalysis data from 1958 to 2002 and ARPEGE from 2002 to 2020). Meteorological
data were calibrated, homogenized, and improved by in situ meteorological observations data
assimilation (Vernay et al., 2021). Durand et al. (1999; 2009a; 2009b) provided further technical
details of the SAFRAN system. Previous studies used the SAFRAN system for the long-term HS
trends (López-Moreno et al., 2020), extreme snowfall (Roux et al., 2021), and snow ablation
analysis (Bonsoms et al., 2022). SAFRAN system has been extensively validated for the
meteorological modelling of continental Spain (Quintana-Seguí et al., 2017), France (Vidal et al.,
2010) or alpine snowpack climate projections (Verfaille et al., 2018), among other works.

**3.4 Snow sensitivity to temperature and precipitation change analysis**

Snow sensitivity to temperature and precipitation change was analyzed using a delta-change
methodology (López-Moreno et al., 2008; Beniston et al., 2016; Musselman et al., 2017b; Marty
et al., 2017; Alonso-González et al., 2020a; Sanmiguel-Vallelado et al., 2022). In this method, air
temperature and precipitation were perturbed for each massif and elevation range based the
historical period (1980–2019). Air temperature was increased from 1 to 4ºC at 1ºC intervals,
assuming an increase of $LW_{inc}$ accordingly (Jennings and Molotch, 2020). Precipitation was
changed from −10% to +10% at 10% intervals, in accordance with climate model uncertainties
and the maximum and minimum precipitation projections for the Pyrenees (Amblar-Frances et
al., 2020).

**3.5 Snow climate indicators**

Snow sensitivity to temperature and precipitation change was analyzed using five key indicators:
(*i*) seasonal average HS, (*ii*) seasonal maximum absolute HS peak (peak HS max), (*iii*) date of the
maximum HS (peak HS date), (*iv*) number of days with HS > 1 cm on the ground (snow duration),
and (*v*) daily average snow ablation per season (snow ablation, hereafter). Snow ablation was
calculated as the difference between the maximum daily HS recorded on two consecutive days
(Musselman et al., 2017a), and only days with decreases of 1 cm or more were recorded. Some
seasons had more than one peak HS; for this reason, peak HS date was determined after applying
a moving average of 5-days. All indicators were computed according to massif and elevation
range.

**3.6 Definitions of compound temperature and precipitation seasons**

The snow season was from October 1 to June 30 (inclusive). Snow duration was defined by snow
onset and snow ablation dates in situ observations (Bonsoms, 2021a), and results from the
baseline scenario snow duration presented in this work. A "compound temperature and
precipitation season" (season type) was assessed based on each massif and the elevation historical
climate record (1980–2019) using a joint quantile approach (Beniston and Goyette, 2007;
Beniston, 2009; López-Moreno et al., 2011a). Compound season types were defined according to
López-Moreno et al. (2011a), based on the seasonal $40^{th}$ percentiles (T40 for temperature and P40
for precipitation) and the seasonal $60^{th}$ percentiles (T60 and P60). There were four types of
seasons based on seasonal temperature (Tseason) and seasonal precipitation (Pseason) data:
Cold and Dry (CD): Tseason ≤ T40 and Pseason ≤ P40;
Cold and Wet (CW): Tseason ≤ T40 and Pseason ≥ P60;
Warm and Dry (WD): Tseason > T40 and Pseason ≤ P40;
Warm and Wet (WW): Tseason > T60 and Pseason is > P60.
All remaining seasons were classified as having average (Avg) temperature and precipitation.
Note that the number of compound season type is different depending on the Pyrenees massif
(Figure S1). However, by applying the joint-quantile approach described, we are comparing the
snow sensitivity to temperature and precipitation change between similar climate conditions,
independently where each compound season type was recorded.

**3.7 Spatial regionalization**

We have examined spatial differences in the snow sensitivity to temperature and precipitation
change by compound season types. Following previous studies, massifs were grouped into four
sectors by applying a Principal Component Analysis (PCA) (i.e., López-Moreno et al., 2020b;
Matiu et al., 2020, among others). We applied a PCA over HS data for each month, year, massif,
and elevation. Massifs were grouped into fours sectors depending on the maximum correlation to
PC1 and PC2 scores (see Figures S2). The number of season types per sector are shown at Figure
S3 and the spatial regionalization is presented at Figure 1.

**4. Results**

We validated the FSM2 at Section 4.1. Subsequently, we analyzed the snow sensitivity to
temperature and precipitation change based on five snow climate indicators, namely the seasonal
HS, peak HS max, peak HS date, snow duration and snow ablation. Compound season types show
similar relative importance on the snow sensitivity to temperature and precipitation change
regardless of the Pyrenean sector. For this reason, our results have been focused on seasonal snow
changes due to increments of temperature, elevation, and compound season type. These are the
key factors that ruled the snow sensitivity to temperature and precipitation change, and an accurate
analysis is provided at Section 4.2. Spatial differences on the snow sensitivity to temperature and
precipitation change during compound season types are examined at Section 4.3.

**4.1 Snow model validation**

Our snow model validation analysis (Figures 2 and 3) confirmed that FSM2 accurately reproduces
the observed HS values. On average, the FSM2 had a $R^2$ greater than 0.83 for all stations. In
general, the snow model slightly overestimated the maximum HS values. The highest $R^2$ values
were at A4 and A2 ($R^2 = 0.85$ in both stations), and the lowest were at A3 and A1 ($R^2 = 0.79$ and
$R^2 = 0.82$, respectively). The highest accuracy was at A4 (RMSE = 18.5 cm, MAE = 8.9 cm), and
the largest errors were at A2 (RMSE = 45.8 cm, MAE = 29.0 cm).

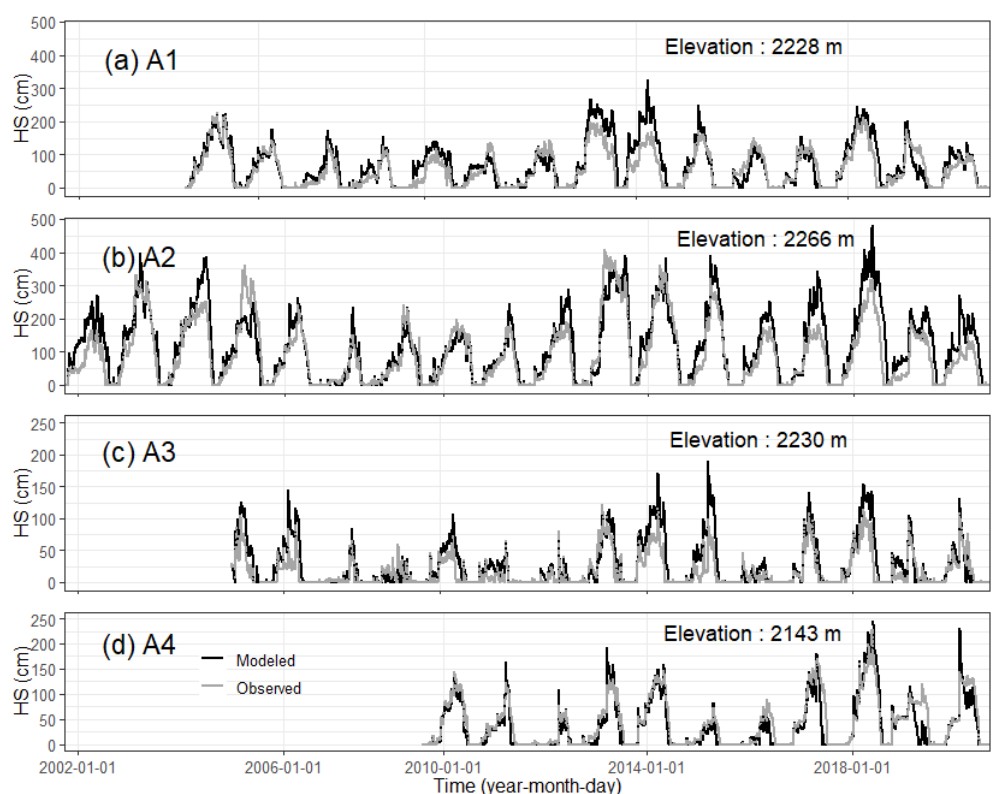


**Figure 2.** Time series of the observed (gray) and modelled (black) HS values at the four AWSs.




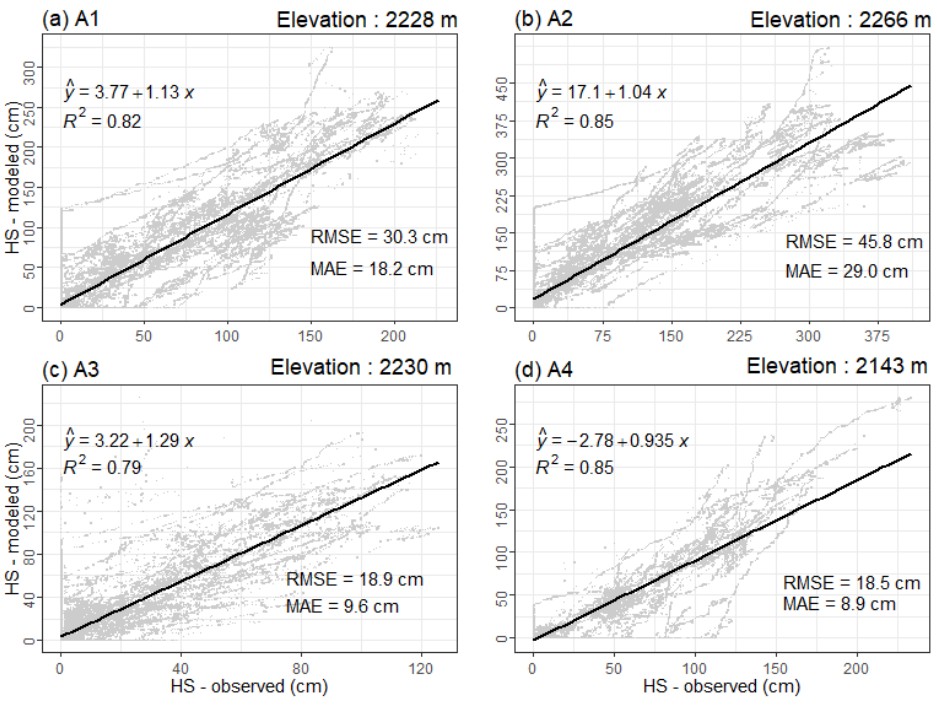


**Figure 3.** Regression analysis of observed (x-axis) and simulated (y-axis) HS values.



## 4.2 Snow sensitivity to temperature and precipitation change

We then determined seasonal HS profiles for each perturbed climate scenario and compound season type (Figure 4). The results show a non-linear response between seasonal HS loss and temperature increase. When the temperature increased at 1ºC intervals, the largest relative seasonal HS decrease from the baseline was at + 1ºC for all elevations and all compound season type. High elevation areas had lower seasonal HS variability between compound season types than low elevations (Figure S4). At low elevations, snow was greater during CW seasons than other seasons. All the snowpack-perturbed scenarios indicated that snowpack decreased for all elevations under warming climate scenarios. Snowpack sensitivity to temperature and precipitation change depended on the compound season type (Figures 5 and 6). At low elevations, the seasonal changes in HS ranged from −37% (WW) to −28% (CD) per ºC increase. For mid-elevation ranges, there were no remarkable differences among compound season types (Table 2), and the seasonal HS changes ranged from −34% (WW) to −30% (CW) per ºC increase. Low and mid-elevations had greater snowpack reductions than high elevations. In the latter, a 10% increase of precipitation counterbalanced a temperature increase of about 1ºC, and there were no remarkable differences in the seasonal HS from the baseline scenario especially in the coldest months of the season (Figure S5 and Figure S6). The maximum seasonal HS sensitivity to temperature and precipitation was during WD seasons (27%/ºC), and the minimum was during CW seasons (−22%/ºC).

334

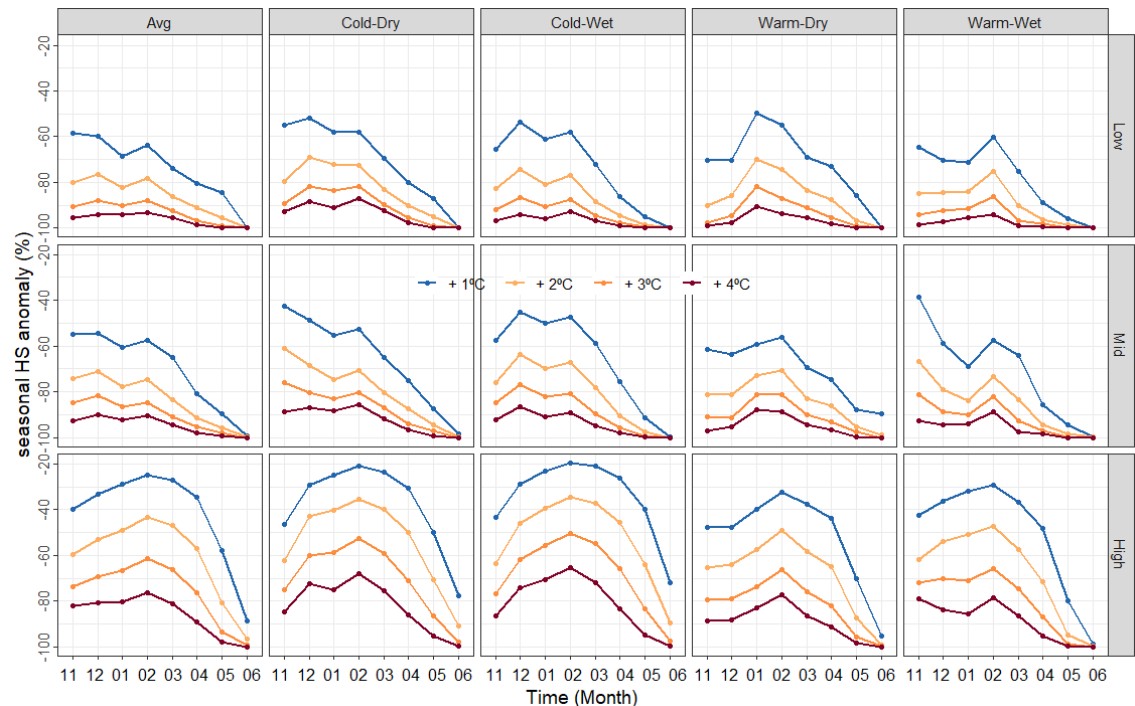

**Figure 4.** Anomalies of seasonal HS for low, mid and high elevation (rows), compound season type (columns), and different temperature increases (colors).

**Table 2.** Average and seasonal HS and peak HS sensitivity to temperature and precipitation change during the four different compound temperature and precipitation seasons at three different elevations.

| Season type | %HS/ ºC | | | %peak HS max/ºC | | |
|---|---|---|---|---|---|---|
| | Low | Mid | High | Low | Mid | High |
| Avg. | −33 | −33 | −25 | −20 | −20 | −16 |
| CD | −28 | −30 | −22 | −17 | −17 | −14 |
| CW | −33 | −32 | −22 | −22 | −20 | −15 |
| WD | −32 | −30 | −27 | −19 | −16 | −16 |
| WW | −37 | −34 | −26 | −24 | −24 | −16 |

340

At low and mid elevations, the peak HS max was greatest during WW seasons (−24%/ºC) and lowest during the CD and WD seasons (−17%/ºC for both). At high elevations, there were no clear differences in the peak HS max for the different seasons. The maximum peak HS max was during WD seasons (−16%/ºC) and the minimum was during CD seasons (−14%/ºC).

We also determined average seasonal snow duration for each elevation range and compound season type for different temperature increases (Table 3 and Figure 5c). The minimum snow duration was during CW seasons (−13%/ºC at low elevations, −10%/ºC at mid-elevations, −5%/ºC

at high elevations). At low elevations, the snow duration was most sensitive during WW seasons
(−17%/ºC). On the contrary, at mid-elevations and high elevations, the snow duration was most
sensitive during WD seasons (−13%/ºC at mid-elevations and −8%/ºC at high elevations).

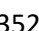


**Figure 5.** Anomalies of seasonal HS (a), peak HS max (b) and snow duration (c) for different temperature increases relative to baseline at three different elevations during the four different compound season types. The solid black lines within each boxplot are the average. Lower and upper hinges correspond to the 25th and 75th percentiles, respectively. The whisker is a horizontal line at 1.5 interquartile range of the upper quartile and lower quartile, respectively. Dots represent the outliers. Data is grouped by season, compound season type, increment of temperature, precipitation variation, elevation, and massif.

361

The peak HS date occurred earlier due to warming, independently of precipitation changes. During WD seasons, the peak HS date per °C was anticipated by 3 days at low elevations, 3 days at mid-elevations, and 6 days at high elevations; during CD seasons, the peak HS date per °C was anticipated by 4 days at low elevations, 5 days at mid-elevations, and 9 days at high elevations. In low and mid elevation areas, if the temperature increase was no more than about 1°C above baseline, there was little change in the peak HS date (Figure 6). In addition, the minimum peak HS date change is found during WW seasons (Table 3), because the snowpack would be scarce at those times, and there were no defined peaks (Figure S4).

We determined the snow ablation sensitivity to temperature and precipitation change in response to different temperature increases at different elevations and during different compound season types. The results show there were low differences in absolute snow ablation values in a warmer climate (Figure 7). At low elevations, the average snow ablation sensitivity to temperature and precipitation change in all four compound seasons was 12%/°C (Table 3). At mid-elevations and high elevations, the maximum snow ablation sensitivity to temperature and precipitation change was during dry seasons; WD seasons had a snow ablation sensitivity to temperature and precipitation change of 13%/°C at mid-elevations and 10%/°C at high elevations. On the other hand, the minimum values for mid-elevations were during WW seasons, when the snow ablation sensitivity to temperature and precipitation change was 8%/°C; the minimum values at high elevations were during CW seasons, when was 5%/°C.

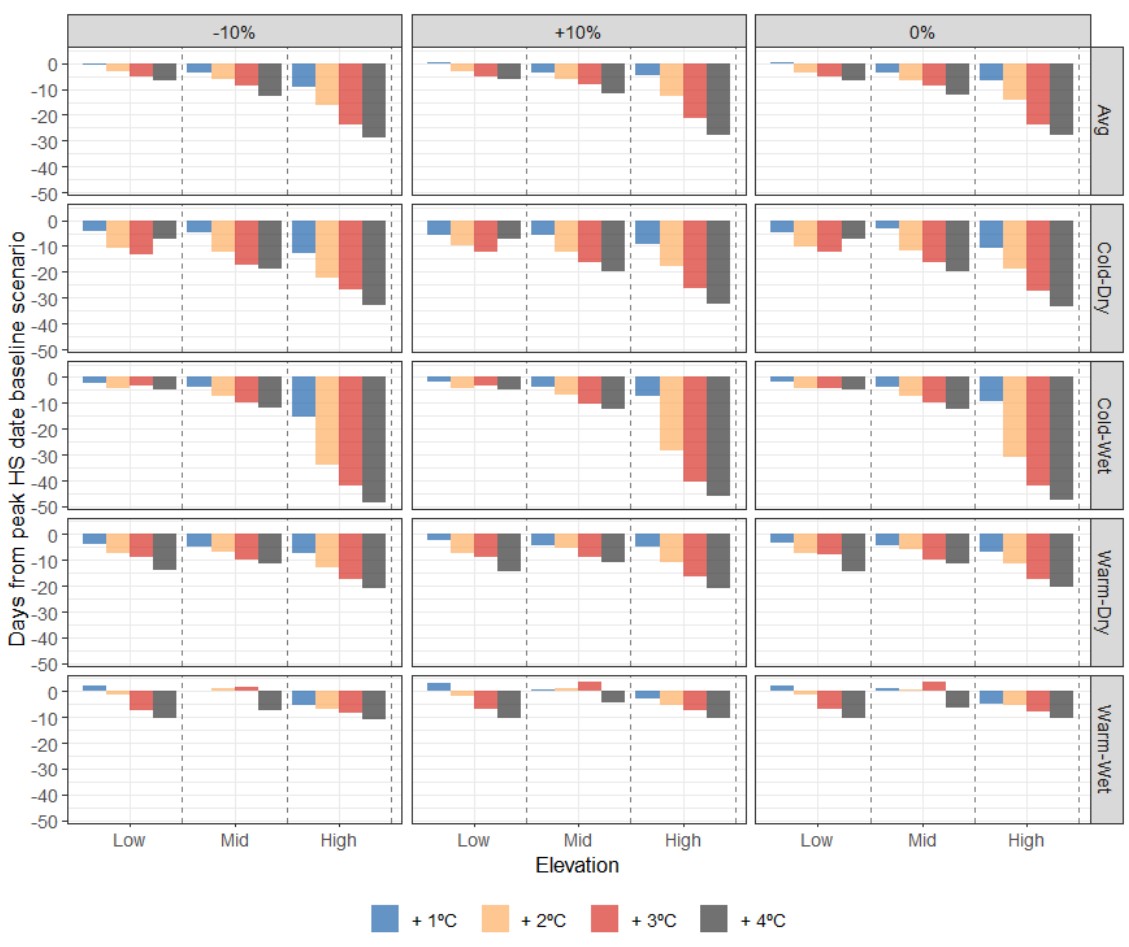


**Figure 6.** Difference (days) from baseline Peak HS date at three different elevations and during
the four different temperature (colors) and precipitation shifts (columns) for each season (boxes).


**Table 3.** Snow duration, snow ablation, and peak HS date sensitivity to temperature and
precipitation change during the four different compound season types.

| Season Type | Snow duration (%/°C) | | | Snow ablation (%/°C) | | | Peak HS date (days/°C) | | |
|---|---|---|---|---|---|---|---|---|---|
| | Low | Mid | High | Low | Mid | High | Low | Mid | High |
| Avg. | −15 | −12 | −6 | 12 | 11 | 7 | -1 | -3 | -7 |
| CD | −13 | −11 | −5 | 12 | 13 | 8 | -4 | -5 | -9 |
| CW | −13 | −10 | −5 | 12 | 10 | 5 | -2 | -3 | -13 |
| WD | −16 | −13 | −8 | 12 | 13 | 10 | -3 | -3 | -6 |
| WW | −17 | −13 | −7 | 12 | 8 | 7 | -1 | 0 | -3 |



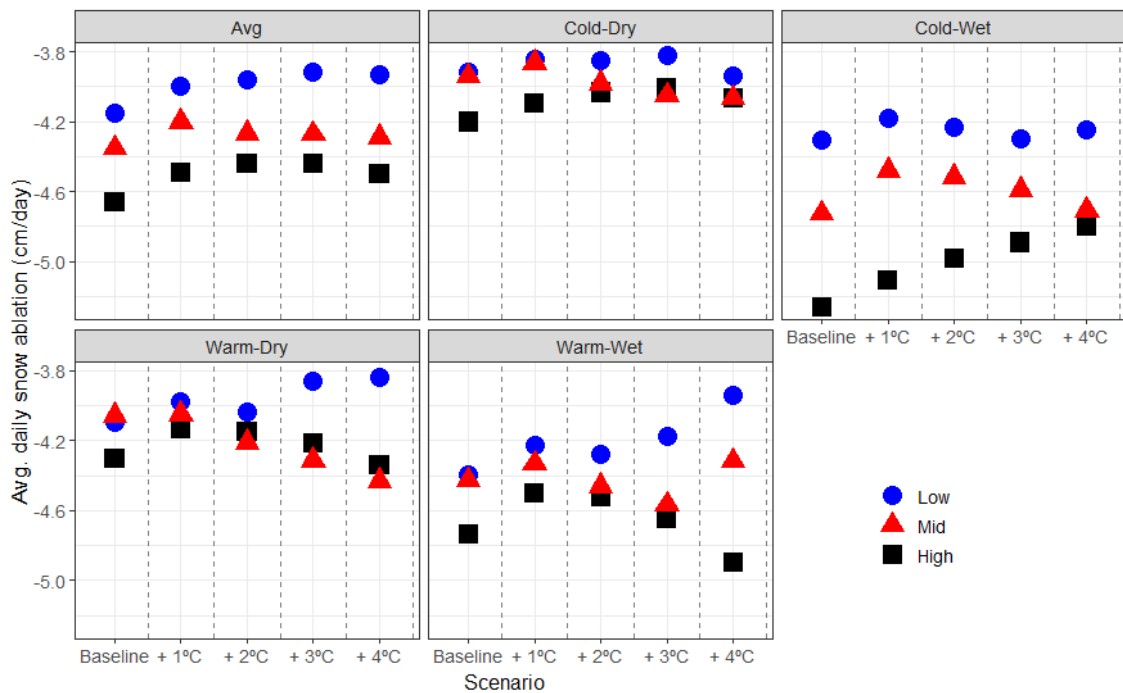

**Figure 7.** Absolute snow ablation values (cm/day) (y-axis) at three different elevations during four different compound temperature and precipitation for baseline and different increments of temperature (x-axis). seasons.

**4.3 Spatial patterns**

PCA analysis reveals four Pyrenean sectors, namely northern-western (NW), northern-eastern (NE), southern-western (SW), and southern-eastern (SE). No remarkable differences between sectors are found in the relative importance of each compound season type in the snow sensitivity to temperature and precipitation change (Figure 8). Snow sensitivity to temperature and precipitation change absolute values are generally lower at northern slopes (NW and NE) than at the southern slopes (SW and SE) (Figure S7 and Figure S8). In detail, seasonal HS ranged from −26%/°C during CD (NW) to −36%/°C during WW (SE). Similarly, the maximum peak HS max sensitivity to temperature and precipitation change was at SE during WW seasons (25%/°C) and the minimum was during CD seasons at NW (15%/°C). The snow duration sensitivity to temperature and precipitation change increased during WW seasons, and the maximum changes were at SE (−16%/°C); in contraposition, the lowest sensitivity to temperature and precipitation change are found at NW, during CD and CW seasons (−8%/°C, in both seasons). Snow ablation sensitivity to temperature and precipitation change increases towards the eastern Pyrenees, particularly during WD seasons (14%/°C and 13%/°C for NE and SE, respectively). Finally, no remarkable peak HS date differences are observed between sectors and maximum values are

found during CD and CW seasons, when the peak HS date is anticipated >= 5 per °C for all
sectors.

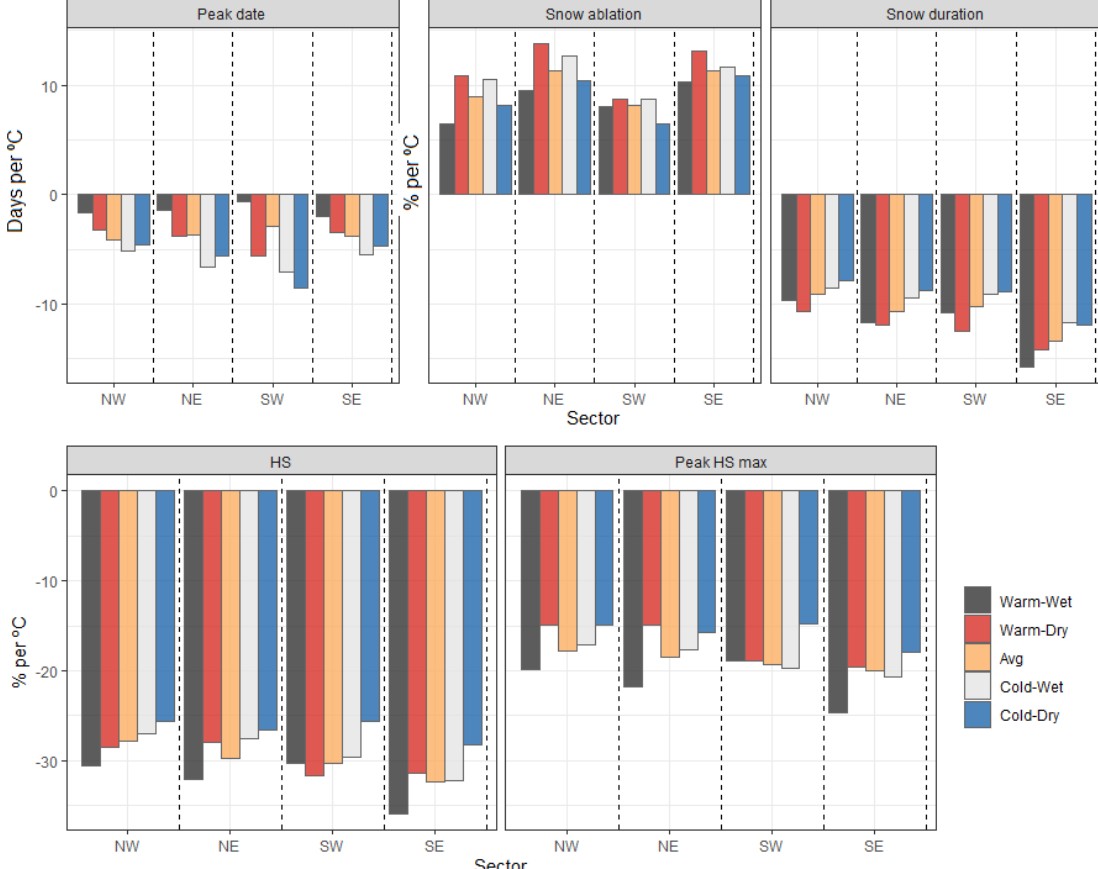


**Figure 8.** Average snow sensitivity to temperature and precipitation change (y-axis) grouped by
419         sector (x-axis), season type (color bars) and snow climate indicator (boxes).


**5. Discussion**

The spatial and temporal patterns of snow in the Pyrenees are highly variable, and climate
projections indicate that extreme events will likely increase during future decades (Meng et al.,
2022). Therefore, we analyzed factors that affect the snowpack sensitivity to temperature and
precipitation change gain insight into how future climate changes may affect the snow regime.

**5.1 Snow sensitivity to temperature and precipitation change and relationship with**
**historical and future snow trends**

**5.1.1 Snow accumulation phase**

The snow losses due warming that we described here are mainly associated with increases in the rain/snowfall ratio (Figure S9), changes in the snow onset and offset dates (Figure S4), and increases in the energy available for snow ablation during the later months of the snow season, as it was previously reported by literature (e.g., Pomeroy et al., 2015; Lynn et al., 2020; Jennings and Molotch, 2020). At high elevations, a trend of increasing precipitation (+10%) could counterbalance temperature increases (<1ºC; Figure S5), consistent with the results previously reported for specific sites of the central Pyrenees (Izas, 2000m; López-Moreno et al., 2008). Rasouli et al. (2014) also found that a 20% increase of precipitation could compensate for 2ºC increase of temperature in subarctic Canada. A climate sensitivity analysis in the western Cascades (western USA) found that increases of precipitation due to warming modulated the snowpack accumulation losses by about 5%/1°C (Minder, 2010). These results are consistent with recent data that examined snow above 1000 m in the Pyrenees, which found that an increase in the frequency of west circulation weather types since the 1980s increased the HS (Serrano-Notivoli et al., 2018; López-Moreno et al., 2020), snow accumulation (Bonsoms et al., 2021a), and changes in winter snow days (Buisan et al., 2016). There are similar trends in the Alps, with an increase of extreme (exceeding the 100-year return level) snowfall above 3000 m during recent decades (Roux et al., 2021) and increases in extreme winter precipitation (Rajczak and Schär, 2017).

451

**5.1.2 Snow ablation phase**

453

Climate warming leads to a cascade of physical changes in the SEB that increase snow ablation near the 0ºC isotherm. On overall, the snow ablation showed low to inexistent changes due to warming. Comparison between low and high elevations indicated slightly faster snow ablation at high elevations (Figure 7). This higher rate of snow ablation per season at high elevations (which have deeper snowpacks) are probably because the snow there lasts until late spring, when more energy is available for snow ablation (Bonsoms et al., 2022). Temperature increase does not imply remarkable changes in snow ablation per season because warming decreases the magnitude of the snowpack (seasonal HS and peak HS max) and triggers an earlier onset of snowmelt (Wu et al., 2018). The earlier peak HS date (Table 3 and Figure 6) implies lower rates of net shortwave radiation, because snow melting starts earlier in warmer climates (Pomeroy et al., 2015), coinciding with the shorter days and lower solar zenith angle (Lundquist et al., 2013; Sanmiguel-Vallelado et al., 2022). Our results agree with the slow snow melt rates reported in the Northern Hemisphere from 1980 to 2017 (Wu et al., 2018). The results of previous studies were similar for subarctic Canada (Rasouli et al., 2014) and western USA snowpacks (Musselman et al., 2017b), but Arctic sites had faster melt rates (Krogh and Pomeroy, 2019).

469

**5.1.3 Snow sensitivity to temperature and precipitation change and snowpack projections**

471

Our results suggest that warming had a non-linear effect on snowpack reduction. Our largest snow losses were for seasonal HS when the temperature increased by 1ºC above baseline. At low and mid elevations, the average seasonal HS decrease was more than 40% for all compound season types, and the maximum sensitivity was during WW seasons. Previous research in the Pyrenees and other mid-latitude mountain ranges reported similar results. A study in the central Pyrenees reported the peak SWE was 29%/ºC, whereas snow season duration decreased by about 20 to 30 days at about 2000 m (López-Moreno et al., 2013). The average peak HS max at high elevations in the Pyrenees (−16 %/ºC; Figure 6 and Table 2), was similar to the average peak SWE sensitivity (−15%/°C) reported in the Iberian Peninsula mountains at 2500 m (Alonso-González et al., 2020a). These results are also consistent with climate projections for this mountain range. In particular, for a 2ºC or more increase of temperature, the snow season declined by 38% at the lowest ski resorts (~1500 m) in the SE Pyrenees (Pons et al., 2015). However, high emission climate scenarios projected an increase in the frequency and intensity of high snowfall at high elevations (López-Moreno et al., 2011b). Snow sensitivity in the easternmost areas could decline during the winter because of a trend for an increase of about 10% in precipitation in this area (Amblar-Francés et al., 2020). Our projected changes in the Pyrenean snowpack dynamics are similar to the expected snow losses in other mountain ranges. For example, a study of the Atlas Mountains of northern Africa concluded that snowpack decreases were greater in the lowlands and projected seasonal SWE declines of 60% under the RCP4.5 scenario and 80% under the RCP8.5 scenario for the entire range (Tuel et al., 2022). A study in the Washington Cascades (western USA) found that snowpack decline was 19 to 23% per 1ºC (Minder, 2010), similar to the values in the present study at high elevations. A study of the French Alps (Chartreuse, 1500 m) found that seasonal HS decreases on the order of 25% for a 1.5ºC increase and 32% for a 2ºC increase of global temperature above the pre-industrial years (Verfaille et al., 2018). A study of the Swiss Alps reported a snowpack decrease of about 15%/ºC (Beniston, 2003); in the same alpine country, another study predicted the seasonal HS will decrease by more than 70% in massifs below 1000 m in all future climate projections (Marty et al., 2017). The largest snow reductions will likely occur during the periods between seasons (Steger et al., 2013; Marty et al., 2017). Nevertheless, at high elevations, snow climate projections found no significant trend for maximum HS until the end of the 21st century above 2500 m in the eastern Alps (Willibald et al., 2021), suggesting that internal climate variability is a major source of uncertainty of SWE projections at high elevations (Schirmer et al., 2021).

504

**5.2 Influence of compound temperature and precipitation seasons**

We found that the maximum sensitivities of seasonal HS and peak HS max to temperature and precipitation change were during WW seasons at low and mid-elevations and during WD seasons at high elevations. Brown and Mote (2008) analyzed the sensitivity of snow to climate changes in the Northern Hemisphere and found maximal SWE sensitivities in mid-latitudinal maritime winter climate areas, and minimal SWE sensitivities to temperature and precipitation change in dry and continental zones, consistent with our results. López-Moreno et al. (2017) also found greater decreases of SWE in wet and temperate Mediterranean ranges than in drier regions. Furthermore, Rasouli et al. (2022) studied the northern North American Cordillera and found higher snowpack sensitivities to temperature and precipitation change in wet basins than dry basins. Our maximum snow ablation relative change over the baseline scenario occurred during WD seasons, in accordance with Musselman et al. (2017b), who found a higher snowmelt rate during dry years in the western USA. Low and mid-elevations are highly sensitive to WW seasons because wet conditions favor decreases in the seasonal HS due to advection from sensible heat fluxes. The temperature in the Pyrenees is still cold enough to allow snowfall at high elevations during WW seasons, and for this reason we found maximal sensitivities to temperature and precipitation change during WD seasons. Reductions of snowfall in alpine regions can be compensated in a warmer scenario, because warm and wet snow is less susceptible to blowing wind transport and losses from sublimation (Pomeroy and Li, 2000; Pomeroy et al., 2015). During spring, snow runoff could be also greater in wet climates due to rain-on-snow events (Corripio et al., 2017), coinciding with the availability of more energy for snow ablation.

**5.3 Spatial and elevation factors controlling snow sensitivity to temperature and precipitation change**

Comparison between Pyrenean sectors (Figure 8) reveals no remarkable differences in the relative importance of each compound season type in the snow sensitivity to temperature and precipitation change. This is because by applying a joint-quantile approach for each massif and elevation, we are comparing similar climate seasons between sectors, regardless of the number of compound season types recorded in each massif during the baseline period (Figure S1 and S3). The highest absolute snow sensitivity to temperature and precipitation change values is found in the SE Pyrenees. This is consistent with the snow accumulation and ablation patterns previously reported in this region (Lopez-Moreno, 2005; Navarro-Serrano et al., 2018; Alonso-González et al., 2020a; Bonsoms et al., 2021a; Bonsoms et al., 2021b; Bonsoms et al., 2022). The Atlantic climate has less of an influence in the SE sector, and in situ observations indicated there was about half of the

seasonal snow accumulation amounts as in northern and western areas at the same elevation (>2000 m; Bonsoms et al., 2021a). The snow in the SE Pyrenees is more sensitive to temperature and precipitation change because these massifs are exposed to higher turbulence and radiative heat fluxes (Bonsoms et al., 2022), and 0ºC isotherm is closer. Similar conclusions are found for low elevations, where the results show an upward displacement of the snow line due to warming. Previous studies described the sensitivity of the snow pattern to elevation at specific stations of the central Pyrenees (López-Moreno et al., 2013; 2017), Iberian Peninsula mountains (Alonso-González et al., 2020a), and other ranges such as the Cascades (Jefferson, 2011; Sproles et al., 2013), the Alps (Marty et al., 2017), and western USA (Pierce et al., 2013; Musselman et al., 2017b). In these regions, the models suggest larger snowpack reductions due to warming at subalpine sites than at alpine sites (Jennings and Molotch, 2020) due to closer isothermal conditions (Brown and Mote, 2009; Lopez-Moreno et al., 2017; Mote et al., 2018).

**5.4 Environmental and socioeconomic implications**

Our results indicated there will be an increase of snow ablation days and imply a disappearance of the typical sequence of snow accumulation and snow ablation seasons. Climate warming triggers the simultaneous occurrence of several periods of snowfall and melting, snow droughts during the winter, and ephemeral snowpacks between seasons. These expected decreases in snow will likely have important impacts on the ecosystem. During spring, a snow cover cools the soil (Luetschg et al., 2008), delays the initiation of freezing (Oliva et al., 2014), functions as a thick active layer (Hrbáček et al., 2016), and protects alpine rocks from exposure to solar radiation and high air temperatures (Magnin et al., 2017). Due to warming temperatures, the remaining glaciers in the Pyrenees are shrinking and are expected to disappear before the 2050s (Vidaller et al., 2021). The shallower snowpack that we identified in this work will increase the vulnerability of glaciers, because snow has a higher albedo than dark ice and debris-covered glaciers and functions as a protective layer for glaciers (Fujita and Sakai, 2014).

The earlier onset of snowmelt suggested by our results, which is greater at low and mid-elevations during WD seasons, is in line with previous global studies that reported earlier streamflow due to earlier runoff dates (Adam et al., 2009; Stewart, 2009), and with a study of changes in the Iberian Peninsula River flows (Morán-Tejeda et al., 2014). Overall, our results are consistent with the slight decrease of the river peak flows that have occurred in the southern slopes of the Pyrenees since the 1980s (Sanmiguel-Vallelado et al., 2017). The reductions of seasonal HS that we identified , suggest that snowmelt-dominated stream flows will likely shift to rainfall dominated regimes. Although high elevation meltwater might increase and contribute to earlier groundwater

recharging (Evans et al., 2018), the increased evapotranspiration in the lowlands (Bonsoms et al.,
2022) could counter this effect, so there is no net change in downstream areas (Stahl et al., 2010).
Snow ephemerality triggers lower spring and summer flows (Barnett et al., 2005; Adam et al.,
2009; Stahl et al., 2010) and has an impact on the hydrological management strategies. Winter
snow accumulation affects hydrological availability during the months when water and
hydroelectric demands are higher. This is because reservoirs store water during periods of peak
flows (winter and spring), and release water during the driest season in the lowlands (summer)
(Morán-Tejeda et al., 2014). Recurrent snow-scarce seasons may intensify these hydrological
impacts and lead to competition for water resources among different ecological and
socioeconomic systems. The economic viability of mountain ski-resorts in the Pyrenees depends
on a regular deep snow cover (Gilaberte-Burdalo et al., 2014; Pons et al., 2015), but this is highly
variable, especially at low and mid-elevations. The expected increase in snow-scarce seasons that
we identified here is consistent with climate projections for this region, which suggest that no
Pyrenean ski resorts will be viable under RCP 8.5 scenario by the end of the 21[st] century (Spandre
et al., 2019).

## 5.5 Limitations and uncertainties

The meteorological input data that we used to model snow were estimated for flat slopes and the
regionalization system we used was based on the SAFRAN system. According to this system, a
mountain range is divided into massifs with homogeneous topography. The SAFRAN system has
negative biases in shortwave radiation, a temperature precision of about 1 K, and biases in the
accumulated monthly precipitation of about 20 kg/m$^2$ (Vernay et al., 2021). Our estimates of snow
sensitivity to temperature and precipitation change were based on the delta-approach, which
considers changes in temperature and precipitation based on climate projections for the Pyrenees
(Amblar-Francés et al., 2020), but assumes that the meteorological patterns of the reference period
will be constant over time. In this work we used a physical-based snow model since it provides
better results for future snow climate change estimations than degree-day models (Carletti et al.,
2022). The FSM2 is a physics-based model of intermediate complexity, and the estimates of snow
densification are simpler than those from more complex models of snowpack. However, a more
complex model does not necessarily provide better performance in terms of snowpack and runoff
estimation (Magnusson et al., 2015). The FSM2 configuration implemented in this work includes
snow meltwater retention, snowpack refreezing and snow albedo based on snow age, which are
the physical parameters included in the best-performing snow models according to Essery et al.
(2013). Snow model sensitivity studies reveal that intermediate complexity models exhibit similar
snow depth accuracies than most complex multi-layer snow models, as well as robust
performances across seasons (Terzago et al., 2020).

**6 Conclusions**

Our study assessed the impact of temperature and precipitation change on the Pyrenean snowpack
during compound cold-hot and wet-dry seasons, using a physical-based snow model that was
forced by reanalysis data. We determined the snow sensitivity to temperature and precipitation
change using five key indicators of snow accumulation and snow ablation. The lowest snow
sensitivity to temperature and precipitation change was at high elevations of the NW Pyrenees
and increased at lower elevations and in the SE slopes. An increase of 1ºC at low and mid
elevation regions led to remarkable decreases in the seasonal HS and snow duration. However, at
high elevations, precipitation plays a key role, and temperature is far from the isothermal 0ºC
during the middle of winter. In this region, a 10% increase of precipitation, as suggested by the
Spanish Meteorological Agency (AEMET) over the eastern regions of this range, could
compensate for temperature increases on the order of about < 1ºC. The impact of climate warming
depends on the combination of temperature and precipitation during compound seasons. Our
analysis of seasonal HS and peak HS max indicated the greatest declines were during WW seasons
and the smallest declines were during CD seasons, independently of the Pyrenean sector. For
snow duration, however, the highest (lowest) sensitivity to temperature and precipitation change
is found during WD (CW) seasons. Similarly, snow ablation had slightly greater sensitivities to
temperature and precipitation change during WD seasons, in that snow ablation variation is less
than 10% and the peak HS date occurred about 5 days earlier per ºC. Our findings thus provide
evidence that the Pyrenean snowpack is highly sensitive to climate warming and suggest that the
snowpacks of other mid-latitude mountain ranges may also show similar response to warming.

**Data availability**

Snow model (FSM2) is open access and provided by Essery (2015) and available at
https://github.com/RichardEssery/FSM2 (last access 16 December 2022). Climate forcing data is
provided by Vernay et al. (2021), through AERIS (https://www.aeris-data.fr/landing-
page/?uuid=865730e8-edeb-4c6b-ae58-80f95166509b#v2020.2; last access 16 December 2022).
Data of this work is available upon request (contact: josepbonsoms5@ub.edu).

**Acknowledgements**

This work falls within the research topics examined by the research group "Antarctic, Artic,
Alpine Environments-ANTALP" (2017-SGR-1102) funded by the Government of Catalonia,
HIDROIBERNIEVE (CGL2017-82216-R) and MARGISNOW (PID2021-124220OB-100), from
the Spanish Ministry of Science, Innovation and Universities. JB is supported by a pre-doctoral
University Professor FPI grant (PRE2021-097046) funded by the Spanish Ministry of Science,
Innovation and Universities. The authors are grateful to Marc Oliva, who reviewed an early
version of this manuscript. We acknowledge the SAFRAN data provided by Météo-France –
CNRS and the CNRM Centre d'Etudes de la Neige, through AERIS.

**Author contributions**

JB analyzed the data and wrote the original draft. JB, JILM and EAG contributed to the
manuscript design and draft editing. JB, JILM and EAG read and approved the final manuscript.

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

sensitivity to temperature and precipitation during four different compounds weather conditions
(cold-dry [CD], cold-wet [CW], warm-dry [WD], and warm-wet [WW]) at low elevations (1500
m), mid-elevations (1800 m), and high elevations (2400 m) in the Pyrenees. In particular, we
forced a physically based energy and mass balance snow model (FSM2), with validation by
ground-truth data, and applied this model to the entire range, with forcing of perturbed reanalysis
climate data for the period 1980 to 2019 as the baseline. The FSM2 model results successfully
reproduced the observed snow depth (HS) values ($R^2 > 0.8$), with relative root-mean square error
and mean absolute error values less than 10% of the observed HS values. Overall, the snow
sensitivity to temperature and precipitation change decreased with elevation and increased
towards the eastern Pyrenees. When the temperature increased progressively at 1ºC intervals, the
largest seasonal HS decreases from the baseline were at +1ºC. ~~(47% at low elevation, 48% at~~
~~mid elevation, and 25% at high elevation).~~ A 10% increase of precipitation counterbalanced the
temperature increases (≤1ºC) at high elevations during the coldest months, because temperature
was far from the isothermal 0ºC conditions. The maximal seasonal HS and peak HS max
reductions were during WW seasons, and the minimal reductions were during CD seasons. During
WW (CD) seasons, the seasonal HS decline per °C was 37% (28 %) at low elevations, 34% (30%)
at mid elevations, and 27% (22 %) at high elevations. Further, the peak HS date was on average
anticipated 2, 3 and 8 days at low, mid and high elevation, respectively. Results suggest~~s~~ snow
sensitivity to temperature and precipitation change will be similar at other mid-latitude mountain
areas, where snowpack reductions will have major consequences on the nearby ecological and
socioeconomic systems.
**Keywords:** Snow, Climate change, Sensitivity, Alpine, Mediterranean Mountains, Mid-latitude,
Pyrenees.

## 1    Introduction

Snow is a key element of the Earth's climate system (Armstrong and Brun, 1998) because it cools
the planet (Serreze and Barry, 2011) by altering the Surface Energy Balance (SEB), increasing
the albedo, and modulating surface and air temperatures (Hall, 2004). Northern-Hemispheric
snowpack patterns have changed rapidly during recent decades (Hammond et al., 2018; Hock et
al., 2019; Notarnicola et al., 2020). It is crucial to improve our understanding of the timing of
snow ablation and snow accumulation due to changing climate conditions because snowpack
affects many nearby social and environmental systems. From the hydrological point of view, snow
melt controls mountain runoff rate during the spring (Barnett et al., 2005; Adams et al., 2009;
Stahl et al., 2010), river flow magnitude and timing (Morán-Tejeda et al., 2014; Sanmiguel-
Vallelado et al., 2017), water infiltration and groundwater storage (Gribovszki et al., 2010; Evans
et al., 2018), and transpiration rate (Cooper et al., 2020). The presence and duration of snowpack
affects terrestrial ecosystem dynamics because snow ablation date affects photosynthesis
(Woelber et al., 2018), forest productivity (Barnard et al., 2018), freezing and thawing of the soil
(Luetschg et al., 2008; Oliva et al., 2014), and thickness of the active layer in permafrost
environments (Hrbáček et al., 2016; Magnin et al., 2017). Snowpack also has remarkable
economic impacts. For example, the snowpack at high elevations and surrounding areas
determines the economic success of many mountain ski-resorts (Scott et al., 2003; Pons et al.,
2015; Gilaberte-Búrdalo et al., 2017). Changes in the snowpack of mountainous regions also
influence associated lowland areas because it affects the availability of snow meltwater that is
used for water reservoirs, hydropower generation, agriculture, industries, and other applications
(e.g., Sturm et al., 2017; Beniston et al., 2018).
Mid-latitude snowpacks have among the highest snow sensitivities worldwide (Brown and Mote,
2009; López-Moreno et al., 2017; 2020b). In regions at high latitudes or high elevations,
increasing precipitation can partly counterbalance the effect of increases of temperature on snow
cover duration (Brown and Mote, 2009). Climate warming decreases the maximum and seasonal
snow depth (HS), the snow water equivalent (SWE) (Trujillo and Molotch, 2014; Alonso-
González et al., 2020a; López-Moreno et al., 2013; 2017), and the fraction total precipitation as
snowfall (snowfall ratio; e.g., Mote et al., 2005; Lynn et al., 2020; Jeenings and Molotoch, 2020;
Marshall et al., 2019), and also delays the snow onset date (Beniston, 2009; Klein et al., 2016).
However, warming can slow the early snow ablation rate on the season (Pomeroy et al., 2015;
Rasouli et al., 2015; Jennings and Molotch, 2020; Bonsoms et al., 2022; Sanmiguel-Vallelado et
al., 2022) because of the earlier HS and SWE peak dates (Alonso-González et al., 2022), which
coincide with periods of low solar radiation (Pomeroy et al., 2015; Musselman et al., 2017a).
The Mediterranean basin is a region that is critically affected by climate change (Giorgi, 2006)
being densely populated (>500 million inhabitants) and affected by an intense anthropogenic
activity. Warming of the Mediterranean basin will accelerate for the next decades, and
temperatures will continue to increase in this region during the warm months (Knutti and
Sedlacek, 2013; Lionello and Scarascia 2018; Cramer et al., 2018; Evin et al., 2021; Cos et al.,
2022), increasing atmospheric evaporative demands (Vicente-Serrano et al., 2020), drought
severity (Tramblay et al., 2020), leading to water-scarcity over most of this region (García-Ruiz
et al., 2011). Mediterranean mid-latitude mountains, such as the Pyrenees, where this research
focuses, are the main runoff generation zones of the downstream areas (Viviroli and Weingartner,
2004) and provide most of the water used by major cities in the lowlands (Morán-Tejeda et al.,

80  2014).


Snow patterns in the Pyrenees have high spatial diversity (Alonso-González et al., 2019), due to
internal climate variability of mid-latitude precipitation (Hawkins and Sutton 2010; Deser et al.,
2012), high interannual and decadal variability of precipitation in the Iberian Peninsula (Esteban-
Parra et al., 1998; Peña-Angulo et al., 2020) as well as the abrupt topography and the different
mountain exposure to the Atlantic air masses (Bonsoms et al., 2021a). Thus, snow accumulation
per season is almost twice as much in the northern slopes as in the southern slopes (Navarro-
Serrano and López-Moreno, 2017), and there is a high interannual variability of snow in regions
at lower elevations (Alonso-González et al., 2020a) and in the southern and eastern regions of the
Pyrenees (Salvador-Franch et al., 2014; Salvador-Franch et al., 2016; Bonsoms et al., 2021b).
Since the 1980s, the energy available for snow ablation has significantly increased in the Pyrenees
(Bonsoms et al., 2022), and winter snow days and snow accumulation have non-statically
significantly increased (Buisan et al., 2016; Serrano-Notivoli et al., 2018; López-Moreno et al.,
2020a; Bonsoms et al., 2021a) due to the increasing frequency of positive west and south-west
advections (Buisan et al., 2016). 21$^{st}$ century climate projections for Pyrenees anticipate a
temperature increase of more than 1°C to 4°C (relative to 1986–2005), and an increase (decrease)
of precipitation by about 10% for the eastern (western) regions during winter and spring (Amblar-
Frances et al., 2020). Therefore, changes in snow patterns in regions with high elevations are
uncertain because winter snow accumulation is affected by precipitation (López-Moreno et al.,
2008) and Mediterranean basin winter precipitation projections have uncertainties up to 80% of
the total variance (Evin et al., 2021).

Previous studies in the central Pyrenees (López-Moreno et al., 2013), Iberian Peninsula Mountain
ranges (Alonso-González et al., 2020a), and mountain areas that have Mediterranean climates
(López-Moreno et al., 2017) demonstrated that snowpack sensitivity to changes in climate ~~are~~
~~mostly~~are mostly controlled by elevation. Despite the impact of climate warming in mountain
hydrological processes, there is limited understanding of the snow sensitivity to temperature and
precipitation changes and seasonality of mid-latitude Mediterranean mountain snowpacks. Some
studies reported different snowpack sensitivities during wet and dry years (López-Moreno et al.,
2017; Musselman et al., 2017b; Rasouli et al., 2022; Roche et al., 2018). However, the sensitivity
of snow during periods when there are seasonal compound weather (temperature and
precipitation) conditions has not yet been analyzed. The high interannual variability of the
Pyrenean snowpack, which is expected to increase according to climate projections (López-
Moreno et al., 2008), indicates a need to examine snowpack sensitivity to temperature and
precipitation change focusing on the year-to-year variability. Warm seasons in the Mediterranean
basin require special attention because these are likely to increase in the future (e.g., Vogel et al.,
2019; De Luca et al., 2020; Meng et al., 2022). Further, the occurrence of different HS trends at
mid- and high-elevation areas of this range (López-Moreno et al., 2020a) suggest that elevation
and spatial factors contribute to the wide variations of the sensitivity of snow to the climate.

Therefore, the main objective of this research is to quantify snow (accumulation, ablation, and
timing) sensitivity to temperature and precipitation change during compound temperature and
precipitation seasons in the Pyrenees.

**2 Geographical area and climate setting**
The Pyrenees is a mountain range located in the north of the Iberian Peninsula (south Europe;
42ºN-43ºN to 2ºW-3ºE) that is aligned east-to-west between the Atlantic Ocean and the
Mediterranean Sea. The highest elevations are in the central region (Aneto, 3404 m) and
elevations decrease towards the west and east (Figure 1). The Mediterranean basin, including the
Pyrenees, is in a transition area, and is influenced by the continental climate and the subtropical
temperate climate. Precipitation is mostly driven by large-scale circulation patterns (Zappa et al.,
2015; Borgli et al., 2019), the jet-stream oscillation during winter (Hurell, 1995), and land-sea
temperature differences (Tuel and Eltahir, 2020). During the summer, the northward movement
of the Azores high pressure region brings stable weather, and precipitation is mainly convective
at that time (Xercavins, 1985). Precipitation is highly variable depending on mountain exposure
to the main circulation weather types; it ranges from about 1000 mm/year to about 2000 mm/year
(in the mountain summits), with lower levels in the east and south (Cuadrat et al., 2007). There is
a slight disconnection of the general climate circulation towards the eastern Pyrenees, where the
Mediterranean climate and East Atlantic/West Russia (EA-WR) oscillations have greater effects
on snow accumulation (Bonsoms et al., 2021a). In the southern, western, and central massifs of
the range, the Atlantic climate and the negative North Atlantic Oscillation (NAO) phases regulate
snow accumulation (W and SW wet air flows; López-Moreno, 2005; López-Moreno and Vicente-
Serrano, 2007; Buisan et al., 2016; Alonso-González et al., 2020b). In the northern slopes, the
positive phases of the Western Mediterranean Oscillation (WeMO) linked with NW and N
advections trigger the most episodes of snow accumulation (Bonsoms et al., 2021a). The seasonal
snow accumulation in the northern slopes is almost double the amount (about 500 cm more) as in
the southern slopes at an elevation of about 2000 m (Bonsoms et al., 2021a). The
temperature/elevation gradient is about 0.55°C/100 m (Navarro-Serrano and López-Moreno,
2018) and the annual 0°C isotherm is at about 2750 to 2950 m (López-Moreno and García-Ruiz,
2004). Net radiation and latent heat flux governs the energy available for snow ablation; the
former heat flux increases at high elevations and the latter towards the east (Bonsoms et al., 2022).

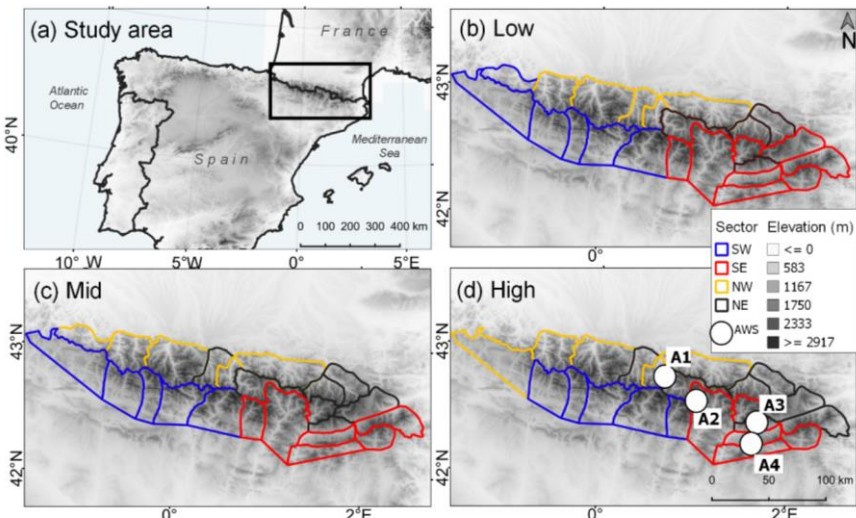

**Figure 1** (a) Study area. Pyrenean massifs grouped by sectors for (b) low, (c) mid and (d) high elevation. The white dots indicate the locations of the automatic weather stations (AWS) shown at Table 1. Massifs delimitation is based on the spatial regionalization of the SAFRAN system, which groups massifs according to topographical and meteorological characteristics (modified from Durand et al., 1999).

**3 Data and methods**

**3.1 Snow model**

Snowpack was modelled using a physical-based snow model, the Flexible Snow Model (FSM2; Essery, 2015). This model resolves the SEB and mass balance to simulate the state of the snowpack. FSM2 is open access and available at https://github.com/RichardEssery/FSM2 (last access 16 December 2022). Previous studies tested the FSM2 (Krinner et al., 2018), and its application in different forest environments (Mazzoti et al., 2021), and hydro-climatological mountain zones such the Andes (Urrutia et al., 2019), Alps (Mazzoti et al., 2020), Colorado (Smyth et al., 2022), Himalayas (Pritchard et al., 2020), Iberian Peninsula Mountains (Alonso-González et al., 2020a; Alonso-González et al., 2022), Lebanese mountains (Alonso-González et al., 2021), providing confidential results. The FSM2 requires forcing data of precipitation, air temperature, relative humidity, surface atmospheric pressure, wind speed, incoming shortwave radiation ($SW_{inc}$), and incoming long wave radiation ($LW_{inc}$). We have evaluated different FSM2 model configurations (not shown) without remarkable differences in the accuracy and

performance metrics. Thus, the FSM2 configuration included in this work estimates snow cover fraction based on a linear function of HS and albedo based on a prognostic function, with increases due to snowfall and decreases due to snow age. Atmospheric stability is calculated as function of the Richardson number. Snow density is calculated as a function of viscous compaction by overburden and thermal metamorphism. Snow hydrology is estimated by gravitational drainage, including internal snowpack processes, runoff, refreeze rates, and thermal conductivity. Table S1 summarizes the FSM2 configuration and the FSM2 compile numbers.

**3.2 Snow model validation**

FSM2 configuration was validated by in situ snow records of four automatic weather stations (AWSs) that were at high elevations in the Pyrenees. Precipitation in mountainous and windy regions is usually affected by undercatch (Kochendorfer et al., 2020). Thus, the instrumental records of precipitation were corrected for undercatch by applying an empirical equation validated for the Pyrenees (Buisan et al., 2019). Precipitation type was classified by a threshold method (Musselman et al., 2017b; Corripio et al., 2017): snow when the air temperature was below 1°C and rain when the air temperature was above 1°C, according to previous research in the study area (Corripio et al., 2017). The $LW_{inc}$ heat flux of the AWSs (Table 1) were estimated as previously described (Corripio et al., 2017). Due to the wide instrumental data coverage (99.3% of the total dataset), gap-filling was not performed. The HS records were measured each 30 min using an ultrasonic snow depth sensor. The meteorological data used in the validation process were provided and managed by the local meteorological service of Catalonia (https://www.meteo.cat/wpweb/serveis/formularis/peticio-dinformes-i-dades-meteorologiques/peticio-de-dades-meteorologiques/; data requested: 14/01/2021). Quality-checking of the data was performed using an automatic error filtering process in combination with a climatological, spatial, and internal coherency control defined by the SMC (2011).

**Table 1.** Characteristics of the four AWSs.

| Area | Code | Lat/Lon° | Elevation (m) | Atlantic Ocean, Distance (km) | Mediterranean Sea, Distance (km) | Validation period (years) | Years |
|---|---|---|---|---|---|---|---|
| Central-Pyrenees, Northern slopes | A1 | 42.77/0.73 | 2228 | 200 | 190 | 2004–2020 | 16 |
| | A2 | 42.61/0.98 | 2266 | 225 | 170 | 2001–2020 | 19 |
| Eastern Pyrenees, Southern slopes | A3 | 42.46/1.78 | 2230 | 295 | 115 | 2005–2020 | 15 |

| Eastern Pre-Pyrenees, Northern slopes | A4 | 42.29/1.71 | 2143 | 300 | 110 | 2009–2020 | 11 |

Model accuracy was estimated based on the mean absolute error (MAE) and the root mean square error (RMSE), and model performance was estimated by the coefficient of determination ($R^2$). The MAE and the RMSE indicate the mean differences of the modelled and observed values.

**3.3 Atmospheric forcing data**

We forced the FSM2 with the open access climate reanalysis dataset provided by Vernay et al. (2021), which consists of the modelled values from the SAFRAN meteorological analysis. The FSM2 was run at an hourly resolution for each massif, each elevation range, and each climate baseline and perturbation scenario from 1980 to 2019. The SAFRAN system provides data for homogeneous meteorological and topographical mountain massifs every 300 m, from 0 to 3600 m (Durand et al., 1999; Vernay et al., 2021). We analyzed three elevation bands: low (1500 m), middle (1800 m), and high (2400 m). Precipitation type was classified using the same threshold approach used for model validation. Atmospheric emissivity was derived from the SAFRAN $LW_{inc}$ and air temperature. SAFRAN was forced using numerical weather prediction models (ERA-40 reanalysis data from 1958 to 2002 and ARPEGE from 2002 to 2020). Meteorological data were calibrated, homogenized, and improved by in situ meteorological observations data assimilation (Vernay et al., 2021). Durand et al. (1999; 2009a; 2009b) provided further technical details of the SAFRAN system. Previous studies used the SAFRAN system for the long-term HS trends (López-Moreno et al., 2020), extreme snowfall (Roux et al., 2021), and snow ablation analysis (Bonsoms et al., 2022). SAFRAN system has been extensively validated for the meteorological modelling of continental Spain (Quintana-Seguí et al., 2017), France (Vidal et al., 2010) or alpine snowpack climate projections (Verfaille et al., 2018), among other works.

**3.4 Snow sensitivity to temperature and precipitation change analysis**

Snow sensitivity to temperature and precipitation change was analyzed using a delta-change methodology (López-Moreno et al., 2008; Beniston et al., 2016; Musselman et al., 2017b; Marty et al., 2017; Alonso-González et al., 2020a; Sanmiguel-Vallelado et al., 2022). In this method, air temperature and precipitation were perturbed for each massif and elevation range based the historical period (1980–2019). Air temperature was increased from 1 to 4ºC at 1ºC intervals, assuming an increase of $LW_{inc}$ accordingly (Jennings and Molotch, 2020). Precipitation was

changed from −10% to +10% at 10% intervals, in accordance with climate model uncertainties
and the maximum and minimum precipitation projections for the Pyrenees (Amblar-Frances et
al., 2020).

**3.5 Snow climate indicators**

Snow sensitivity to temperature and precipitation change was analyzed using five key indicators:
(*i*) seasonal average HS, (*ii*) seasonal maximum absolute HS peak (peak HS max), (*iii*) date of the
maximum HS (peak HS date), (*iv*) number of days with HS > 1 cm on the ground (snow duration),
and (*v*) daily average snow ablation per season (snow ablation, hereafter). Snow ablation was
calculated as the difference between the maximum daily HS recorded on two consecutive days
(Musselman et al., 2017a), and only days with decreases of 1 cm or more were recorded. Some
seasons had more than one peak HS; for this reason, peak HS date was determined after applying
a moving average of 5-days. All indicators were computed according to massif and elevation
range.

**3.6 Definitions of compound temperature and precipitation seasons**

The snow season was from October 1 to June 30 (inclusive). Snow duration was defined by snow
onset and snow ablation dates in situ observations (Bonsoms, 2021a), and results from the
baseline scenario snow duration presented in this work. A "compound temperature and
precipitation season" (season type) was assessed based on each massif and the elevation historical
climate record (1980–2019) using a joint quantile approach (Beniston and Goyette, 2007;
Beniston, 2009; López-Moreno et al., 2011a). Compound season types were defined according to
López-Moreno et al. (2011a), based on the seasonal 40[th] percentiles (T40 for temperature and P40
for precipitation) and the seasonal 60[th] percentiles (T60 and P60). There were four types of
seasons based on seasonal temperature (Tseason) and seasonal precipitation (Pseason) data:
Cold and Dry (CD): Tseason ≤ T40 and Pseason ≤ P40;
Cold and Wet (CW): Tseason ≤ T40 and Pseason ≥ P60;
Warm and Dry (WD): Tseason > T40 and Pseason ≤ P40;
Warm and Wet (WW): Tseason > T60 and Pseason is > P60.
All remaining seasons were classified as having average (Avg) temperature and precipitation.
Note that the number of compound season type is different depending on the Pyrenees massif
(Figure S1). However, by applying the joint-quantile approach described, we are comparing the
snow sensitivity to temperature and precipitation change between similar climate conditions,
independently where each compound season type was recorded.

### 3.7 Spatial regionalization

We have examined spatial differences in the snow sensitivity to temperature and precipitation change by compound season types. Following previous studies, massifs were grouped into four sectors by applying a Principal Component Analysis (PCA) (i.e., López-Moreno et al., 2020b; Matiu et al., 2020, among others). We applied a PCA over HS data for each month, year, massif, and elevation. Massifs were grouped into fours sectors depending on the maximum correlation to PC1 and PC2 scores (see Figures S2). Massifs were grouped into four sectors by applying a Principal Component Analysis (PCA) of HS data (i.e., López-Moreno et al., 2020b; Matiu et al., 2020) and for each elevation depending on PC1 and PC2 scores. PCA scores are shown at Figure S2, whereas Tthe number of season types per sector are shown at Figure S3 and the spatial regionalization is presented at Figure 1.

### 4. Results

We validated the FSM2 at Section 4.1. Subsequently, we analyzed the snow sensitivity to temperature and precipitation change based on five snow climate indicators, namely the seasonal HS, peak HS max, peak HS date, snow duration and snow ablation. Compound season types show similar relative importance on the snow sensitivity to temperature and precipitation change regardless of the Pyrenean sector. For this reason, our results have been focused on seasonal snow changes due to increments of temperature, elevation, and compound season type. These are the key factors that ruled the snow sensitivity to temperature and precipitation change, and an accurate analysis is provided at Section 4.2. Spatial differences on the snow sensitivity to temperature and precipitation change during compound season types are examined at Section 4.3.

### 4.1 Snow model validation

Our snow model validation analysis (Figures 2 and 3) confirmed that FSM2 accurately reproduces the observed HS values. On average, the FSM2 had a $R^2$ greater than 0.83 for all stations. In general, the snow model slightly overestimated the maximum HS values. The highest $R^2$ values were at A4 and A2 ($R^2 = 0.85$ in both stations), and the lowest were at A3 and A1 ($R^2 = 0.79$ and $R^2 = 0.82$, respectively). The highest accuracy was at A4 (RMSE = 18.5 cm, MAE = 8.9 cm), and the largest errors were at A2 (RMSE = 45.8 cm, MAE = 29.0 cm).

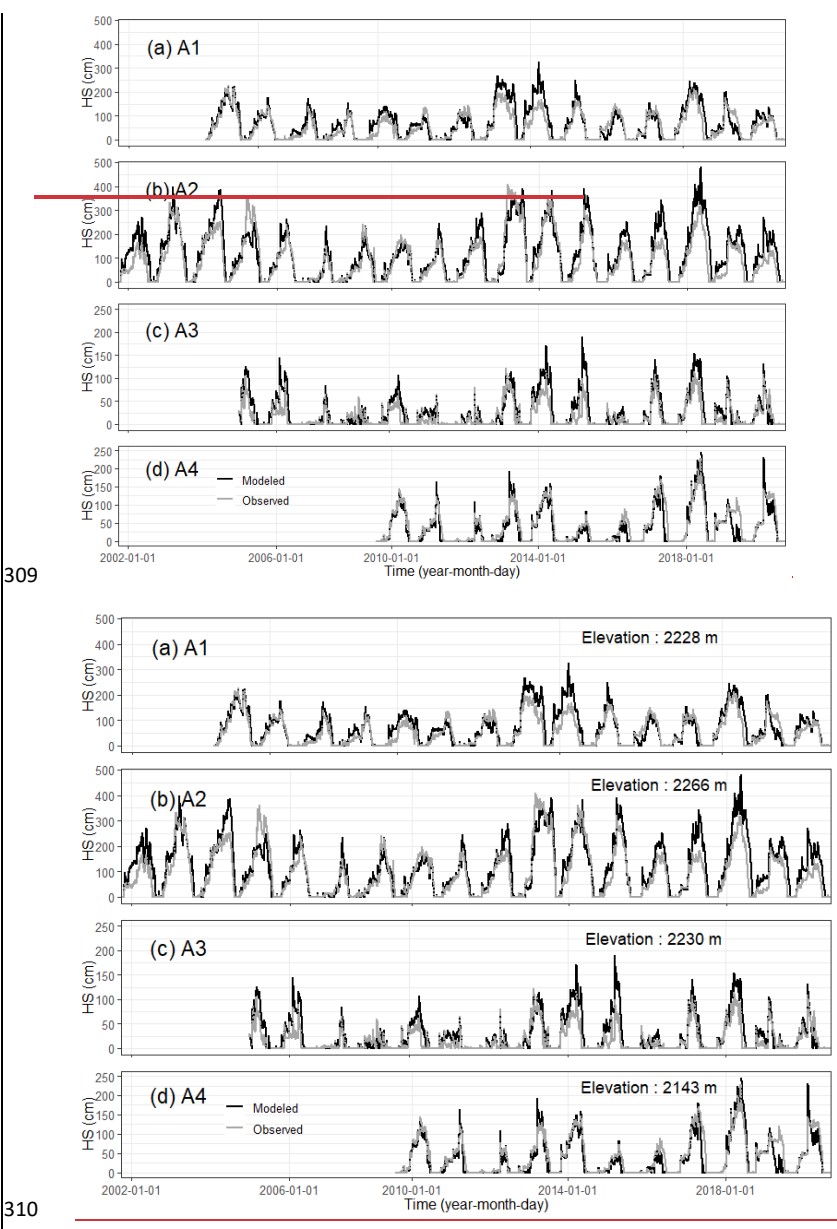

**Figure 2.** Time series of the observed (gray) and modelled (black) HS values at the four AWSs.


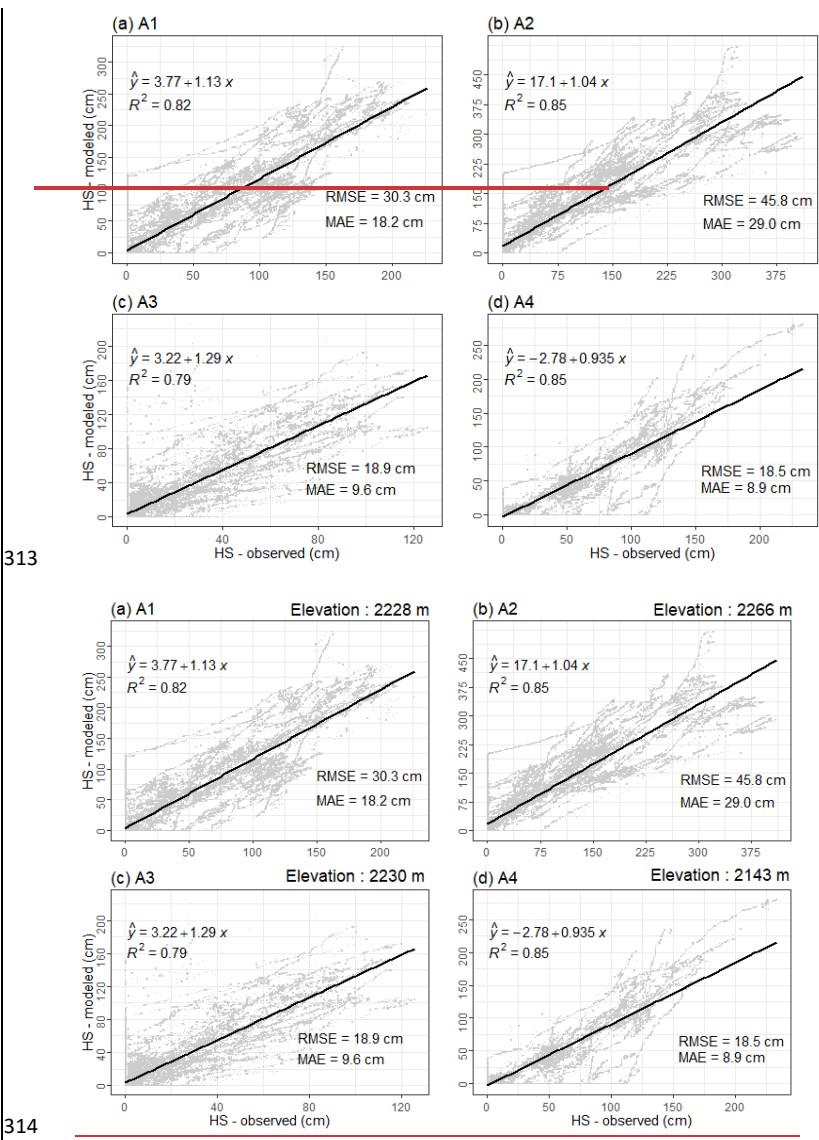


**Figure 3.** Regression analysis of observed (x-axis) and simulated (y-axis) HS values.


**4.2 Snow sensitivity to temperature and precipitation change**


We then determined seasonal HS profiles for each perturbed climate scenario and compound season type (Figure 4). The results show a non-linear response between seasonal HS loss and

temperature increase. When the temperature increased at 1℃ intervals, the largest relative
seasonal HS decrease from the baseline was at + 1℃ for all elevations and all compound season
type. High elevation areas had lower seasonal HS variability between compound season types
than low elevations (Figure S4). At low elevations, snow was- greater during CW seasons than
other seasons. All the snowpack-perturbed scenarios indicated that snowpack decreased for all
elevations under warming climate scenarios. Snowpack sensitivity to temperature and
precipitation change depended on the compound season type (Figures 5 and 6). At low elevations,
the seasonal changes in HS ranged from −37% (WW) to −28% (CD) per ℃ increase. For mid-
elevation ranges, there were no remarkable differences among compound season types (Table 2),
and the seasonal HS changes ranged from −34% (WW) to −30% (CW) per ℃ increase. Low and
mid-elevations had greater snowpack reductions than high elevations. In the latter, a 10% increase
of precipitation counterbalanced a temperature increase of about 1℃, and there were no
remarkable differences in the seasonal HS from the baseline scenario especially in the coldest
months of the season (Figure S5 and Figure S6). The maximum seasonal HS sensitivity to
temperature and precipitation was during WD seasons (27%/℃), and the minimum was during
CW seasons (−22%/℃).

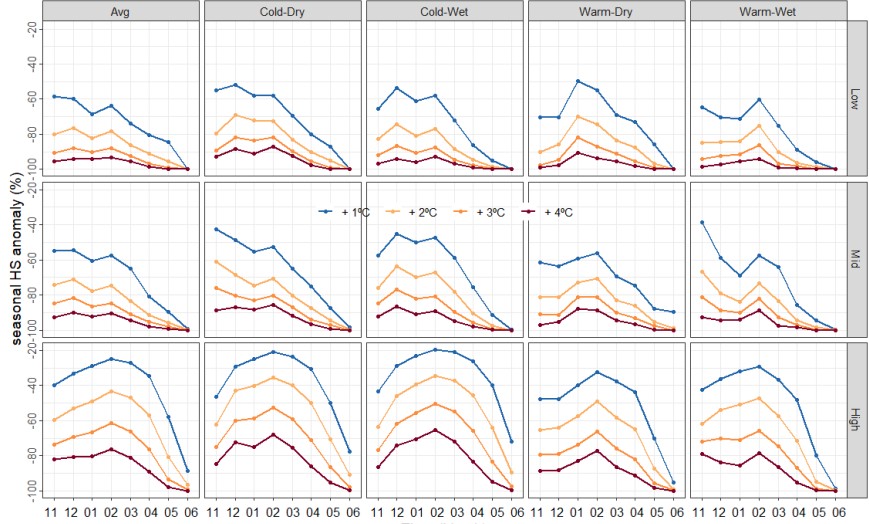

**Figure 4.** Anomalies of seasonal HS for low, mid and high elevation (rows), compound season type
(columns), and different temperature increases (colors).

**Table 2.** Average and seasonal HS and peak HS sensitivity to temperature and precipitation
change during the four different compound temperature and precipitation seasons at three
different elevations.

| Season type | %HS/ ºC | | | %peak HS max/ºC | | |
|---|---|---|---|---|---|---|
| | Low | Mid | High | Low | Mid | High |
| Avg. | −33 | −33 | −25 | −20 | −20 | −16 |
| CD | −28 | −30 | −22 | −17 | −17 | −14 |
| CW | −33 | −32 | −22 | −22 | −20 | −15 |
| WD | −32 | −30 | −27 | −19 | −16 | −16 |
| WW | −37 | −34 | −26 | −24 | −24 | −16 |


At low and mid elevations, the peak HS max was greatest during WW seasons (−24%/ºC) and
lowest during the CD and WD seasons (−17%/ºC for both). At high elevations, there were no
clear differences in the peak HS max for the different seasons. The maximum peak HS max was
during WD seasons (−16%/ºC) and the minimum was during CD seasons (−14%/ºC).

We also determined average seasonal snow duration for each elevation range and compound
season type for different temperature increases (Table 3 and Figure 5c). The minimum snow
duration was during CW seasons (−13%/ºC at low elevations, −10%/ºC at mid-elevations, −5%/ºC
at high elevations). At low elevations, the snow duration was most sensitive during WW seasons
(−17%/ºC). On the contrary, at mid-elevations and high elevations, the snow duration was most
sensitive during WD seasons (−13%/ºC at mid-elevations and −8%/ºC at high elevations).

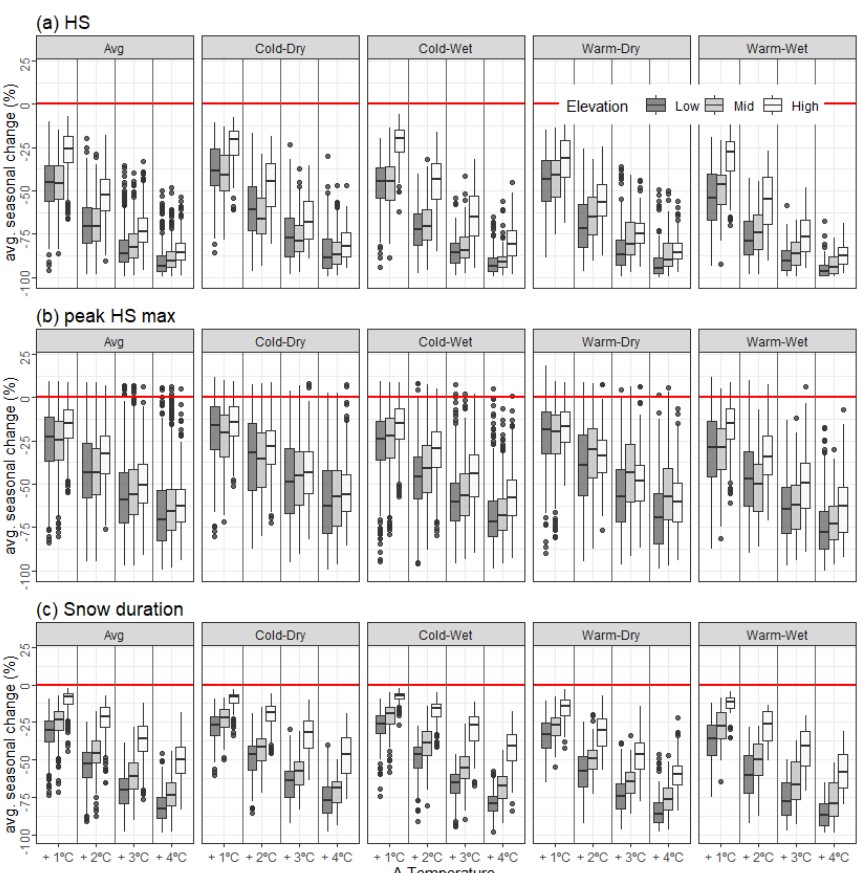

**Figure 5.** Anomalies of seasonal HS (a), peak HS max (b) and snow duration (c) for different temperature increases relative to baseline at three different elevations during the four different compound season types. The solid black lines within each boxplot are the average. Lower and upper hinges correspond to the 25th and 75th percentiles, respectively. The whisker is a horizontal line at 1.5 interquartile range of the upper quartile and lower quartile, respectively. Dots represent the outliers. Data is grouped by season, compound season type, increment of temperature, precipitation variation, elevation, and massif.

The peak HS date occurred earlier due to warming, independently of precipitation changes. During WD seasons, the peak HS date per °C was anticipated by 3 days at low elevations, 3 days at mid-elevations, and 6 days at high elevations; during CD seasons, the peak HS date per °C was anticipated by 4 days at low elevations, 5 days at mid-elevations, and 9 days at high elevations. In low and mid elevation areas, if the temperature increase was no more than about 1°C above

baseline, there was little change in the peak HS date (Figure 6). In addition, the minimum peak HS date change is found during WW seasons (Table 3), because the snowpack would be scarce at those times, and there were no defined peaks (Figure S4).

We determined the snow ablation sensitivity to temperature and precipitation change in response to different temperature increases at different elevations and during different compound season types. The results show there were low differences in absolute snow ablation values in a warmer climate (Figure 7). At low elevations, the average snow ablation sensitivity to temperature and precipitation change in all four compound seasons was 12%/°C (Table 3). At mid-elevations and high elevations, the maximum snow ablation sensitivity to temperature and precipitation change was during dry seasons; WD seasons had a snow ablation sensitivity to temperature and precipitation change of 13%/°C at mid-elevations and 10%/°C at high elevations. On the other hand, the minimum values for mid-elevations were during WW seasons, when the snow ablation sensitivity to temperature and precipitation change was 8%/°C; the minimum values at high elevations were during CW seasons, when was 5%/°C.

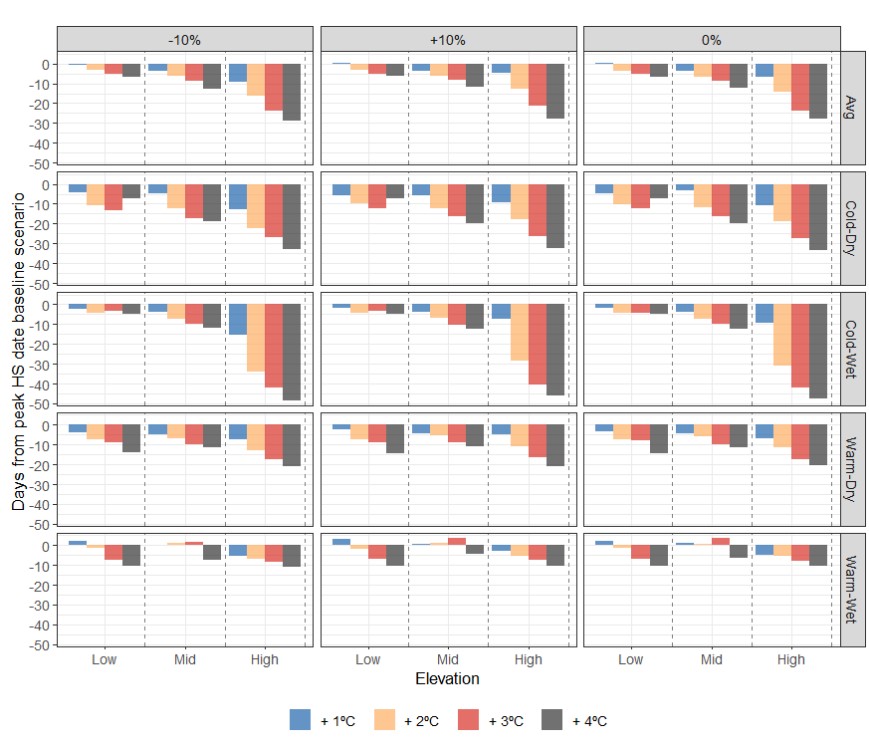


**Figure 6.** Difference (days) from baseline Peak HS date at three different elevations and during the four different temperature (colors) and precipitation shifts (columns) for each season (boxes).


**Table 3.** Snow duration, snow ablation, and peak HS date sensitivity to temperature and precipitation change during the four different compound season types.

| Season Type | Snow duration (%/ºC) | | | Snow ablation (%/ºC) | | | Peak HS date (days/ºC) | | |
|---|---|---|---|---|---|---|---|---|---|
| | Low | Mid | High | Low | Mid | High | Low | Mid | High |
| Avg. | −15 | −12 | −6 | 12 | 11 | 7 | -1 | -3 | -7 |
| CD | −13 | −11 | −5 | 12 | 13 | 8 | -4 | -5 | -9 |
| CW | −13 | −10 | −5 | 12 | 10 | 5 | -2 | -3 | -13 |
| WD | −16 | −13 | −8 | 12 | 13 | 10 | -3 | -3 | -6 |
| WW | −17 | −13 | −7 | 12 | 8 | 7 | -1 | 0 | -3 |



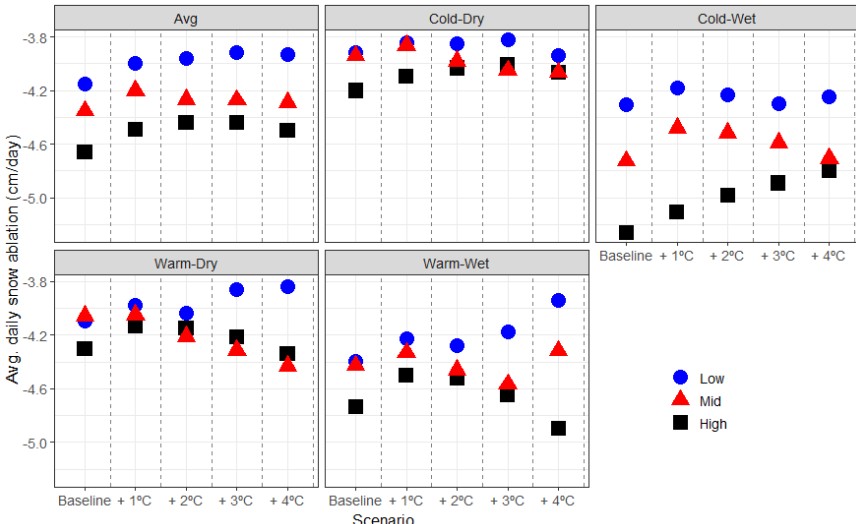

**Figure 7.** Absolute snow ablation values (cm/day) (y-axis) at three different elevations during four different compound temperature and precipitation for baseline and different increments of temperature (x-axis). seasons.

**4.3 Spatial patterns**

PCA analysis reveals four Pyrenean sectors, namely northern-western (NW), northern-eastern (NE), southern-western (SW), and southern-eastern (SE). No remarkable differences between sectors are found in the relative importance of each compound season type in the snow sensitivity to temperature and precipitation change (Figure 8). Snow sensitivity to temperature and precipitation change absolute values are generally lower at northern slopes (NW and NE) than at the southern slopes (SW and SE) (Figure S7 and Figure S8). In detail, seasonal HS ranged from −26%/°C during CD (NW) to −36%/°C during WW (SE). Similarly, the maximum peak HS max sensitivity to temperature and precipitation change was at SE during WW seasons (25%/ºC) and the minimum was during CD seasons at NW (15%/ºC). The snow duration sensitivity to temperature and precipitation change increased during WW seasons, and the maximum changes were at SE (−16%/ºC); in contraposition, the lowest sensitivity to temperature and precipitation change are found at NW, during CD and CW seasons (−8%/ºC, in both seasons). Snow ablation sensitivity to temperature and precipitation change increases towards the eastern Pyrenees, particularly during WD seasons (14%/ºC and 13%/ºC for NE and SE, respectively). Finally, no remarkable peak HS date differences are observed between sectors and maximum values are

found during CD and CW seasons, when the peak HS date is anticipated >= 5 per ℃ for all
sectors.

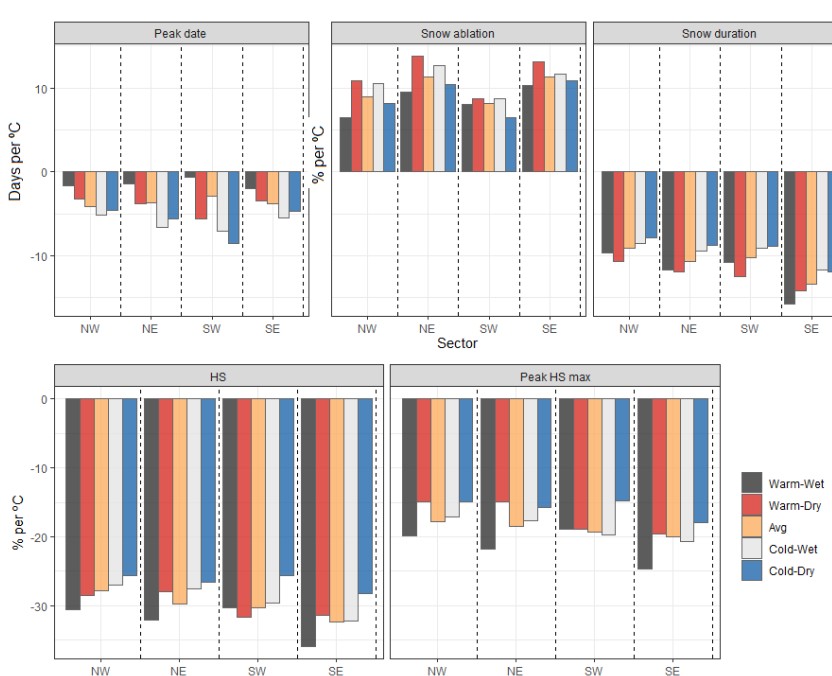

**Figure 8.** Average snow sensitivity to temperature and precipitation change (y-axis) grouped by
sector (x-axis), season type (color bars) and snow climate indicator (boxes).

**5. Discussion**

The spatial and temporal patterns of snow in the Pyrenees are highly variable, and climate
projections indicate that extreme events will likely increase during future decades (Meng et al.,
2022). Therefore, we analyzed factors that affect the snowpack sensitivity to temperature and
precipitation change gain insight into how future climate changes may affect the snow regime.

**5.1 Snow sensitivity to temperature and precipitation change and relationship with**
**historical and future snow trends**

**5.1.1 Snow accumulation phase**

The snow losses due warming that we described here are mainly associated with increases in the rain/snowfall ratio (Figure S9), changes in the snow onset and offset dates (Figure S4), and increases in the energy available for snow ablation during the later months of the snow season, as it was previously reported by literature (e.g., Pomeroy et al., 2015; Lynn et al., 2020; Jennings and Molotch, 2020). At high elevations, a trend of increasing precipitation (+10%) could counterbalance temperature increases (<1ºC; Figure S5), consistent with the results previously reported for specific sites of the central Pyrenees (Izas, 2000m; López-Moreno et al., 2008). Rasouli et al. (2014) also found that a 20% increase of precipitation could compensate for 2ºC increase of temperature in subarctic Canada. A climate sensitivity analysis in the western Cascades (western USA) found that increases of precipitation due to warming modulated the snowpack accumulation losses by about 5%/1°C (Minder, 2010). These results are consistent with recent data that examined snow above 1000 m in the Pyrenees, which found that an increase in the frequency of west circulation weather types since the 1980s increased the HS (Serrano-Notivoli et al., 2018; López-Moreno et al., 2020), snow accumulation (Bonsoms et al., 2021a), and changes in winter snow days (Buisan et al., 2016). There are similar trends in the Alps, with an increase of extreme (exceeding the 100-year return level) snowfall above 3000 m during recent decades (Roux et al., 2021) and increases in extreme winter precipitation (Rajczak and Schär, 2017).

**5.1.2 Snow ablation phase**

Climate warming leads to a cascade of physical changes in the SEB that increase snow ablation near the 0ºC isotherm. On overall, the snow ablation showed low to inexistent changes due to warming. Comparison between low and high elevations indicated slightly faster snow ablation at high elevations (Figure 7). This higher rate of snow ablation per season at high elevations (which have deeper snowpacks) are probably because the snow there lasts until late spring, when more energy is available for snow ablation (Bonsoms et al., 2022). Temperature increase does not imply remarkable changes in snow ablation per season because warming decreases the magnitude of the snowpack (seasonal HS and peak HS max) and triggers an earlier onset of snowmelt (Wu et al., 2018). The earlier peak HS date (Table 3 and Figure 6) implies lower rates of net shortwave radiation, because snow melting starts earlier in warmer climates (Pomeroy et al., 2015), coinciding with the shorter days and lower solar zenith angle (Lundquist et al., 2013; Sanmiguel-Vallelado et al., 2022). Our results agree with the slow snow melt rates reported in the Northern Hemisphere from 1980 to 2017 (Wu et al., 2018). The results of previous studies were similar for subarctic Canada (Rasouli et al., 2014) and western USA snowpacks (Musselman et al., 2017b), but Arctic sites had faster melt rates (Krogh and Pomeroy, 2019).

474

**5.1.3 Snow sensitivity to temperature and precipitation change and snowpack projections**

476

Our results suggest that warming had a non-linear effect on snowpack reduction. Our largest snow losses were for seasonal HS when the temperature increased by 1°C above baseline. At low and mid elevations, the average seasonal HS decrease was more than 40% for all compound season types, and the maximum sensitivity was during WW seasons. Previous research in the Pyrenees and other mid-latitude mountain ranges reported similar results. A study in the central Pyrenees reported the peak SWE was 29%/°C, whereas snow season duration decreased by about 20 to 30 days at about 2000 m (López-Moreno et al., 2013). The average peak HS max at high elevations in the Pyrenees (−16 %/°C; Figure 6 and Table 2), was similar to the average peak SWE sensitivity (−15%/°C) reported in the Iberian Peninsula mountains at 2500 m (Alonso-González et al., 2020a). These results are also consistent with climate projections for this mountain range. In particular, for a 2°C or more increase of temperature, the snow season declined by 38% at the lowest ski resorts (~1500 m) in the SE Pyrenees (Pons et al., 2015). However, high emission climate scenarios projected an increase in the frequency and intensity of high snowfall at high elevations (López-Moreno et al., 2011b). Snow sensitivity in the easternmost areas could decline during the winter because of a trend for an increase of about 10% in precipitation in this area (Amblar-Francés et al., 2020). Our projected changes in the Pyrenean snowpack dynamics are similar to the expected snow losses in other mountain ranges. For example, a study of the Atlas Mountains of northern Africa concluded that snowpack decreases were greater in the lowlands and projected seasonal SWE declines of 60% under the RCP4.5 scenario and 80% under the RCP8.5 scenario for the entire range (Tuel et al., 2022). A study in the Washington Cascades (western USA) found that snowpack decline was 19 to 23% per 1°C (Minder, 2010), similar to the values in the present study at high elevations. A study of the French Alps (Chartreuse, 1500 m) found that seasonal HS decreases on the order of 25% for a 1.5°C increase and 32% for a 2°C increase of global temperature above the pre-industrial years (Verfaille et al., 2018). A study of the Swiss Alps reported a snowpack decrease of about 15%/°C (Beniston, 2003); in the same alpine country, another study predicted the seasonal HS will decrease by more than 70% in massifs below 1000 m in all future climate projections (Marty et al., 2017). The largest snow reductions will likely occur during the periods between seasons (Steger et al., 2013; Marty et al., 2017). Nevertheless, at high elevations, snow climate projections found no significant trend for maximum HS until the end of the 21st century above 2500 m in the eastern Alps (Willibald et al., 2021), suggesting that internal climate variability is a major source of uncertainty of SWE projections at high elevations (Schirmer et al., 2021).

509

**5.2 Influence of compound temperature and precipitation seasons**

511

We found that the maximum sensitivities of seasonal HS and peak HS max to temperature and precipitation change were during WW seasons at low and mid-elevations and during WD seasons at high elevations. Brown and Mote (2008) analyzed the sensitivity of snow to climate changes in the Northern Hemisphere and found maximal SWE sensitivities in mid-latitudinal maritime winter climate areas, and minimal SWE sensitivities to temperature and precipitation change in dry and continental zones, consistent with our results. López-Moreno et al. (2017) also found greater decreases of SWE in wet and temperate Mediterranean ranges than in drier regions. Furthermore, Rasouli et al. (2022) studied the northern North American Cordillera and found higher snowpack sensitivities to temperature and precipitation change in wet basins than dry basins. Our maximum snow ablation relative change over the baseline scenario occurred during WD seasons, in accordance with Musselman et al. (2017b), who found a higher snowmelt rate during dry years in the western USA. Low and mid-elevations are highly sensitive to WW seasons because wet conditions favor decreases in the seasonal HS due to advection from sensible heat fluxes. The temperature in the Pyrenees is still cold enough to allow snowfall at high elevations during WW seasons, and for this reason we found maximal sensitivities to temperature and precipitation change during WD seasons. Reductions of snowfall in alpine regions can be compensated in a warmer scenario, because warm and wet snow is less susceptible to blowing wind transport and losses from sublimation (Pomeroy and Li, 2000; Pomeroy et al., 2015). During spring, snow runoff could be also greater in wet climates due to rain-on-snow events (Corripio et al., 2017), coinciding with the availability of more energy for snow ablation.

**5.3 Spatial and elevation factors controlling snow sensitivity to temperature and precipitation change**

Comparison between Pyrenean sectors (Figure 8) reveals no remarkable differences in the relative importance of each compound season type in the snow sensitivity to temperature and precipitation change. This is because by applying a joint-quantile approach for each massif and elevation, we are comparing similar climate seasons between sectors, regardless of the number of compound season types recorded in each massif during the baseline period (Figure S1 and S3). The highest absolute snow sensitivity to temperature and precipitation change values is found in the SE Pyrenees. This is consistent with the snow accumulation and ablation patterns previously reported in this region (Lopez-Moreno, 2005; Navarro-Serrano et al., 2018; Alonso-González et al., 2020a; Bonsoms et al., 2021a; Bonsoms et al., 2021b; Bonsoms et al., 2022). The Atlantic climate has less of an influence in the SE sector, and in situ observations indicated there was about half of the

seasonal snow accumulation amounts as in northern and western areas at the same elevation (>2000 m; Bonsoms et al., 2021a). The snow in the SE Pyrenees is more sensitive to temperature and precipitation change because these massifs are exposed to higher turbulence and radiative heat fluxes (Bonsoms et al., 2022), and 0ºC isotherm is closer. Similar conclusions are found for low elevations, where the results show an upward displacement of the snow line due to warming. Previous studies described the sensitivity of the snow pattern to elevation at specific stations of the central Pyrenees (López-Moreno et al., 2013; 2017), Iberian Peninsula mountains (Alonso-González et al., 2020a), and other ranges such as the Cascades (Jefferson, 2011; Sproles et al., 2013), the Alps (Marty et al., 2017), and western USA (Pierce et al., 2013; Musselman et al., 2017b). In these regions, the models suggest larger snowpack reductions due to warming at subalpine sites than at alpine sites (Jennings and Molotch, 2020) due to closer isothermal conditions (Brown and Mote, 2009; Lopez-Moreno et al., 2017; Mote et al., 2018).

**5.4 Environmental and socioeconomic implications**

Our results indicated there will be an increase of snow ablation days and imply a disappearance of the typical sequence of snow accumulation and snow ablation seasons. Climate warming triggers the simultaneous occurrence of several periods of snowfall and melting, snow droughts during the winter, and ephemeral snowpacks between seasons. These expected decreases in snow will likely have important impacts on the ecosystem. During spring, a snow cover cools the soil (Luetschg et al., 2008), delays the initiation of freezing (Oliva et al., 2014), functions as a thick active layer (Hrbáček et al., 2016), and protects alpine rocks from exposure to solar radiation and high air temperatures (Magnin et al., 2017). Due to warming temperatures, the remaining glaciers in the Pyrenees are shrinking and are expected to disappear before the 2050s (Vidaller et al., 2021). The shallower snowpack that we identified in this work will increase the vulnerability of glaciers, because snow has a higher albedo than dark ice and debris-covered glaciers and functions as a protective layer for glaciers (Fujita and Sakai, 2014).

The earlier onset of snowmelt suggested by our results, which is greater at low and mid-elevations during WD seasons, is in line with previous global studies that reported earlier streamflow due to earlier runoff dates (Adam et al., 2009; Stewart, 2009), and with a study of changes in the Iberian Peninsula River flows (Morán-Tejeda et al., 2014). Overall, our results are consistent with the slight decrease of the river peak flows that have occurred in the southern slopes of the Pyrenees since the 1980s (Sanmiguel-Vallelado et al., 2017). The reductions of seasonal HS that we identified , suggest that snowmelt-dominated stream flows will likely shift to rainfall dominated regimes. Although high elevation meltwater might increase and contribute to earlier groundwater

recharging (Evans et al., 2018), the increased evapotranspiration in the lowlands (Bonsoms et al.,
2022) could counter this effect, so there is no net change in downstream areas (Stahl et al., 2010).
Snow ephemerality triggers lower spring and summer flows (Barnett et al., 2005; Adam et al.,
2009; Stahl et al., 2010) and has an impact on the hydrological management strategies. Winter
snow accumulation affects hydrological availability during the months when water and
hydroelectric demands are higher. This is because reservoirs store water during periods of peak
flows (winter and spring), and release water during the driest season in the lowlands (summer)
(Morán-Tejeda et al., 2014). Recurrent snow-scarce seasons may intensify these hydrological
impacts and lead to competition for water resources among different ecological and
socioeconomic systems. The economic viability of mountain ski-resorts in the Pyrenees depends
on a regular deep snow cover (Gilaberte-Burdalo et al., 2014; Pons et al., 2015), but this is highly
variable, especially at low and mid-elevations. The expected increase in snow-scarce seasons that
we identified here is consistent with climate projections for this region, which suggest that no
Pyrenean ski resorts will be viable under RCP 8.5 scenario by the end of the 21[st] century (Spandre
et al., 2019).

## 5.5 Limitations and uncertainties

The meteorological input data that we used to model snow were estimated for flat slopes and the
regionalization system we used was based on the SAFRAN system. According to this system, a
mountain range is divided into massifs with homogeneous topography. The SAFRAN system has
negative biases in shortwave radiation, a temperature precision of about 1 K, and biases in the
accumulated monthly precipitation of about 20 kg/m$^2$ (Vernay et al., 2021). Our estimates of snow
sensitivity to temperature and precipitation change were based on the delta-approach, which
considers changes in temperature and precipitation based on climate projections for the Pyrenees
(Amblar-Francés et al., 2020), but assumes that the meteorological patterns of the reference period
will be constant over time. In this work we used a physical-based snow model since it provides
better results for future snow climate change estimations than degree-day models (Carletti et al.,
2022). The FSM2 is a physics-based model of intermediate complexity, and the estimates of snow
densification are simpler than those from more complex models of snowpack. However, a more
complex model does not necessarily provide better performance in terms of snowpack and runoff
estimation (Magnusson et al., 2015). The FSM2 configuration implemented in this work includes
snow meltwater retention, snowpack refreezing and snow albedo based on snow age, which are
the physical parameters included in the best-performing snow models according to Essery et al.
(2013). Snow model sensitivity studies reveal that intermediate complexity models exhibit similar

snow depth accuracies than most complex multi-layer snow models, as well as robust performances across seasons (Terzago et al., 2020).

**6 Conclusions**

Our study assessed the impact of temperature and precipitation change on the Pyrenean snowpack during compound cold-hot and wet-dry seasons, using a physical-based snow model that was forced by reanalysis data. We determined the snow sensitivity to temperature and precipitation change using five key indicators of snow accumulation and snow ablation. The lowest snow sensitivity to temperature and precipitation change was at high elevations of the NW Pyrenees and increased at lower elevations and in the SE slopes. An increase of 1ºC at low and mid elevation regions led to remarkable decreases in the seasonal HS and snow duration. However, at high elevations, precipitation plays a key role, and temperature is far from the isothermal 0ºC during the middle of winter. In this region, a 10% increase of precipitation, as suggested by ~~many climate projections~~ the Spanish Meteorological Agency (AEMET) over the eastern regions of this range, could compensate for temperature increases on the order of about < 1ºC. The impact of climate warming depends on the combination of temperature and precipitation during compound seasons. Our analysis of seasonal HS and peak HS max indicated the greatest declines were during WW seasons and the smallest declines were during CD seasons, independently of the Pyrenean sector. For snow duration, however, the highest (lowest) sensitivity to temperature and precipitation change is found during WD (CW) seasons. Similarly, snow ablation had slightly greater sensitivities to temperature and precipitation change during WD seasons, in that snow ablation variation is less than 10% and the peak HS date occurred about 5 days earlier per ºC. Our findings thus provide evidence that the Pyrenean snowpack is highly sensitive to climate warming and suggest that the snowpacks of other mid-latitude mountain ranges may also show similar response to warming.

**Data availability**

Snow model (FSM2) is open access and provided by Essery (2015) and available at https://github.com/RichardEssery/FSM2 (last access 16 December 2022). Climate forcing data is provided by Vernay et al. (2021), through AERIS (https://www.aeris-data.fr/landing-page/?uuid=865730e8-edeb-4c6b-ae58-80f95166509b#v2020.2; last access 16 December 2022). Data of this work is available upon request (contact: josepbonsoms5@ub.edu).

**Acknowledgements**
This work falls within the research topics examined by the research group "Antarctic, Artic,
Alpine Environments-ANTALP" (2017-SGR-1102) funded by the Government of Catalonia,
HIDROIBERNIEVE (CGL2017-82216-R) and MARGISNOW (PID2021-124220OB-100), from
the Spanish Ministry of Science, Innovation and Universities. JB is supported by a pre-doctoral
University Professor FPI grant (PRE2021-097046) funded by the Spanish Ministry of Science,
Innovation and Universities. The authors are grateful to Marc Oliva, who reviewed an early
version of this manuscript. We acknowledge the SAFRAN data provided by Météo-France –
CNRS and the CNRM Centre d'Etudes de la Neige, through AERIS.
**Author contributions**
JB analyzed the data and wrote the original draft. JB, JILM and EAG contributed to the
manuscript design and draft editing. JB, JILM and EAG read and approved the final manuscript.

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
