# Peer review of "Snow sensitivity to temperature and precipitation change during compound cold-hot and wet-dry seasons in the Pyrenees."

_EGUsphere, 2022_

## Referee Comment (RC2)

**Review of "Snow sensitivity to climate change during compound cold-hot and wet-dry seasons in the Pyrenees"**

Dear authors, dear editor,

The paper submitted discusses the impact of climate change on snow cover in the Pyrenean for different air temperature and precipitation pathways, and for different seasonal conditions. In general, the paper is clear and shows clean figures. As I detail below, there are some important points to be addressed, mainly enhancing clarity of the description of the method and analysis (to allow reproducibility), and focusing more the analysis on the main question.

I have no doubt that these points can be clarified and/or enhanced by the authors and that a reviewed version will fit for a publication in TC. Indeed, if the authors are able to re-focus the analysis on the main point of the paper (i.e. the difference between compound cold-hot and wet-dry seasons), this work will brin some valuable contribution for the community.

Do not hesitate to contact me for further discussions.

Best regards,

Adrien Michel (adrien.michel@epfl.ch)

**Major comments:**

**Use of "climate sensitivity" term**

Throughout the introduction (and the rest of the paper), the term climate sensitivity is used several times, mostly in the form "climate sensitivity of snow". Climate sensitivity is defined as: "*Climate sensitivity refers to the change in the annual global mean surface temperature in response to a change in the atmospheric $CO_2$ concentration or other radiative forcing.*" [IPCC glossary[1]]. In your case it is rather used to describe the response of snowpack to climate change. E.g. lines 123-124: "*[…] suggest the existence of a wide variety of climate sensitivities of snow depending on elevation and spatial factors.*", where you mean "a wide range of responses to climate change". I'd recommend to reformulate all the instances of "climate sensitivity" throughout the manuscript since in the climate change language this corresponds to something really specific. You should use "climate change impact", which is in my opinion the correct word, or at least stick to "sensitivity of snow to climate change".

**Validation process**

The whole validation process is not clearly described. In P6 you say "*In this work, the FSM2 model configuration was selected on a trial-and-error basis (not shown here), validated by in-situ snow records of four automatic weather stations (AWS) placed at high elevation areas of the Pyrenees. Then, the FSM2 was forced with the SAFRAN reanalysis dataset for the entire mountain range (see Section 3.2).*" and finally you describe some corrections of the data from AWS

1.  Did you run at stations with SAFRAN data of with AWS data for the validation?

2.  If run with AWS, when then do you validate the model with SAFRAN data? This is a crucial step.
* * *
[1] https://www.ipcc.ch/site/assets/uploads/2018/11/sr15_glossary.pdf

3. Did you use the AWS for the mentioned trial-and-error setup? In this case, this is a calibration, not a validation. It should then be validated at stations not used to calibrate the parameters

I think Section 3.1 should only describe the model (and here you should add few lines giving some details about the main model physical principle, assumptions, and parameters), and then a new Section 3.2 should describe accurately the calibration/validation procedure. The final model parameters need also to be available in order to allow the reproducibility of the study.

**Analysis description**

In line with the lack of details mentioned above, the actual simulations performed is not really well described. In Section 3.3 you say: *"Temperature and precipitation are perturbed for each massif and elevation range based the historical period"*, but never clearly say: *"The model is run for XXX regions, YYY years, etc."*. Moreover, for all the first part of the analysis, the spatial patterns are not discussed, and the difference in massifs only appears in the discussion). As a reader I was confused until reaching the bottom of page 15 to know whether the model was really run for different locations, or only for different elevation bands.

The procedure should be really explained (see my minor comment about a missing global study description at the end of the intro, which can help). Naming the massifs in Figure 1 and having a table briefly describing each massif (e.g. with min/mean/max elevation) would be useful for the analysis and help to clarify that the model is indeed run per massif. Another unclear point for me is the elevation used. Did you run only three elevations of some groups of elevation based on the 300m discretization of SAFRAN? In Section 3.4, you do not explain if the quantiles analysis is done per massif (i.e. looking at each season in comparison to other seasons in this given massif), or globally (i.e. looking at each season in comparison to the whole set of seasons for all massifs).

In Section 4.2, you should clearly state that all massifs are grouped together for the present analysis (and that the spatial analysis is performed later on). As far as I understand, Figure 4 shows the average across all massif. This should be clearly stated. Also, in the whole Section 4.2 changes in precipitation are not mentioned (except in the caption of Figure 5), and only shown in Supplementary Figures. However, the fact that precipitation (+10%) could contract a 1°C is presented as one of your key results. The corresponding Figures should thus be properly shown, introduced, and described in the main text and in the Results Section (now Supplementary figures are just mentioned in the Discussion section).

**Impact study, determining factors, uncertainty**

I have the feeling that Section 4.2 is a long list of numbers a bit hard to follow, and in many cases the text repeats numbers shown in the Figures and Tables. Moreover, I feel a significant part of the numbers mentioned in Section 4.2 are nor really useful to support the latter analysis.

In addition, this study inspects many aspects: different temperature and precipitation pathways, different kinds of compound seasons, and many sub-regions. In addition, they are analysed using 5 indicators, resulting on hundreds of different "numbers" to discuss. In the discussion, it is hard to really see the direction. Indeed, while the title suggests a focus on compound seasons, this is not really present in some part of the discussion (i.e., 5.4, which summarizes well known impacts and is in my opinion not necessary here, or 5.2.1, winch basically say that if we have more solid precipitation, we have higher snowpack). I would

encourage to maybe reduce and reorganize the discussion and to only focus on few points (e.g. compound season and spatial distribution). A large amount of data has been produced for this study and it can be tempting to discuss every aspects of the data obtained from the model, but this makes it harder to read, and hide what is really the novelty of this work. Note that the plots about spatial distribution are introduced in the discussion, while in my opinion they belong to the results Section.

I've one concern about the method itself. As far as I understand, seasons "classes" (WW, CW, etc.) are determined for each subregion and elevation range separately (Figure S1). And thus, figures like 4 are obtained by averaging all the regions together for each elevation band and season class. My problem is that from Figure S1 we see that some classes of season are manly dominated by some regions (e.g. cold wet is dominated by south-west regions). So, when comparing the different season class, we do not really know if the difference is due to the meteorological input, of due to some other aspects differing between regions. In addition, the season class is (maybe?) determined for each region separately (see my comment above), so a CW in one region might not be CW in another region. As a consequence, because of the approach chosen, I do not think the differences observed between compound seasons is only due to the specific weather of the seasons. This is probably the dominant factor, but the spatial difference would add some uncertainty there. This should at least be discussed. Note that there is no discussion about uncertainty and limitation, this should be added.

**Minor comments:**

**Abstract:** Please do not use abbreviations in abstract, only full words.

**P2 L38-39, L44, …:** Please sort citation in ascending order by year (throughout the whole manuscript).

**P3 L67-68:** I do not understand what "*coincides*" with "*low solar radiation periods*".

**P4 L95-96:** What do you mean by "mid-end 21$^{st}$ century"?

**P4 L107**: "." Missing

**P5 L112-113**: "*To date, some studies pointed out different climate sensitivies on wet or dry years*". Can you please explain in one sentence the different results found.

**P4 L126-128:** Here I would briefly describe the main steps used to achieve this objective

**P5 L139**: Which "lapse-rate"? Elevation lapse rate of precipitation?

**P5 L142**: "being ~ 1000" change to "being on average …". Please clarify in the rest of the paragraph where "~" means "around" and where it means "on average".

**P6 L177-178**: You should provide the final retained configuration for reproducibility.

**P7 Table 1**: Seems coordinates are in lat/lon °, not in UTM. Units are missing for the two "distance" column. "Reference period" is never explained in the text (see also major comments on the calibration/validation description).

**P8 L208**: Please provide a reference for the implementation

**P8 L217:** "in" section

**P9 L266:** What do you mean by "by massif"?

**P9 Section 4.1:** R2 should be $R^2$

**P11 L292:** Refer to Figure 5 at the end of the sentence. What does "Here" refer to?

**P11 Figure 4**: For comparison, you should also show the reference simulation (+0°C) in the Figure. There are some strange drops in snow height (see below).

[Figure]

**P13 Figure5**: how are the boxes constructed? Different seasons (i.e different years) + change in precipitation + different massifs? Or do you have only one point averaging across all seasons for a given massif? You should explain how are the boxes (1 and 3 quartiles?), whiskers, and outliers are defined.

**P14 Figure 6**: Why not using a boxplot here as in Figure 5?

**P15 figure 7**: How is this exactly computed? By "season" you mean the exact length of the ablation season (i.e. time between $HS_{max}$ and HS=0)?

**P16 L383-386**: This kind of statement should be in the Introduction section

**P16 L393-398**: I do not really see the added value of this information here

**P17 Figure 8, P20 Figure 9**: Units missing

**P17 L418-420**: You should show plots supporting it (e.g. a plot of precipitation phase)

**P18 L438-440**: Something is missing in this sentence, e.g. "The higher average […]"

**P18 Section 5.2.2**: This is really interesting. In my work on hydrology, I found that on a warmer world discharge peak from snowmelt will occur earlier, but also be "flatter" (see Michel, 2022[2]). I never went deeper in the analysis of the cause of the flattening. Your analysis on slower melt rate seems really relevant to answer this question.

**P19 L464**: "if" → "in"

**P19 L464-466**: With all the uncertainty involved, I would say "is similar"

**P19 L467**: A reference is needed here.
* * *
[2] https://hess.copernicus.org/articles/26/1063/2022/hess-26-1063-2022.html

---

## Author Comment (AC1)

[1] Department of Geography, University of Barcelona, Barcelona, Spain.

[2] Instituto Pirenaico de Ecología (IPE-CSIC), Campus de Aula Dei, Zaragoza, Spain.

[3] Centre d'Etudes Spatiales de la Biosphère (CESBIO), Université de Toulouse, CNES/CNRS/IRD/UPS, Toulouse, France.

Response to Reviewer 1. Comment posted on 20 September 2022.

Reviewer comments are in bold and responses in blue.

**General Comments:**

**The submitted manuscript investigates the sensitivity of climatological snow indicators on compound temperature and precipitation changes. The analysis is based on the snow model FSM, which is forced by daily reanalysis data between 1980 and 2019 and assimilated in-situ data. The results focus on seasonal data and three elevation levels. The topic is definitely of interest for readers of TC. I liked reading the manuscript, which has a clear structure and illustrative figures. However, the language needs some proofreading by English native person. I suggest to accept the manuscript as soon as the following points, have been addressed:**

The authors want to express their sincere gratitude to the reviewer comments. All the recommendations suggested by the reviewer were carefully taken into consideration and have improved the rigor and clarity to our findings presented in this paper.

**Chapter 3.1 is missing a common thread and therefor hard to understand. Please restructure the entire chapter. If I got it right then the data of the 4 AWS were used to correct the reanalysis data. But how? What do you mean with "by trial and error basis"?**

Sorry for the misunderstanding.

SAFRAN system data-assimilated in-situ (meteorological) records of the mountain range. We compared in-situ HS records (4 AWS) against FSM2 HS outputs (forced by meteorological AWS data) to validate the snow model. We have tried different snow model configurations (that is what we mean by "trial and error basis"), but we did not find significant differences in the performance and accuracy metrics. Therefore, we applied the most complex configuration, except for snow cover fraction estimation - we found good results with a linear function of HS-, and we forced the snow model using re-analysis data assimilated SAFRAN data.

We have rearranged the entire chapter 3.1, and we have added a new chapter "3.2 Snow model validation".

We also added the FSM2 configuration:

"We have evaluated different FSM2 model configurations (not shown) without significant differences in the accuracy and performance metrics. Therefore, we selected the most complex FSM2 configuration, except for snow cover fraction that was based on a linear function of HS. In detail, albedo is calculated based on a prognostic function, with increases due to snowfall and decreases due to snow age. Atmospheric stability is calculated as function of the Richardson number. Snow density is calculated as a function of viscous compaction by overburden and thermal metamorphism. Snow hydrology is estimated by gravitational drainage, including internal snowpack processes, runoff, refreeze rates, and thermal conductivity.

**The reanalysis data set of Vernay (2021) covers 1958-2020. Why do you analyze 1980 until 2019 only?**

, we have performed a snow sensitivity analysis (1980-2019 temporal period as baseline), according to climate change projections for the range (Amblar-Francés et al., 2020), which are based on the average 1980s onwards temperature and precipitation used as a reference period, As we have mentioned in the 3.5 section.

**According to Fig. 4 the main (average) snow cover even at high elevation last from November to Mai. This implies that extreme temperature or precipitation in October and June have no or only very marginal impact on the snow cover. However, you define the compound extremes based on October to June values. This makes not much sense!**

We are sorry for the misunderstanding. The season is defined based on previous studies, and the modeled snow for the baseline climate (1980 – 2019). Previous Figure 4 included only the climate perturbed seasonal snow evolution (which are not used for the season limits definition). We have changed Figure 4 and added the baseline climate seasonal snow. We must include the months between October and June for comparison between seasons and elevation.

**I don't understand the explanation why no change in the peak HS date can be detected (L242), which is also in contradiction to your statement (L582) in conclusions?**

The reference was for WW seasons. Peak HS date occurred earlier for most of the season types due to warming (Figure 7). However, for WW seasons, there are not relevant differences because maximum HS peak is significantly reduced, and the snow profile is flat (Figure 4).

We modified our statements and added Figure 7 to the main text.

We have changed: "*Climate warming decreases the peak HS date (Figure S4). The maximum peak HS date climate sensitivity is found during dry seasons. During WD (CD) seasons, the peak HS date will take place 9 (15), 3 (8) and 17 (1) days earlier on the season per ºC for low, mid and high elevations, respectively. The minimum peak HS date climate sensitivity is observed during WW seasons (Table 4). The peak HS date does not show any change due to warming, since the snowpack would be scarce during the season, and no defined maximum peaks would occur in any elevation range (Figure 4). In high elevation areas, if temperature increase does not exceed ~ 1ºC 345 respect the baseline scenario, the peak HS date is not expected to drastically change (Figure S4), except during dry seasons...*" to:

"Overall, the peak HS date occurred earlier due to warming (Figure 7), independently of precipitation shifts. During WD seasons, the peak HS date per °C was earlier by 9 days at low elevations, 3 days at mid-elevations, and 17 days at high elevations; during CD seasons, the peak HS date per °C was earlier by 15 days at low elevations, 8 days at mid-elevations, and 1 day at high elevations. In high elevation areas, if the temperature increase was no more than about 1ºC above baseline, there was little change in the peak HS date (Figure S4), except during dry seasons. The maximum peak HS date was during dry seasons. On the contrary, the peak HS date did not change significantly due to warming during WW seasons (Table 4), because the snowpack would be scarce at those times, and there were no defined peaks (Figure 4)."

**Minor points: L: 46: please rephrase**

Thank you. Done

**L 47: snow offset dates! You use also ablation dates and snowmelt dates. Please decide.**

Thanks. We have replaced "snow offset dates" and "snowmelt dates" for "snow ablation dates".

**L57: in regard to snow duration**

Thank you. Added.

**L82: spatially highly diverse**

Thank you. Modified

**L105: repetition of L57**

Thank you. We have moved 103-105 to L57 paragraph.

**L144: please rephrase**

Thank you. Changed. We have modified:

"However, no study has yet analyzed the climate sensitivity of snow during compound temperature and precipitation extreme seasons, caused by high-low temperatures (Warm-Cold seasons) or precipitation (Wet-Dry seasons)" to

"However, the sensitivity of snow during periods when there are seasonal extremes of temperature and precipitation has not yet been analyzed"

**L168: Snow model and validation data**

Done. We have changed the entire 3.1 order, according to comment 3.

**L190: wrong reference format**

Thank you. Changed.

**L191: What do you mean with were excluded? If there is no data, then there is nothing to evaluate!**

Thank you. We have delated our statement.

**L192: ultrasonic snow depth sensor**

Thanks. Changed.

**L193: Please provide a reference where to get the data**

Added:

"https://www.meteo.cat/wpweb/serveis/formularis/peticio-dinformes-i-dades-meteorologiques/peticio-de-dades-meteorologiques/; data requested: 14/01/2021)"

**L196: I'm not able to access the pdf given in the reference**

Thank you, now the reference is available (https://static-m.meteo.cat/wordpressweb/wp-content/uploads/2014/11/18120559/Les_Estacions_XEMA.pdf) .

**L198: units of the 5th and 6th column is missing.**

Added.

**L218: LWinc and temperature**

Added.

**L220: Meteorological data therein…**

Thank you. Changed.

**L251: two times "perdentiles"**

Thank you. Delated.

**L253: average compound temperature and precipitation seasons.**

Thank you. Changed.

**L260: What did you when the same peak HS was reached at several dates?**

Thank you for your suggestion. There is only one maximum peak HS for season.

**L262: This makes no sense. Please rephrase.**

We have changed "the average daily snow ablation per season (snow ablation)" for "daily average snow ablation per season (snow ablation)".

**L274: the best performance …**

Changed for "highest $R^2$ values".

**L278: the better performance?**

Changed for "highest accuracy".

**L279: observations are usually black...**

Thank you for your suggestion. We aim to maintain the snow model values in black since it can be more visible than in grey color.

**L288: non-linear (see also other occurrences)**

Thank you. Changed.

**L290: absolute or relative decreases**

Relative. Added:

**"**When progressively warmed at 1ºC intervals, the largest **relative** seasonal HS decreases from baseline climate are found at + 1ºC"

**L293: not surprising**

We have kept our statement since we consider that the information provided is required for the results interpretation.

**L306: please change temperature legend**

Thank you for your suggestion, we have modified Figure 4.

**L311: Average seasonal sensitivity of…**

Changed.

**L313: I'd suggest to replace the table with a bar plot**

Thank you. We replaced the table with a figure (a boxplot, in order to be consistent with Figure 3 and following reviewer 2 suggestion).

**L330: Please change the title of the y-axis to: average seasonal HS change (%)**

Thank you. Done.

**L331: Anomalies of…**

Done.

**L345: with respect to..**

Changed.

**L361: Sensitivity of..**

Changed.

**L368: Snow climate sensitivity (expressed as mean HS)**

Thank you for your suggestion. We have changed "snow climate sensitivity" for "HS climate sensitivity".

**L373: "lasts area" is no English!**

Changed.

**L377: Where can I see that "Snow duration sensitivity clearly increases during WW seasons"?**

We have added a reference to Figure 10 at L377, where it is observed that during WW seasons snow duration sensitivity increases at low elevation for the South-East.

**L408: Add percentage to the legend and rephrase figure caption.**

Changed.

**L419: "increases in the energy available for snow ablation". This in contradiction to what you wrote earlier, because the snow offset is moving to times with lower sun angles.**

We have changed the phrase for "…increases in the energy available for snow ablation **during the latest months of the seasons".**

**L432: the increase in winter precipitations was mainly based on low elevation data, which is usually rain and not snow.**

Thank you for your suggestion.

**L437: slightly faster**

Changed.

**L438. This higher average …**

Changed: "…This higher rate of snow ablation per season at high elevations (which have deeper snowpacks) are probably because the snow there lasts until late spring…".

**L443: Therefore, slower snow ablation rate… (where is this shown?)**

We have changed "slower snow ablation" for "lack of changes"

**L448: The earlier peak HS date a low and mid elevation …**

Thank you for your suggestion. We have changed "the earlier peak HS date" to "the earlier peak HS date at low and mid elevation".

**L449: starts earlier (i.e. in winter)**

Changed.

**L467: mountain range**

Changed.

**L473L in this area**

Changed.

**L486: no significant trend for maximum HS**

Done.

**L488: in high elevations**

Changed.

**L493: Sensitivities of maximum seasonal HS…**

Changed.

**L503: highly sensitive**

Changed.

**L506: High elevation snowfall**

Done.

**L513: Add percentage to the legend and rephrase figure caption.**

Done.

**L521: disappearance of the typical sequence…**

Done.

**L522: triggers the simultaneous occurrence of several periods of…**

Thank you for your suggestion. We have changed: "Climate warming triggers the simultaneously occurrence of snow accumulation and ablation episodes…" to "Our results indicated there will be an increase of snow ablation days and imply a disappearance of the typical sequence of snow accumulation seasons and snow ablation seasons."

**L524: on the ecosystem**

Done.

**L525: please rephrase**

Done.

**L533. The earlier snowmelt onset**

Thank you. Changed.

**L547: please rephrase**

We have changed: "The reservoirs operation strategies include hydrological resources storage during peak flows and water releases during summer; which coincides with the driest season in the lowlands, and when there are higher water and hydropower demands than in winter" to:

"Winter snow accumulation affects hydrological availability during the months when water and hydroelectric demands are higher. This is because reservoirs store water during periods of peak flows (winter and spring), and release water during the driest season in the lowlands (summer) (Morán-Tejeda et al., 2014)"

**L551 is dependent on a regular deep enough snow cover, which has been…**

Done.

**L553: The expected increase in snow scarce seasons pointed out in this work, is consistent with snow projections…**

Changed.

**L571: core month of the winter season**

Changed.

**L575: Repetition of L565**

We have delated L575.

**L581: show slightly larger sensitivities**

Done

**L582: increases about… and the peak HS date occurs about …**

Done

**L584: unclear, please rephrase**

Done. We have changed "This work provides evidence of the high climate sensitivity of the Pyrenean snowpack in comparison with global mountain ranges, suggesting the existence of similar climate sensitivities in other mid-latitude mountain areas" to

"Our findings thus provide evidence that the Pyrenean snowpack is highly sensitive to climate change, and suggest that the snowpacks of other mid-latitude mountain ranges may also show similar response to warming"

Thank you very much for your constructive comments.

---

## Author Comment (AC2)

[1] Department of Geography, University of Barcelona, Barcelona, Spain.

[2] Instituto Pirenaico de Ecología (IPE-CSIC), Campus de Aula Dei, Zaragoza, Spain.

[3] Centre d'Etudes Spatiales de la Biosphère (CESBIO), Université de Toulouse, CNES/CNRS/IRD/UPS, Toulouse, France.

Response to Reviewer 2. Comment posted on 06 October 2022.
Reviewer comments are in bold and responses in blue.

**Dear authors, dear editor,**

**The paper submitted discusses the impact of climate change on snow cover in the Pyrenean for different air temperature and precipitation pathways, and for different seasonal conditions. In general, the paper is clear and shows clean figures. As I detail below, there are some important points to be addressed, mainly enhancing clarity of the description of the method and analysis (to allow reproducibility), and focusing more the analysis on the main question.**

**I have no doubt that these points can be clarified and/or enhanced by the authors and that a reviewed version will fit for a publication in TC. Indeed, if the authors are able to re-focus the analysis on the main point of the paper (i.e. the difference between compound cold-hot and wet-dry seasons), this work will brin some valuable contribution for the community.**

We would like to express our sincere gratitude to Dr. Michel for their extensive constructive suggestions and comments. All the recommendations suggested by the reviewers were carefully taken into consideration and have improved the rigor and clarity to our findings presented in this paper.

**Major comments:**

**Use of "climate sensitivity" term**

**Throughout the introduction (and the rest of the paper), the term climate sensitivity is used several times, mostly in the form "climate sensitivity of snow". Climate sensitivity is defined as: "*Climate sensitivity refers to the change in the annual global mean surface temperature in response to a change in the atmospheric $CO_2$ concentration or other radiative forcing.*" [IPCC glossary[1]]. In your case it is rather used to describe the response of snowpack to climate change. E.g. lines 123-124: "*[…] suggest the existence of a wide variety of climate sensitivities of snow depending on elevation and spatial factors.*", where you mean "a wide range of responses to climate change". I'd recommend to reformulate all the instances of "climate sensitivity" throughout the manuscript since in the climate change language this corresponds to something really specific. You should use "climate change impact", which is in my opinion the correct word, or at least stick to "sensitivity of snow to climate change".**

Thank you for your suggestion. Accordingly to comments from reviewer 1, we have changed "climate sensitivity" for "sensitivity of snow to climate change" and "snow sensitivity" depending on the context.

**Validation process**

**The whole validation process is not clearly described. In P6 you say** *"In this work, the FSM2 model configuration was selected on a trial-and-error basis (not shown here), validated by in-situ snow records of four automatic weather stations (AWS) placed at high elevation areas of the Pyrenees. Then, the FSM2 was forced with the SAFRAN reanalysis dataset for the entire mountain range (see Section 3.2)."* **and finally you describe some corrections of the data from AWS.**

We have changed Section 3.2 and split the information into two sections:

Section 3.2, Snow model: where we describe the FSM2 configuration.

Section 3.3, Snow validation: where we provide a description of the snow model validation.

**Did you run at stations with SAFRAN data of with AWS data for the validation? If run with AWS, when then do you validate the model with SAFRAN data? This is a crucial step.**

We run the FSM2 with meteorological AWS data and compared the accuracy against HS records. It is not possible to compare the AWS between the AWS records and the SAFRAN system due to:

1. The different resolution and elevation bands. The SAFRAN system provides data by homogeneous (around 1000 km2) meteorological and topographical mountain massifs every 300 m, from 0 to 3600 m (Durand et al., 1999; Vernay et al., 2021), that do not coincide with the AWS elevation used for validating the FSM2.

2. The SAFRAN dataset that we used in this work was data-assimilated with in-situ meteorological observations of the mountain range. We cannot validate in-situ records that were previously data-assimilated by the SAFRAN system. In addition, the SAFRAN system has been extensively validated before our work.

**Did you use the AWS for the mentioned trial-and-error setup? In this case, this is a calibration, not a validation. It should then be validated at stations not used to calibrate the parameters**

We have validated the FSM2 against in-situ (AWS) snow simulations. We have evaluated different configurations, but no significant differences were observed in the accuracy and performance metrics.

We have added (also in response to reviewer 1)

"We have evaluated different FSM2 model configurations (not shown) without significant differences in the accuracy and performance metrics. Therefore, we selected the most complex FSM2 configuration, except for the snow cover fraction estimation, that is based on a linear function of HS. In detail, albedo is calculated based on a prognostic function, with increases due to snowfall and decreases due to snow age. Atmospheric stability is calculated as function of the Richardson number. Snow density is calculated as a

function of viscous compaction by overburden and thermal metamorphism. Snow hydrology is estimated by gravitational drainage, including internal snowpack processes, runoff, refreeze rates, and thermal conductivity"

**I think Section 3.1 should only describe the model (and here you should add few lines giving some details about the main model physical principle, assumptions, and parameters), and then a new Section 3.2 should describe accurately the calibration/validation procedure. The final model parameters need also to be available in order to allow the reproducibility of the study.**

We have added the model configuration. We also have added a chapter (5.5 Limitations and uncertainty) where we detailed the limitations of the input, model and method used.

**Analysis description**

**In line with the lack of details mentioned above, the actual simulations performed is not really well described. In Section 3.3 you say:** *"Temperature and precipitation are perturbed for each massif and elevation range based the historical period"*, **but never clearly say:** *"The model is run for XXX regions, YYY years, etc.***".**

We are sorry for the misunderstanding. We have changed: "The data includes flat slopes at low, mid and high elevation ranges and Pyrenean massifs (Figure 1) at hourly resolution" for:

"The FSM2 was run at an hourly resolution for each massif, each elevation range, and each climate perturbation scenario from 1980 to 2019".

**Moreover, for all the first part of the analysis, the spatialpatterns are not discussed, and the difference in massifs only appears in the discussion). As a reader I was confused until reaching the bottom of page 15 to know whether the model wasreally run for different locations, or only for different elevation bands.**

Thank you for your suggestion. The spatial patterns were already included in the results section (manuscript first version, L368 paragraph: … Snow climate sensitivity shows remarkable spatial contrasts… etc), not only on the discussion section.

The model was run at hourly resolution for each massif, elevation band and climate perturbed scenario (it is mentioned in the methodological section, and it can be observed at Figure 9 and 10).

**The procedure should be really explained (see my minor comment about a missing global study description at the end of the intro, which can help). Naming the massifs in Figure 1 andhaving a table briefly describing each massif (e.g. with min/mean/max elevation) would be useful for the analysis and help to clarify that the model is indeed run per massif. Another unclear point for me is the elevation used. Did you run only three elevations of some groups of elevation based on the 300m discretization of SAFRAN?**

Thank you for your suggestion. It does not exist different elevations (min/mean/max) for each massif, given that SAFRAN system provides data every 300 m, from 0 to 3600 m. We defined the low, mid, and high elevation bands that we used: Low, mid and high elevation corresponds to 1500, 1800 and 2400 m, respectively, specific elevation bands. The model was run at hourly resolution per each massif, elevation band and climate (baseline and perturbed) scenario.

**In Section 4.2, you should clearly state that all massifs are grouped together for the present analysis (and that the spatial analysis is performed later on). As far as I understand, Figure 4 shows the average across all massif. This should be clearly stated.**

Figure 4 is the average for each elevation band. We have modified the figure and figure caption:

"Figure 4. Average daily values for season type, baseline climate and different temperature increases at (a) high (b) mid and (c) low elevation."

**Also, in the whole Section 4.2 changes in precipitation are not mentioned (except in the caption of Figure 5), and only shown in Supplementary Figures. However, the fact that precipitation (+10%) could contract a 1°C is presented as one of your key results. The corresponding Figures should thus be properly shown, introduced, and described in the main text and in the Results Section (now Supplementary figures are just mentioned in the Discussion section).**

Thank you for your suggestion. We have rearranged the information and figures. We have added Figure 7 (previous Figure S4), following your suggestion to show the influence of precipitation in the snowpack evolution. Our results have been focused on seasonal snow-related changes due to increments of temperature, elevation, interannual variability – season type –, and spatial differences. These are the key factors that ruled the snowpack variation. We prefer to not add more details about precipitation given that precipitation only can counterbalance warming at high elevations, during the core months of winter, and if temperature do not exceed > 1ºC with respect of the 1980-2019 climate.

**Impact study, determining factors, uncertainty**

**I have the feeling that Section 4.2 is a long list of numbers a bit hard to follow, and in many cases the text repeats numbers shown in the Figures and Tables. Moreover, I feel a significant part of the numbers mentioned in Section 4.2 are nor really useful to support the latter analysis. In addition, this study inspects many aspects: different temperature and precipitation pathways, different kinds of compound seasons, and many sub-regions. In addition, they are analysed using 5 indicators, resulting on hundreds of different "numbers" to discuss. In the discussion, it is hard to really see the direction. Indeed, while the title suggests a focus on compound seasons, this is not really present in some part of the discussion (i.e., 5.4, which summarizes well known impacts and is in my opinion not necessary here, or 5.2.1, winch basically say that if we have more solid precipitation, we have higher snowpack). I would encourage to maybe reduce and reorganize the discussion and to only focus on few points (e.g. compound season and spatial distribution). A large amount of data has been produced for this study and it can be tempting to discuss every aspects of the data obtained from the model, but this makes it harder to read, and hide what is**

**really the novelty of this work. Note that the plots about spatial distribution are introduced in the discussion, while in my opinion they belong to the results Section.**

We are grateful of the reviewer comment, but we consider that we have followed a chronological order to discuss the results. We have focused the results and discussion on snow accumulation, ablation, season type differences, spatial patterns, environmental impacts, limitations, and uncertainties of the work. As far as we could, we have avoided to express numbers in the text. We have discussed the main results, and unfortunately there is not many more research that analyze the links between compound extremes seasons and snowpack evolution.

We agree with the reviewer and Figure 9 and 10 and associated text have been moved from the discussion to the results section.

**I've one concern about the method itself. As far as I understand, seasons "classes" (WW, CW,etc.) are determined for each subregion and elevation range separately (Figure S1). And thus,figures like 4 are obtained by averaging all the regions together for each elevation band and season class. My problem is that from Figure S1 we see that some classes of season are manly dominated by some regions (e.g. cold wet is dominated by south-west regions). So, when comparing the different season class, we do not really know if the difference is due to the meteorological input, of due to some other aspects differing between regions. In addition,the season class is (maybe?) determined for each region separately (see my comment above),so a CW in one region might not be CW in another region. As a consequence, because of theapproach chosen, I do not think the differences observed between compound seasons is onlydue to the specific weather of the seasons. This is probably the dominant factor, but the spatial difference would add some uncertainty there. This should at least be discussed. Note that there is no discussion about uncertainty and limitation, this should be added.**

Thank you for your comment. The information about the season type classification was detailed in the methodological section: "Compound temperature and precipitation extreme season (season type) is performed using a joint quantile approach (Beniston and Goyette, 2007; Beniston, 2009; López-Moreno et al., 2011a), for each massif and elevation ranges".

We have changed that for: "Compound temperature and precipitation extreme season (season type) is performed based on each massif and elevation historical climate record (1980-2019), using a joint quantile approach (Beniston and Goyette, 2007; Beniston, 2009; López-Moreno et al., 2011a). Season types are classified based on the massif and elevation historical record. We are not comparing season types between massifs. If we classify the season types based on the entire range percentiles, some extreme season types, such as CW, will be significantly reduced in the driest zones.

In the methodological section we have already mentioned that snow is modeled for flat slopes. We consider that we are already presenting the spatial differences for each season type in the results and discussion. Differences between regions (Figure 9 and 10) are due to meteorological input data. The massifs of the Eastern area are exposed to higher rates of radiative and turbulent heat fluxes and the snowpack is near to the isothermal conditions during the season shoulders. Therefore, a small increase of temperature leads to higher snow losses, especially during WW seasons.

**Limitations and uncertainties**

We have followed the reviewer suggestion and we have included a limitation and uncertainty section:

"**5.5 Limitations and uncertainties**

The meteorological input data that we used to model snow were estimated for flat slopes and the regionalization system we used was based on the SAFRAN system. According to this system, a mountain range is divided into massifs with homogeneous topography. The SAFRAN system has negative biases in shortwave radiation, a temperature precision of about 1 K, and biases in the accumulated monthly precipitation of about 20 kg/m$^2$ (Vernay et al., 2021). The snow model used in this work (FSM2) is a physics-based model of intermediate complexity, and the estimates of snow densification are simpler than those from more complex models of snowpack; however, a more complex model does not necessarily provide better performance in terms of snowpack and runoff estimation (Magnusson et al., 2015). Biases in the SAFRAN system and biases related to the FSM were minimal because we quantified relative changes between a modeled snow scenario (climate baseline) and several perturbed scenarios. Finally, our estimates of snow sensitivity were based on the delta-approach, which considers changes in temperature and precipitation based on climate projections for the Pyrenees (Amblar-Francés et al., 2020), but assumes that the snow patterns of the reference climate period will be constant over time."

**Minor comments:**

**Abstract: Please do not use abbreviations in abstract, only full words.**

Changed.

**P2 L38-39, L44, …: Please sort citation in ascending order by year (throughout the whole manuscript).**

Done.

**P3 L67-68: I do not understand what "*coincides*" with "*low solar radiation periods*".**

The snow ablation onset occurs earlier in the season, coinciding with low solar radiation periods.

"...However, warming can slow the early snow ablation rate on the season (Pomeroy et al., 2015; Rasouli et al., 2015; Jennings and Molotch, 2020; Bonsoms et al., 2022; Sanmiguel-Vallelado et al., 2022) because of the earlier HS and SWE peak dates (Alonso-González et al., 2022), which coincide with periods of low solar radiation (Pomeroy et al., 2015; Musselman et al., 2017a)…"

**P4 L95-96: What do you mean by "mid-end 21$^{st}$ century"?**

Changed: "mid-end 21st century" to "for the next decades"

**P4 L107: "." Missing**

Done.

**P5 L112-113: "*To date, some studies pointed out different climate sensitivies on wet or dry years*". Can you please explain in one sentence the different results found.**

We prefer to simplify this section since we already discuss these studies in the 5.3 section.

**P4 L126-128: Here I would briefly describe the main steps used to achieve this objective**

The main steps (input data and model) are already presented in the abstract, data and methodology and conclusions.

**P5 L139: Which "lapse-rate"? Elevation lapse rate of precipitation?**

We have changed this paragraph: "Precipitation is mostly driven by large-scale circulation patterns (i.e., Zappa et al., 2015; Borgli et al., 2019), the jet-stream oscillation during winter (e.g., Hurell, 1995) and land-sea temperature differences (Tuel and Eltahir, 2020)"

**P5 L142: "being ~ 1000" change to "being on average …". Please clarify in the rest of the paragraph where "~" means "around" and where it means "on average".**

Thank you. Done

**P6 L177-178: You should provide the final retained configuration for reproducibility.**

Thank you. Done:

"We have evaluated different FSM2 model configurations (not shown) without significant differences in the accuracy and performance metrics. Therefore, we selected the most complex FSM2 configuration. In detail, albedo is calculated based on a prognostic function, with increases due to snowfall and decreases

due to snow age. Atmospheric stability is calculated as function of the Richardson number. Snow density is calculated as a function of viscous compaction by overburden and thermal metamorphism. Snow hydrology is estimated by gravitational drainage, including internal snowpack processes, runoff, refreeze rates, and thermal conductivity. Snow cover fraction is based on a linear function of HS."

**P7 Table 1: Seems coordinates are in lat/lon °, not in UTM. Units are missing for the two "distance" column. "Reference period" is never explained in the text (see also major comments on the calibration/validation description).**

We have added:"Lat/Lon °" and Reference period for "Validation period (years)"

**P8 L208: Please provide a reference for the implementation**

It is mentioned in the manuscript first version L171 (Essery, 2015). Also, we have added the snow model configuration.

**P8 L217: "in" section**

Changed for "Precipitation type was classified following the threshold approach used for the model validation" (according to reviewer 1).

**P9 L266: What do you mean by "by massif"?**

Each snow-climatological indicator is calculated for each massif and elevation band.

**P9 Section 4.1: R2 should be $R^2$**

Done.

**P11 L292: Refer to Figure 5 at the end of the sentence. What does "Here" refer to?**

We refer to low elevation (Figure 4).

We have changed "Here" for "At low elevation".

**P11 Figure 4: For comparison, you should also show the reference simulation (+0°C) in the Figure. There are some strange drops in snow height (see below).**

Thank you for your suggestion. We have added the baseline climate snow profile in Figure 4 and resolved the error (one day was missed when plotting the results).

**P13 Figure5: how are the boxes constructed? Different seasons (i.e different years) + changein precipitation + different massifs? Or do you have only one point averaging across all seasons for a given massif? You should explain how are the boxes (1 and 3 quartiles?), whiskers, and outliers are defined.**

Thank you for your recommendation. We have added in each boxplot:

"The solid black lines within each boxplot are the average. Lower and upper hinges correspond to the 25th and 75th percentiles, respectively. The whisker is a horizontal line at 1.5 interquartile range of the upper quartile and lower quartile, respectively. Dots are outliers. Data is grouped by season, season type, increment of temperature, precipitation variation, elevation, and massif".

**P14 Figure 6: Why not using a boxplot here as in Figure 5?**

Thank you. Wehave added a boxplot following your suggestion.

**P15 figure 7: How is this exactly computed? By "season" you mean the exact length of the ablation season (i.e. time between $HS_{max}$ and HS=0)?**

We detailed in the methodological section how snow ablation is calculated (average daily snow ablation for a snow ablation day).

**P16 L383-386: This kind of statement should be in the Introduction section:**

Thank you for your comment. It is mentioned in the introduction (first manuscript version L21), however, we intentionally repeated our statement since it is crucial to introduce the reader to the discussion section and reinforce the relevance and novelty of our work.

**P16 L393-398: I do not really see the added value of this information here**

Thank you for your suggestion. We cannot remove this information since it is necessary to understand the spatial patterns of snow in the mountain range, as well as the different spatial responses to warming that we have detected.

**P17 Figure 8, P20 Figure 9: Units missing**

Added.

**P17 L418-420: You should show plots supporting it (e.g. a plot of precipitation phase)**

We are grateful of your suggestion. In this sentence we refer to the changes in the snow dynamics reported by scientific literature. However, we have added Figure S4 where the snowfall fraction shifts due to warming can be found.

**P18 L438-440: Something is missing in this sentence, e.g. "The higher**

**average […]"**

Thank you. Changed.

**P18 Section 5.2.2: This is really interesting. In my work on hydrology, I found that on a warmer world discharge peak from snowmelt will occur earlier, but also be "flatter" (see Michel, 2022). I never went deeper in the analysis of the cause of the flattening. Your analysis on slower melt rate seems really relevant to answer this question.**

We appreciate your comment in this active research topic. We have provided a plausible explanation based on our work and previous studies:

"Climate warming leads to a cascade of physical changes in the SEB that increase snow ablation near the 0ºC isotherm. On overall, the average daily snow ablation showed moderate to low changes due to warming. Comparison between low and high elevations indicated slightly faster snow ablation at high elevations (Figure 8). This higher rate of snow ablation per season at high elevations (which have deeper snowpacks) are probably because the snow there lasts until late spring, when more energy is available for snow ablation (Bonsoms et al., 2022). Temperature increase does not imply significant changes in the daily snow ablation rate per season because warming decreases the magnitude of the snowpack (seasonal HS and peak HS max) and triggers an earlier onset of snowmelt (Wu et al., 2018). The earlier peak HS date at low and mid elevations (Table 4 and Figure 7) implies lower rates of net shortwave radiation, because snow melting starts earlier in warmer climates (Pomeroy et al., 2015), coinciding with the shorter days and lower solar zenith angle (Lundquist et al., 2013; Sanmiguel-Vallelado et al., 2022). Our results agree with the slow snow melt rates reported in the Northern Hemisphere from 1980 to 2017 (Wu et al., 2018). The results of previous studies were similar for subarctic Canada (Rasouli et al., 2014) and western USA snowpacks (Musselman et al., 2017b), but Arctic sites had faster melt rates (Krogh and Pomeroy, 2019)."

**P19 L464: "if" ☐ "in"**
Done.

**P19 L464-466: With all the uncertainty involved, I would say "is similar"**

Done.

**P19 L467: A reference is needed here.**
Done.

Thank you for your constructive suggestions.

---

## Referee Report (RR1)

**Second review of "Snow sensitivity to climate change during compound cold-hot and wet-dry seasons in the Pyrenees"**

Dear authors, dear editor,

The revised version of the manuscript addresses some of the points raised in the first round of review. The method is better detailed, a section on limitations and uncertainty has been added, and the language has been improved. However, I still have some concerns. One is about the vocabulary used, which can easily be corrected, and a more fundamental one about the method, or rather how the results are discussed, which was already raised in the first round of comments and remained unanswered. While this may require more work than was done for the first iteration, I really encourage you to address this issue to increase the robustness of your results.

Best regards,
Adrien Michel

**Important note:** The updated manuscript does not agree with the track change version! E.g. P2L36 " … and **increasing** surface and air temperature …" in the manuscript, and " … and **reducing** surface and air temperature**s** …." in the track change version. The last sentence of the conclusion also differs (I did not check further). My comments are based on the updated manuscript. The author should carefully check on which version they are working on for further edits.

**Vocabulary**

There are been many improvements in the usage of the term "sensitivity" compared to the first version. However, in many locations, the word "sensitivity" is still used alone, while "sensitivity of snow to climate change should be used". E.g. section 4.2 is called "snow sensitivity", while it should be "snow sensitivity to climate change", or more precisely "Snow sensitivity to change in air temperature and precipitation".

In the text, you mainly use "extreme compounds seasons", while in the title you use "compound cold-hot and wet-dry seasons" only. While the term "extreme" could be disputable here (based on percentile 60), this is not unique in the literature (despite usually higher percentiles are used), so it is fine. However, for consistency you should maybe also use "extreme" in the title.

The term "ablation" is often used alone, while I think you are most of the time referring to "ablation rate", this should be corrected. Also, it is not clear if you talk about absolute ablation rate (cm/day) or relative ablation rate (in %/day or %/°C). Both of them are relevant, but it should be clarified to which one you refer to.

**Analysis description**

You added some details on P6L170-178. However, we still do not know exactly which model setup you used. If I want now to reproduce your work, I need the exact name of the models' parameters. I Imagine these parameters are chosen at compilation time form the compile.sh file (https://github.com/RichardEssery/FSM2/blob/master/compil.sh):

```
**define ALBEDO 2   /* snow albedo               : 1, 2    */**
**define CANMOD 1   /* forest canopy layers      : 1, 2    */**
**define CANRAD 1   /* canopy radiative properties: 1, 2    */**
**define CONDCT 1   /* snow thermal conductivity : 0, 1.   */**
**define DENSTY 1   /* snow density              : 0, 1, 2 */**
**define EXCHNG 1   /* turbulent exchange        : 0, 1    */**
**define HYDROL 1   /* snow hydraulics           : 0, 1, 2 */**
**define SNFRAC 1   /* snow cover fraction       : 1, 2    */**
/* Driving data options                 : Possible values.   */
**define DRIV1D 1   /* 1D driving data format    : 1, 2    */**
**define SETPAR 1   /* parameter inputs          : 0, 1    */**
**define SWPART 0   /* SW radiation partition    : 0, 1    */**
**define ZOFFST 0   /* measurement height offse  : 0, 1    */**
/* Output options                       : Possible values   */
**define PROFNC 0   /* netCDF output             : 0, 1    */**
```

A table in supplementary with the exact names (and maybe the exact version of the model, i.e. git commit number) is necessary for reproducibility.

There is no "data and code availability" indicated in the paper. I highly encourage you to publicly share your data.

**Impact study, determining factors, uncertainty**

On P11 L284-285 you say: "The results show a non-linear response between seasonal HS loss and temperature increase." Which is clear from figure 4 and an interesting result. However, later in the analysis, you mainly use linear indicators (in %/°C): e.g. P11-12 L292-319, Tables 2-3. I think these numbers do not add anything to the analysis and they contradict the non-linearity you found and emphasis in your abstract (saying that the greater relative change is for +1°C). I would recommend to remove the above-mentioned lines and tables. This will make this Section easier to read, without any loss of information. The same information is obtained by commenting the boxplots on Figs 5-6-7. As I already mentioned in the first review round, there are many numbers listed in the text, which are all visible from figures, and not really useful later on in the analysis.

In the first revision round, I raised this important concern:
"'I have one concern about the method itself. As far as I understand, seasons "classes" (WW,CW,etc.) are determined for each subregion and elevation range separately (Figure S1). And thus, figures like 4 are obtained by averaging all the regions together for each elevation band and season class. **My problem is that from Figure S1 we see that some classes of**

**season are manly dominated by some regions (e.g. cold wet is dominated by south-west regions). So, when comparing the different season class, we do not really know if the difference is due to the meteorological input, or due to some other aspects differing between regions.** In addition, the season class is (maybe?) determined for each region separately (see my comment above), so a CW in one region might not be CW in another region. **As a consequence, because of the approach chosen, I do not think the differences observed between compound seasons is only due to the specific weather of the seasons. This is probably the dominant factor, but the spatial difference would add some uncertainty there. This should at least be discussed.**"

Many points have been answered and clarified, but not what is in bold font in the paragraph above (you just answered "We are not comparing season types between massifs"). I'll reformulate here this concern.

In Figures 4 to 7, and in most of the analysis, you split the data per elevation range and per season type. In Figure S1, we see that some seasons types occur mor often in some regions (e.g. some regions in the south have no warm-wet, some regions have a total of ~80 extremes seasons, while some have only ~40 in total). As a consequence, when looking at one class of season and one elevation band, the different regions are not represented equally, on some seasons type signal will thus be influenced by the dominant regions in the sub-ensemble (e.g. col-wet mid altitude are dominated by the western regions). In Figures 9-10 you show that the response to climate change is different between region. So, when in the end you assess the change for a season type and an elevation band (e.g. col-wet mid altitude), we do not know if the response is more dependent on the season type, or on the region which is dominating this subset. Looking at the spread in the boxplots of Figure 6, I'm not sure that for all case we have a proper statistical difference between low and mid elevation band (this can be statistically tested). This can be explained by the fact that going from low to mid altitude the representation of the regions in the sub-ensemble considered in not the same. In other words, we cannot with certainty attribute the difference to the season type as you do in the analysis. Note that you can do some statistical tests to see if the different response to climate change between region, elevation band, and season-type are significant, excluding the two other parameters (note: using only one variable you would not have this problem, because by definition all region will have the same number of extreme, 40% of the seasons with the percentiles you use).

This imbalance between regions, induced by the joint quantile approach used, should be discussed in the text (note: it indeed totally makes sense to compute extreme per regions). This is not an insurmountable problem, but this is a clear drawback of the method used (as every method has). A clear example is the following sentence in the conclusion: "In particular, snowpack losses were greatest during WW seasons at low and mid-elevations and were greatest during WD seasons at high elevations". At mid-elevation, the eastern region has more WW event that the western one (Figure S1), at the same time, HS is more sensitive to climate change in the eastern regions (Figure 9). Now the question is: Is eastern more sensitive because of the local conditions (e.g. closer to isothermal conditions in the baseline simulation), and thus since it has more WW season, the WW signal will appear more sensitive simply because it contains more seasons from this region? Or is it the opposite: WW season are for some reason more sensitive to climate change, and eastern region having more WW season compared to the other regions, it appears to be more sensitive to climate change? Are the local conditions or the season types dominant here? The fact is that with this analysis we

cannot answer this question. We can see some correlation between season and sensitivity to climate change, yes, but we cannot attribute the observed different sensitivity to the season type, this is a major difference.

The extreme compound season is really emphasised in your abstract/title/conclusion, but:
1. The analysis suffers from the problem discussed above. We cannot do a proper attribution.
2. Is not that much discussed in the text in the end. Indeed, only one paragraph (Section 5.3) discusses it in the whole discussion. Finally, most of the discussion is based on more general results.

I think here lies my main problem with the current status of the manuscript. Either the focus is kept on the compound seasons (then maybe the general discussion on well know impacts of climate change on snow (Section 5.1 and 5.2) should be highly shortened), and the strength of the analysis on the difference in the signal between seasons should be improved (by using proper statistical test showing that season type is significant despite the imbalance in regions) and the problems mentioned above arising from the season type construction need to be discussed (which I encourage you to do); or you decide to be more generalist (as you are now in some parts of the discussion), and then you remove most of the emphasis on the compound season.

**Minor comments**

P1 L35: Should be "increasing" the albedo.

P2 L36: Should be "decreasing surface and air temperature". And it is not absolutely true that snow decreases surface temperature. During winter, the snow/soil interface will mostly remain at 0°C if the soil is snow covered, while if the soil is snow free but the air temperature is cold, the soil surface temperature will further decrease (snow is a really good insulator for the soil). In spring, it is true that snow cover will keep the soil colder. However, I would keep only "decreasing air temperature".

P2 L59: "on snowpack duration" → "on snow cover duration"

P3 L85-86: "and the different mountain exposure to the main air masses". Which "main air masses"?

P4 L95: What is "mid-late"?

P4 L105: "Sensitivity" to what? (Same at lines 107, 108, 110, Sections 3.4 and 4.2 names, etc). Should always be "sensitivity to climate change", this is indeed a bit heavier in the text, but correct (see comment above).

P4 L115: What does "these" refer two? Long sentence with many commas, hard to follow, consider splitting in two sentences.

P9 L259: Do you mean "ablation rate"?

P9 L262: Why isn't snow duration also an accumulation indicator?

P11 L287-288: "High elevation areas had lower season-to-season snow variability than low elevations for all season types (Figure 4)". I don't see any information about season-to-season variability in Figure 4.

P11 L 289: Avoid using the word "significantly" if not in the context of a proper statistical analysis (and thus a statistical "significance").

P11 L289-291: "All the snowpack-perturbed scenarios indicated that snowpack decreased at low and mid elevations under warming climate scenario". This is also the case at high elevation (Figure 4).

P13 L331: "the peak HS date per °C was earlier by 9 days". This does not mean anything to me, should be: "the peak HS was anticipated by 9 days per °C" (but see my comment on linear indicators).

P13 L337: "and because": Remove "and"

P13-14 Figures 5-6: Why not having only one figure with three rows of boxplot?

P14 L350: "At low elevations, the snow ablation in all four extreme seasons was 12%/°C". Something is missing here. Should be "snow ablation rate increase". Same for the rest of the paragraph, should be ablation rate change/increase/decrease.

P16 Figure 8: I think the y-axis units should be cm/day. Explain the numbers in the caption, e.g. "the numbers in the plot show the difference in ablation rate compared to the previous degree". Or maybe I don't understand the figure, and this just are absolute values of ablation rate. So then why stacking them on top of each other and why having a y-axis? Shouldn't it be like figure 7? Also, I can't reconciliate what I shown in this figure with the numbers in Table 4 for the ablation columns. In the text and Table 4, you have ablation in %/°C, and here in cm/day.

P18L398: "The sensitivity of snow to different spatial patterns of climate change that we identified here […]". You do not study different patterns of climate change; you apply the same delta everywhere. Please correct.

P21L504-505: "Our maximum snow ablation and peak HS date occurred during dry seasons …]". Are you talking about change or about absolute value in the baseline simulation?

P21L508-510: "The temperature in the Pyrenees is still cold enough to allow snowfall at high elevations during WW seasons, and for this reason we found maximal sensitivities during WD seasons." I don't understand, temperature is almost the same in WW and in WD, so why is sensitivity to climate change greater in WD?

P23L562-564: "however, a more complex model does not necessarily provide better performance in terms of snowpack and runoff estimation (Magnusson et al., 2015)". But it also

can, especially for climate change study (see Carletti et al, 2022, doi.org/10.5194/hess-26-3447-2022)

P564-566: "Biases in the SAFRAN system and biases related to the FSM2 were minimal because we quantified relative changes between a modeled snow scenario (climate baseline) and several perturbed scenarios". This assumes constant biases. Snow cover involves different variables, non-linear processes, and will accumulate errors along the season, we cannot be that certain.

P23L568-569: "[…] but assumes that the snow patterns of the reference climate period will be constant over time." Don't you mean meteorological patterns? (E.g. you don't capture change in precipitation regime, you just scale the intensity of each events). Please clarify.

---

## Referee Report (RR2)

Dear Editor, Dear Authors,

I would like to acknowledge the work done on the manuscript. All my concerns have been satisfactorily addressed (either by incorporating my recommendations or by clear responses). The additional text and figures improve the manuscript. I have only a few remaining minor points, which are listed below.

Best regards,
Adrien Michel

I fully agree with the extended argumentation on the contradiction between non-linearity and the use of a linear indicator (%/°C) to facilitate interpretation and comparison between massifs and with other studies. I appreciate the more detailed analysis between massifs, which 1) confirms the main results and 2) strengthens the analysis.

I would only recommend adding some details on the PCA. It is not clear to me on which sets of variables the PCA is performed. The two references given did not allow me to fully understand. Just one or two sentences explaining exactly on which data set the PCA is performed would be very helpful.

Finally, in the conclusion (l.628) you say: "[…] in this region, a 10% increase of precipitation, as suggested by many climate projections over the eastern regions of this range, could compensate for temperature increases on the order of about < 1°C. " However, the only references I can find in the text are on line 96: "an increase (decrease) of precipitation by about 10% for the eastern (western) regions during winter and spring (Amblar-Francés et al., 2020)" and on line 479: "Snow sensitivity in the easternmost areas could decline during the winter because of a trend for an increase of about 10% in precipitation in this area (Amblar-Francés et al., 2020)". A single reference is not enough to state: "as suggested by many climate projections". You could add some references or rephrase this sentence.

---

## Author Response (AR3)

Josep M Bonsoms
Department of Geography (University of Barcelona)
Montalegre 6-8, 3er floor
08001 - Barcelona (Spain)
Tel.: +34 655 36 49 42 / +34 934 037 849
josepbonsoms5@ub.edu

[Figure]
* * *
Barcelona, 6th January 2023

Dear Dr. Helbig,

I am submitting the revised version of the manuscript entitled "**Snow sensitivity to climate change during compound cold-hot and wet-dry compound seasons in the Pyrenees**", co-authored by myself, Dr. López-Moreno and Dr. Alonso-González.

We want to express our sincere gratitude for the time you expended reviewing the manuscript, your constructive recommendations and positive feedback.

We would also like to thank the referee's recommendations, which helped to improve the manuscript and added scientific rigor to the research presented here.

The main manuscript modifications are summarized as follows:

**(I) We have changed the manuscript according to your suggestions.** We have changed Figure 4 absolute for relative values. Also, we have carefully checked that the submitted manuscript corresponds to the track change version of the revised manuscript.

**(II) We have included referee 1 methodological correction.** We defined the seasonal peak HS date after applying a moving average of 5-days to resolve the issue mentioned.

**(III) We have included most of referee 2 suggestions**. We have carefully addressed the referee 2 comment about the joint-quantile approach. We have statistical classified the Pyrenean zones, added more data, a new results section and three figures. We have evaluated the snow sensitivity to temperature and precipitation differences due to the Pyrenean sectors, and the number of compound season types recorded during the baseline period. However, no remarkable differences in the relative importance of each compound season type were found between sectors. This is because by applying a joint-quantile approach for each massif and elevation, we are standardizing the climate of the Pyrenean massifs. We are comparing the snow sensitivity to temperature and precipitation change during similar climate seasons, independently where a climate season type was recorded.

A point-by-point answer to reviewer's comments can be found in the following pages.

We expect to fulfil the expectations and we hope the new manuscript version is suitable for publication in **The Cryosphere**. I will be happy to answer any question you might have regarding this work.

Many thanks, kind regards and happy new year,

[Figure]

Josep Bonsoms, on behalf of the co-authors.

**Response to second review of "Snow sensitivity to temperature and precipitation change during compound cold-hot and wet-dry seasons in the Pyrenees"**

**by Josep Bonsoms[1], Juan Ignacio López-Moreno[2] and Esteban Alonso[3]**

[1] Department of Geography, University of Barcelona, Barcelona, Spain.

[2] Instituto Pirenaico de Ecología (IPE-CSIC), Campus de Aula Dei, Zaragoza, Spain.

[3] Centre d'Etudes Spatiales de la Biosphère (CESBIO), Université de Toulouse, CNES/CNRS/IRD/UPS, Toulouse, France.

Corresponding author: J.I López-Moreno (nlopez@ipe.csic.es)

Response to Reviewer 1.

Reviewer comments are in black and responses in blue.

Thanks to authors for the clear and concise reply. The paper improved a lot. I like the updated Figure 4, but the new Figure 7 sheds light on a problem in Figure 4. The shown seasonal evolution of HS in Figure 4 reflects a false accuracy of model results, which are than translated into the interpretation of e.g. Figure 7, especially for the WW compound extreme. There, a clear later peak HS date only for mid elevation under 1°C or 2°C warming (or the positive sensitivity for the same case in Table 4) make not much sense.

Therefore, I'd suggest applying a smoothing filter to HS evolution in Figure 4 in order to prevent to have several peaks.

Thank you very much for your suggestion.
We have applied your suggestion, and we have changed the peak HS date quantification to prevent several peak HS dates per season. The peak HS date is now determined after applying a 5-day moving average, which resolves the issue you mentioned.
Figure 4 is also changed following editor recommendation.
We have added:
"Some seasons had more than one peak HS; for this reason, peak HS date was determined after applying a moving average of 5 days. All indicators were computed according to massif and elevation range."

**Response to second review of "Sensitivity to temperature and precipitation change during compound cold-hot and wet-dry seasons in the Pyrenees"**

**by Josep Bonsoms[1], Juan Ignacio López-Moreno[2] and Esteban Alonso[3]**

[1] Department of Geography, University of Barcelona, Barcelona, Spain.

[2] Instituto Pirenaico de Ecología (IPE-CSIC), Campus de Aula Dei, Zaragoza, Spain.

[3] Centre d'Etudes Spatiales de la Biosphère (CESBIO), Université de Toulouse, CNES/CNRS/IRD/UPS, Toulouse, France.

Corresponding author: J.I López-Moreno (nlopez@ipe.csic.es)

Response to Reviewer 2.

Reviewer comments are in black and responses in blue.

Dear authors, dear editor,

The revised version of the manuscript addresses some of the points raised in the first round of review. The method is better detailed, a section on limitations and uncertainty has been added, and the language has been improved. However, I still have some concerns. One is about the vocabulary used, which can easily be corrected, and a more fundamental one about the method, or rather how the results are discussed, which was already raised in the first round of comments and remained unanswered. While this may require more work than was done for the first iteration, I really encourage you to address this issue to increase the robustness of your results.
Best regards,
Adrien Michel

**Important note:** The updated manuscript does not agree with the track change version! E.g. P2L36 " … and **increasing** surface and air temperature …" in the manuscript, and " … and **reducing** surface and air temperature**s** …." in the track change version. The last sentence of the conclusion also differs (I did not check further). My comments are based on the updated manuscript. The author should carefully check on which version they are working on for further edits.

We would like to thank your time expended reviewing the manuscript, and your constructive suggestions.

We have followed your suggestions and we have corrected the vocabulary, modified the text and figures. We have also added more data and figures to evaluate the regional differences induced by the joint-quantile approach.

A point by point answer to reviewer 2 recommendations can be found in the following lines.

**Vocabulary**

There are been many improvements in the usage of the term "sensitivity" compared to the first version. However, in many locations, the word "sensitivity" is still used alone, while "sensitivity of snow to climate change should be used". E.g. section 4.2 is called "snow sensitivity", while it should be "snow sensitivity to climate change", or more precisely "Snow sensitivity to change in air temperature and precipitation".

We have followed Reviewer 2 suggestion; we have changed "sensitivity" alone to snow sensitivity to temperature and precipitation. Also, we have changed "snow sensitivity to climate change" for "snow sensitivity to temperature and precipitation change" through the manuscript, and we have modified the manuscript title.

**Abstract:**

"…Here, we perform a snow sensitivity to temperature and precipitation change analysis of the Pyrenean snowpack (1980 – 2019 period) using five key snow-climatological indicators.".

**Introduction:**

"…Therefore, the main objective of this research is to quantify snow (accumulation, ablation, and timing) sensitivity to temperature and precipitation change during compound temperature and precipitation seasons in the Pyrenees."

**Method:**

In this method, air temperature and precipitation were perturbed for each massif and elevation range based the historical period (1980–2019). Air temperature was increased from 1 to 4ºC at 1ºC, assuming an increase of $LW_{inc}$ accordingly.

In the text, you mainly use "extreme compounds seasons", while in the title you use "compound cold-hot and wet-dry seasons" only. While the term "extreme" could be disputable here (based on

percentile 60), this is not unique in the literature (despite usually higher percentiles are used), so it is fine. However, for consistency you should maybe also use "extreme" in the title.

We have followed your suggestion and we have included "compound cold-hot and wet-dry seasons" in the title. We have deleted the word "extreme" according to your suggestion.

The term "ablation" is often used alone, while I think you are most of the time referring to "ablation rate", this should be corrected. Also, it is not clear if you talk about absolute ablation rate (cm/day) or relative ablation rate (in %/day or %/°C). Both of them are relevant, but it should be clarified to which one you refer to.

We have defined in the methodological section snow ablation : snow ablation is the difference between the maximum daily HS recorded on two consecutive days.

In the text: "…daily average snow ablation per season (snow ablation, hereafter)". Subsequently, we have applied the term "snow ablation" through the manuscript accordingly.

**Analysis description**

You added some details on P6L170-178. However, we still do not know exactly which model setup you used. If I want now to reproduce your work, I need the exact name of the models' parameters. I Imagine these parameters are chosen at compilation time form the compile.sh file (https://github.com/RichardEssery/FSM2/blob/master/compil.sh). A table in supplementary with the exact names (and maybe the exact version of the model, i.e. git commit number) is necessary for reproducibility.

Thank you for your suggestion.

We have followed your recommendation and added a table in supplementary material with the physical configuration and compile numbers.

In the methodological section we have added: "…. Snowpack was modelled using a physical-based snow model, the Flexible Snow Model (FSM2; Essery, 2015). This model resolves the SEB and mass balance to simulate the state of the snowpack. FSM2 is open access and available at https://github.com/RichardEssery/FSM2 (last access 16 December 2022). Previous studies tested the FSM2 (Krinner et al., 2018), and its application in different forest environments (Mazzoti et al., 2021), and hydro-climatological mountain zones such the Andes (Urrutia et al., 2019), Alps (Mazzoti et al., 2020), Colorado (Smyth et al., 2022), Himalayas (Pritchard et al., 2020), Iberian Peninsula Mountains (Alonso-González et al., 2020a; Alonso-González et al., 2022), Lebanese

mountains (Alonso-González et al., 2021), providing confidential results. The FSM2 requires forcing data of precipitation, air temperature, relative humidity, surface atmospheric pressure, wind speed, incoming shortwave radiation ($SW_{inc}$), and incoming long wave radiation ($LW_{inc}$). We have evaluated different FSM2 model configurations (not shown) without remarkable differences in the accuracy and performance metrics. The FSM2 configuration included in this work estimated snow cover fraction based on a linear function of HS and albedo based on a prognostic function, with increases due to snowfall and decreases due to snow age. Atmospheric stability is calculated as function of the Richardson number. Snow density is calculated as a function of viscous compaction by overburden and thermal metamorphism. Snow hydrology is estimated by gravitational drainage, including internal snowpack processes, runoff, refreeze rates, and thermal conductivity. Table S1 summarizes the FSM2 configuration and the FSM2 compile numbers."

Table S1. FSM2 configuration implemented.

| Physics and driving data options | FSM2 Configuration | FSM2 Compile number |
|---|---|---|
| Albedo | Prognostic age function | 2 |
| Snow conductivity | Function of density | 1 |
| Snow density | Function of overburden | 2 |
| Turbulent exchange | Richardson number atmospheric stability adjustment | 1 |
| Snow hydrology | Gravitational drainage | 2 |
| Snow cover fraction | Linear function of snow depth | 1 |

There is no "data and code availability" indicated in the paper. I highly encourage you to publicly share your data.

We have modified the data and methods section. Also, we have added a data availability section following your suggestion.

**Data and methods :**

"…FSM2 is open access and available at https://github.com/RichardEssery/FSM2 (last access 16 December 2022)."

"…We forced the FSM2 with the open access SAFRAN climate reanalysis dataset described    by Vernay et al. (2021)"

**Data availability:**

Snow model (FSM2) is open access and available at https://github.com/RichardEssery/FSM2 (last access 16 December 2022). Meteorological Forcing data is described by Vernay et al. (2021), through AERIS (https://www.aeris-data.fr/landing-page/?uuid=865730e8-edeb-4c6b-ae58-80f95166509b#v2020.2; last access 16 December 2022). Data of this work is available upon request (contact: josepbonsoms5@ub.edu).

**Impact study, determining factors, uncertainty**

On P11 L284-285 you say: "The results show a non-linear response between seasonal HS loss and temperature increase." Which is clear from figure 4 and an interesting result. However, later in the analysis, you mainly use linear indicators (in %/°C): e.g. P11-12 L292- 319, Tables 2-3. I think these numbers do not add anything to the analysis and they contradict the non-linearity you found and emphasis in your abstract (saying that the greater relative change is for +1°C). I would recommend to remove the above-mentioned lines and tables. This will make this Section easier to read, without any loss of information. The same information is obtained by commenting the boxplots on Figs 5-6-7. As I already mentioned in the first review round, there are many numbers listed in the text, which are all visible from figures, and not really useful later on in the analysis.

Thank you for your recommendation.

We want to show both (I) the average sensitivity to temperature and precipitation change (% per °C) and (II) the seasonal decreases by increments of temperature, elevation, season type and regions because:

1. The reader is informed in the abstract and the main text about the seasonal HS non-linearity evolution when temperature is progressively increased:

"When the temperature increased progressively at 1°C intervals, the largest seasonal HS decreases from the baseline were at +1°C (47% at low elevation, 48% at mid-elevation, and 25% at high elevation)."

And:

"Our results suggest that warming had a non-linear effect on snowpack reduction. Our largest snow losses were for seasonal HS when the temperature increased by 1°C above baseline."

2. We used the average sensitivity to temperature and precipitation change (% per °C) in benefit of the results interpretation. We consider that it is easier to understand for the reader. Details (i.e.,

seasonal losses by increment of each temperature, season type, elevation, etc) can be consulted in the figures.

3. The mean value is comparable between the compound seasons, elevation, and sectors.

4. The average snow sensitivity to changes in meteorological forcing variables (expressed in %/ºC) has been applied and validated in snow hydrology (i.e., Pomeroy et al., 2015; Brown and Mote, 2009; Musselman et al., 2017a; Esteban-Alonso et al., 2020; López-Moreno et al., 2021, etc.). The inclusion of the average allows us to compare to those previous works and make our results comparable in the future. The average (expressed in %/ºC) to changes in model forcing data has been used in other cryosphere topics. Just to name a few:

Anderson, B., and A. Mackintosh (2012), Controls on mass balance sensitivity of maritime glaciers in the Southern Alps, New Zealand: The role of debris cover, J. Geophys. Res., 117, F01003, doi:10.1029/2011JF002064.

Pomeroy J, Fang X, Ellis C. 2012. Sensitivity of snowmelt hydrology in Marmot Creek, Alberta, to forest cover disturbance. Hydrological Processes 26: 1892-1905. doi:10.1002/hyp.9248.

Rasouli K, Pomeroy JW, Marks, DG. 2015. Snowpack sensitivity to perturbed climate in a cool mid-latitude mountain catchment. Hydrological Processes 29: 3925–3940. doi: 10.1002/hyp.10587.

Some of them have been included in The Cryosphere, for instance:

Burke, E. J., Zhang, Y., and Krinner, G.: Evaluating permafrost physics in the Coupled Model Intercomparison Project 6 (CMIP6) models and their sensitivity to climate change, The Cryosphere, 14, 3155–3174. Doi:10.5194/tc-14-3155- 2020, 2020.

Ebrahimi, S., & Marshall, S. J. (2016). Surface energy balance sensitivity to meteorological variability on Haig Glacier, Canadian Rocky Mountains. The Cryosphere, 10, 2799–2819. Doi:10.5194/tc-10-2799-2016

van Pelt, W. J. J., Oerlemans, J., Reijmer, C. H., Pohjola, V. A., Pettersson, R., and van Angelen, J. H.: Simulating melt, runoff and refreezing on Nordenskiöldbreen, Svalbard, using a coupled snow and energy balance model, The Cryosphere, 6, 641–659, doi:10.5194/tc-6-641-2012, 2012.

Reviewer suggestion was not implemented but if the editor considers that we should modify it, we will change the manuscript accordingly.

In the first revision round, I raised this important concern:

"'I have one concern about the method itself. As far as I understand, seasons "classes" (WW,CW,etc.) are determined for each subregion and elevation range separately (Figure S1). And thus, figures like 4 are obtained by averaging all the regions together for each elevation band and season class. **My problem is that from Figure S1 we see that some classes of season are manly dominated by some regions (e.g. cold wet is dominated by south-west regions).**

**So, when comparing the different season class, we do not really know if the difference is due to the meteorological input, or due to some other aspects differing between regions.**

In addition, the season class is (maybe?) determined for each region separately (see my comment above), so a CW in one region might not be CW in another region. **As a consequence, because of the approach chosen, I do not think the differences observed between compound seasons is only due to the specific weather of the seasons. This is probably the dominant factor, but the spatial difference would add some uncertainty there. This should at least be discussed**." Many points have been answered and clarified, but not what is in bold font in the paragraph above (you just answered "We are not comparing season types between massifs"). I'll reformulate here this concern. In Figures 4 to 7, and in most of the analysis, you split the data per elevation range and per season type. In Figure S1, we see that some seasons types occur mor often in some regions (e.g. some regions in the south have no warm-wet, some regions have a total of ~80 extremes seasons, while some have only ~40 in total). As a consequence, when looking at one class of season and one elevation band, the different regions are not represented equally, on some seasons type signal will thus be influenced by the dominant regions in the sub-ensemble (e.g. col-wet mid altitude are dominated by the western regions). In Figures 9-10 you show that the response to climate change is different between region. So, when in the end you assess the change for a season type and an elevation band (e.g. col-wet mid altitude), we do not know if the response is more dependent on the season type, or on the region which is dominating this subset.

Looking at the spread in the boxplots of Figure 6, I'm not sure that for all case we have a proper statistical difference between low and mid elevation band (this can be statistically tested). This can be explained by the fact that going from low to mid altitude the representation of the regions in the sub-ensemble considered in not the same. In other words, we cannot with certainty attribute the difference to the season type as you do in the analysis. Note that you can do some statistical tests to see if the different response to climate change between region, elevation band, and season-type are significant, excluding the two other parameters (note: using only one variable you would not have this problem, because by definition all region will have the same number of extreme, 40% of the seasons with the percentiles you use). This imbalance between regions, induced by the joint quantile approach used, should be discussed in the text (note: it indeed totally makes sense to compute extreme per regions). This is not an insurmountable problem, but this is a clear drawback of the method used (as every method has). A clear example is the following sentence in the conclusion: "In particular, snowpack losses were greatest during WW seasons at low and mid-elevations and were greatest during WD seasons at high elevations". At mid-elevation, the eastern region has more WW event that the western one (Figure S1), at the same

time, HS is more sensitive to climate change in the eastern regions (Figure 9). Now the question is: Is eastern more sensitive because of the local conditions (e.g. closer to isothermal conditions in the baseline simulation), and thus since it has more WW season, the WW signal will appear more sensitive simply because it contains more seasons from this region?

Or is it the opposite: WW season are for some reason more sensitive to climate change, and eastern region having more WW season compared to the other regions, it appears to be more sensitive to climate change? Are the local conditions or the season types dominant here? The fact is that with this analysis we cannot answer this question. We can see some correlation between season and sensitivity to climate change, yes, but we cannot attribute the observed different sensitivity to the season type, this is a major difference.

The extreme compound season is really emphasised in your abstract/title/conclusion, but:

1. The analysis suffers from the problem discussed above. We cannot do a proper attribution.

2. Is not that much discussed in the text in the end. Indeed, only one paragraph (Section 5.3) discusses it in the whole discussion. Finally, most of the discussion is based on more general results.

I think here lies my main problem with the current status of the manuscript. Either the focus is kept on the compound seasons (then maybe the general discussion on well know impacts of climate change on snow (Section 5.1 and 5.2) should be highly shortened), and the strength of the analysis on the difference in the signal between seasons should be improved (by using proper statistical test showing that season type is significant despite the imbalance in regions) and the problems mentioned above arising from the season type construction need to be discussed (which I encourage you to do); or you decide to be more generalist (as you are now in some parts of the discussion), and then you remove most of the emphasis on the compound season.

Thank you for your recommendation.

We have modified the manuscript accordingly. We have added more data and figures to evaluate the results obtained by using the joint-quantile approach and answer reviewer 2 comments. We proceed with the analysis suggested by the reviewer, and we have analyzed sensitivity to temperature and precipitation change by sectors of the range. We statistical classified the massifs by applying a PCA, a broadly applied technique in snow climatology.

Comparison between sectors reveal almost the same relative importance of each compound season type in the sensitivity to temperature and precipitation change for the entire range than for Pyrenean sectors. Thus, our results confirm that although there are different number of season types by sector (Figure S1 and S3), no differences in the relative importance of each compound season type on the sensitivity to temperature and precipitation change are found between Pyrenean regions. For instance, for most indicators, maximum to minimum snow sensitivities to temperature and precipitation ranges from WW to CW, independently of the sector and the number of season types recorded by massif. Thus, we can split the data by sectors (with different number of season types recorded during the baseline period by each massif, as shown in Figure

S1 and S3) or by massifs (it was shown at first version manuscript Figure 9 and 10), but the relative importance of each compound season type is similar than if we do the average for the entire range and by elevation bands. This is because by applying the join-quantile approach (season types are defined by each massif and elevation temperature and precipitation percentiles historical records) we are comparing similar climate seasons, independently of the sector where the season type was recorded. In the end, we are standardizing the climate of the Pyrenean massifs. The maximum sensitivity to temperature and precipitation change (absolute values) is always reached in the southern-eastern Pyrenees, independently of the season type. This is because this leeward side is exposed to higher turbulent and radiative heat fluxes, and this sector is closer to the 0ºC isotherm.

Results section 4.2 are focused on the sensitivity to temperature and precipitation change analysis due to increments of temperature, elevation, and compound season type. These variables have larger influence in the sensitivity to temperature and precipitation change than spatial differences, because the relative importance of the latter is reduced by applying the joint-quantile approach. Spatial differences in the sensitivity to temperature and precipitation change (absolute values) were already examined in our manuscript first version, but now we included a title to the results paragraph (section 4.3), and we provide an analysis by sectors defined by a PCA. Finally, in the discussion section, we explain the reason why the number of seasons by massif does not influence in the results obtained.

In this case, we prefer not reducing section 5.1 and 5.2, especially when we are within the word limit proposed by The Cryosphere. We consider that this information should be included to better understand our results. Without an accurate contextualization the numbers provided in the results section do not have any meaning, and for this reason we consider important to not reducing results discussion. We compare our work with general snow sensitivity studies and snow climate projections because of its similarities. As we state in the introduction, there are not many works comparing snow sensitivities to temperature and precipitation and compound season types (or dry/ wet seasons), which limits the discussion of our results with this literature but provides evidence of the novelty of our findings.

If the editor considers that we should modify it, of course, we will implement such changes.

We have modified the methodological section, results and discussion of the manuscript and added figures according to your comments and our findings:

Method

We have added an accurate description of the standardizing procedure that we are doing by applying the joint-quantile approach:

"…Note that the number of compound season type is different depending on the Pyrenees massif (Figure S1). However, by applying the joint-quantile approach described, we are comparing the

snow sensitivity to temperature and precipitation change between similar climate conditions, independently where each compound season type was recorded."

"3.7 Spatial regionalization

We have examined spatial differences in the sensitivity to temperature and precipitation change by compound season types. Massifs were grouped into four sectors by applying a Principal Component Analysis (PCA) of HS data (i.e., López-Moreno et al., 2020b; Matiu et al., 2020) and for each elevation depending on PC1 and PC2 scores. PCA scores are shown at Figure S2, the number of season types per sector are shown at Figure S3 and the spatial regionalization is presented at Figure 1."

Results

We have added in the first line of results:

"4. Results
We validated the FSM2 at Section 4.1. Subsequently, we analyzed the sensitivity to temperature and precipitation change based on five snow climate indicators, namely the seasonal HS, peak HS max, peak HS date, snow duration and snow ablation. Compound season types show similar relative importance on the sensitivity to temperature and precipitation change regardless of the Pyrenean sector. For this reason, our results have been focused on seasonal snow changes due to increments of temperature, elevation, and compound season type. These are the key factors that ruled the sensitivity to temperature and precipitation change and an accurate analysis is provided in Section 4.2. Spatial differences on the sensitivity to temperature and precipitation change during compound season types are examined at Section 4.3."

4.3 Spatial patterns

PCA analysis reveals four Pyrenean sectors, namely northern-western (NW), northern-eastern (NE), southern-western (SW), and southern-eastern (SE). No differences between sectors are found in the relative importance of each compound season type in the sensitivity to temperature and precipitation change (Figure 8). Snow sensitivity to temperature and precipitation change absolute values are generally lower at northern slopes (NW and NE) than at the southern slopes (SW and SE) (Figure S7 and Figure S8). In detail, seasonal HS ranged from −26%/°C during CD (NW) to −36%/°C during WW (SE) . Similarly, the maximum peak HS max sensitivity to temperature and precipitation was at SE during WW seasons (25%/ºC) and the minimum was during CD seasons at NW (15%/ºC). The snow duration sensitivity to temperature and precipitation increased during WW seasons, and the maximum changes were at SE sector (−16%/ºC); in contraposition, the lowest sensitivity to temperature and precipitation are found at NW sector, during CD and CW seasons (−8%/ºC, in both seasons). Snow ablation sensitivity to

temperature and precipitation increases towards the eastern Pyrenees, particularly during WD seasons (14%/ºC and 13%/ºC for NE and SE, respectively). Finally, no remarkable peak HS date differences are observed between sectors and maximum values are found during CD and CW seasons, when the peak HS date is anticipated >= 5 per ºC for all sectors."

We have modified "5.3 Spatial and elevation factors controlling sensitivity to temperature and precipitation change" section:

"5.3 Spatial and elevation factors controlling sensitivity to temperature and precipitation change

Comparison between Pyrenean sectors (Figure 8) reveals no remarkable differences in the relative importance of each compound season type in the sensitivity to temperature and precipitation change. This is because by applying a joint-quantile approach for each massif and elevation, we are comparing similar climate seasons between sectors, regardless of the number of compound season types recorded in each massif during the baseline period (Figure S1 and S3). The highest absolute sensitivity to temperature and precipitation change values is found in the SE Pyrenees. This is consistent with the snow accumulation and ablation patterns previously reported in this region (Lopez-Moreno, 2005; Navarro-Serrano et al., 2018; Alonso-González et al., 2020a; Bonsoms et al., 2021a; Bonsoms et al., 2021b; Bonsoms et al., 2022). The Atlantic climate has less of an influence in the SE sector, and in situ observations indicated there was about half of the seasonal snow accumulation amounts as in northern and western areas at the same elevation (>2000 m; Bonsoms et al., 2021a). The snow in the SE Pyrenees is more sensitive to temperature and precipitation because these massifs are exposed to higher turbulence and radiative heat fluxes (Bonsoms et al., 2022). Similar conclusions are found for low elevations, where the results show an upward displacement of the snow line due to warming. Previous studies described the sensitivity of the snow pattern to elevation at specific stations of the central Pyrenees (López-Moreno et al., 2013; 2017), Iberian Peninsula mountains (Alonso-González et al., 2020a), and other ranges such as the Cascades (Jefferson, 2011; Sproles et al., 2013), the Alps (Marty et al., 2017), and western USA (Pierce et al., 2013; Musselman et al., 2017b). In these regions, the models suggest larger snowpack reductions due to warming at subalpine sites than at alpine sites (Jennings and Molotch, 2020) due to closer isothermal conditions (Brown and Mote, 2009; Lopez-Moreno et al., 2017; Mote et al., 2018)."

**Minor comments**

P1 L35: Should be "increasing" the albedo.

Changed.

P2 L36: Should be "decreasing surface and air temperature". And it is not absolutely true that snow decreases surface temperature. During winter, the snow/soil interface will mostly remain at 0°C if the soil is snow covered, while if the soil is snow free but the air temperature is cold, the

soil surface temperature will further decrease (snow is a really good insulator for the soil). In spring, it is true that snow cover will keep the soil colder. However, I would keep only "decreasing air temperature".

Changed for "modulating surface and air temperature"

P2 L59: "on snowpack duration" to "on snow cover duration"

Done

P3 L85-86: "and the different mountain exposure to the main air masses". Which "main air masses"?

Changed "main air masses" to "Atlantic air masses".

P4 L95: What is "mid-late"?

Changed: "21st century climate projections"

P4 L105: "Sensitivity" to what? (Same at lines 107, 108, 110, Sections 3.4 and 4.2 names, etc). Should always be "sensitivity to climate change", this is indeed a bit heavier in the text, but correct (see comment above).

Done.

P4 L115: What does "these" refer two? Long sentence with many commas, hard to follow, consider splitting in two sentences.

We refer to warm season.

We have changed

"on warm seasons in the Mediterranean basin (Vogel et al., 2019; De Luca et al., 2020) because these are likely to increase in the future (e.g., Meng et al., 2022)".

to:

"…Warm seasons in the Mediterranean basin require special attention because these are likely to increase in the future (e.g., Vogel et al., 2019; De Luca et al., 2020; Meng et al., 2022"

P9 L259: Do you mean "ablation rate"?

Snow ablation is described in the methodological section:

"…(v) daily average snow ablation per season (snow ablation, hereafter)."

Snow ablation sensitivity to temperature and precipitation change is expressed in % per ⁰C.

P9 L262: Why isn't snow duration also an accumulation indicator?

It can be also. We have deleted this phrase to avoid misunderstandings     .

P11 L287-288: "High elevation areas had lower season-to-season snow variability than low elevations for all season types (Figure 4)". I don't see any information about season-to-season variability in Figure 4.

We have changed:

"High elevation areas had lower season-to-season snow variability than low elevations for all season types (Figure 4)".

To:

"High elevation areas had lower seasonal HS variability between season types than low elevations (Figure 4)".

In Figure 4 it is clearly shown that seasonal HS differences between season types are greater in low elevation areas than at high elevation, which is in accordance with previous works as it is mentioned in the introduction section.

P11 L 289: Avoid using the word "significantly" if not in the context of a proper statistical analysis (and thus a statistical "significance").

Done.

We have deleted "significantly" and "significant" where needed. We have changed "significantly" to "clear", "remarkable" and "important" depending on the context.

P11 L289-291: "All the snowpack-perturbed scenarios indicated that snowpack decreased at low and mid elevations under warming climate scenario". This is also the case at high elevation (Figure 4).

We have changed:

"All the snowpack-perturbed scenarios indicated that snowpack decreased at low and mid elevations under warming climate scenario".
To

"All the snowpack-perturbed scenarios indicated that snowpack decreased for all elevations under warming climate scenario".

P13 L331: "the peak HS date per °C was earlier by 9 days". This does not mean anything to me, should be: "the peak HS was anticipated by 9 days per °C" (but see my comment on linear indicators).

Changed.

P13 L337: "and because": Remove "and"

Done.

P13-14 Figures 5-6: Why not having only one figure with three rows of boxplot?

Changed.

P14 L350: "At low elevations, the snow ablation in all four extreme seasons was 12%/ºC". Something is missing here. Should be "snow ablation rate increase". Same for the rest of the paragraph, should be ablation rate change/increase/decrease.

We have changed all the text according with snow ablation definition provided in the methodological section "…(v) daily average snow ablation per season (snow ablation, hereafter)."

In this case, we have changed:

"At low elevations, the snow ablation in all four extreme seasons was 12%/ºC"
To
"At low elevations, the snow ablation sensitivity to temperature and precipitation in all four extreme seasons was 12%/ºC"

P16 Figure 8: I think the y-axis units should be cm/day. Explain the numbers in the caption, e.g. "the numbers in the plot show the difference in ablation rate compared to the previous degree". Or maybe I don't understand the figure, and this just are absolute values of ablation rate.

Figure 8 (now Figure 7) shows the average daily snow ablation (cm/day) for the baseline climate and by each increment of temperature.

So then why stacking them on top of each other and why having a y-axis? Shouldn't it be like figure 7? Also, I can't reconciliate what I shown in this figure with the numbers in Table 4 for the ablation columns. In the text and Table 4, you have ablation in %/°C, and here in cm/day.

We have changed Figure 7 type.

We show absolute (cm/day) and relative values (%/ºC) because:

1. Relative values are easier to compare with past and future works in other geographical areas.

2. Absolute values are specially interesting to compare between elevation band (i.e., analyze slow snowmelt rates in marginal snowpacks).

3. We must include absolute and relative values for consistency, because we have done the same for the other snow climate indicators.

We have added in the description:

"Figure 7. Absolute snow ablation values (cm/day) at three different elevations during four different compound temperature and precipitation seasons for different temperature increases above baseline (gray)."

If the editor considers that we should modify it, we will implement such changes.

P18L398: "The sensitivity of snow to different spatial patterns of climate change that we identified here […]". You do not study different patterns of climate change; you apply the same delta everywhere. Please correct.

Changed: "The different sensitivity to temperature and precipitation change spatial patterns that we identified here (Figures 9 and 10)"

P21L504-505: "Our maximum snow ablation and peak HS date occurred during dry seasons …]". Are you talking about change or about absolute value in the baseline simulation?

Changed:" Our maximum snow ablation relative change over the baseline scenario…"

P21L508-510: "The temperature in the Pyrenees is still cold enough to allow snowfall at high elevations during WW seasons, and for this reason we found maximal sensitivities during WD seasons." I don't understand, temperature is almost the same in WW and in WD, so why is sensitivity to climate change greater in WD?

This is because larger precipitation in wet seasons will allow more snowfall and a slight increase of temperature will not affect snowpack evolution at high elevation and in the coldest months of the season (Figure S9).

P23L562-564: "however, a more complex model does not necessarily provide better performance in terms of snowpack and runoff estimation (Magnusson et al., 2015)". But it also can, especially for climate change study (see Carletti et al, 2022, doi.org/10.5194/hess-26- 3447-2022)

Thank you for your suggestion and many congratulations for your work.

We have included your reference in our manuscript. We have added:

"In this work we used a physical-based snow model since it provides better results for future snow climate change estimations than degree-day models (Carletti et al., 2022)"

We have modified this paragraph:
"…The FSM2 is a physics-based model of intermediate complexity, and the estimates of snow densification are simpler than those from more complex models of snowpack. The FSM2 configuration implemented in this work includes snow meltwater retention, snowpack refreezing and snow albedo based on snow age, which are the physical parameters included in the best-performing snow models according to Essery et al. (2013). Snow model sensitivity studies reveal that intermediate complexity models exhibit similar SWE accuracies than most complex snow models, as well as robust performances across seasons (Terzago et al., 2020)."

Regarding the robustness of the snow model, the reader is informed in the methodological section:

"…and its application in different forest environments (Mazzoti et al., 2021), and hydro-climatological mountain zones such the Andes (Urrutia et al., 2019), Alps (Mazzoti et al., 2020), Colorado (Smyth et al., 2022), Himalayas (Pritchard et al., 2020), Iberian Peninsula Mountains (Alonso-González et al., 2020a; Alonso-González et al., 2022), Lebanese mountains (Alonso-González et al., 2021), providing confidential results."

If the editor considers that we should change the snow model or add more FSM2 uncertainties, we will proceed accordingly.

P564-566: "Biases in the SAFRAN system and biases related to the FSM2 were minimal because we quantified relative changes between a modeled snow scenario (climate baseline) and several perturbed scenarios". This assumes constant biases. Snow cover involves different variables, non-linear processes, and will accumulate errors along the season, we cannot be that certain.

We have deleted this phrase.

P23L568-569: "[…] but assumes that the snow patterns of the reference climate period will be constant over time." Don't you mean meteorological patterns? (E.g. you don't capture change in precipitation regime, you just scale the intensity of each events). Please clarify.

Thank you for your suggestion, we have changed "climate" for "meteorological".

---

## Author Response (AR5)

Josep M Bonsoms
Department of Geography (University of Barcelona)
Montalegre 6-8, 3er floor (3007)
08001 - Barcelona (Spain)
Tel.: +34 655 36 49 42 / +34 934 037 849
josepbonsoms5@ub.edu

[Figure]
* * *
Barcelona, 22th February 2023

Dear Dr. Helbig,

I am submitting the third version of the manuscript entitled "**Snow sensitivity to temperature and precipitation change during compound cold-hot and wet-dry compound seasons in the Pyrenees**", co-authored by myself, Dr. López-Moreno and Dr. Alonso-González.

Thank you very much for the time you expended reading our manuscript. Thanks a lot for your careful review and for formulating positive and constructive comments.

A point-by-point answer to editor and reviewer 2 suggestions can be found in the following pages.

Yours sincerely,

Josep Bonsoms, on behalf of the co-authors.

**Response to third review of "Snow sensitivity to temperature and precipitation change during compound cold-hot and wet-dry seasons in the Pyrenees"**

**by Josep Bonsoms[1], Juan Ignacio López-Moreno[2] and Esteban Alonso[3]**

[1] Department of Geography, University of Barcelona, Barcelona, Spain.

[2] Instituto Pirenaico de Ecología (IPE-CSIC), Campus de Aula Dei, Zaragoza, Spain.

[3] Centre d'Etudes Spatiales de la Biosphère (CESBIO), Université de Toulouse, CNES/CNRS/IRD/UPS, Toulouse, France.

Corresponding author: J.I López-Moreno (nlopez@ipe.csic.es)

Response to Editor.

Reviewer comments are in black and responses in blue.

I would like to thank you again for your revisions and acknowledge your work on the revised manuscript!

The reviewer, I sent the manuscript back to, also thought the manuscript has improved a lot and has only two minor points with regards to more references or information.

I have the following minor, rather technical comments:

Thank you very much for your recommendations and suggestions.

Line 17-19: Where in the manuscript do you mention these exact numbers? Maybe I missed it, but I couldn't find them in the manuscript.

We have delated the exact numbers since it is not included in the manuscript (it is only shown at Figure 5 a).

We have changed:

When the temperature increased progressively at 1⁰C intervals, the largest seasonal HS decreases from the baseline were at +1⁰C (47% at low elevation, 48% at mid-elevation, and 25% at high elevation)

To:

"When the temperature increased progressively at 1ºC intervals, the largest seasonal HS decreases from the baseline were at +1ºC."

With regards to Section 4.1 including Fig. 2 and 3:

It might help the understanding when you indicate the elevations in the figures, e.g. behind the naming "Ax" or briefly mention station details in the captions or when discussing the performances of A1 to A4?

We have followed your first suggestion and we have changed Fig 2 and 3 and included the elevation behind the naming.

There are some minor editing/language issues or typos:

- abstract Line 25: "Results suggests" -Remove "s"

Done
- Line 252 missing line break;

Done
- End of L. 292: "."

Done
- Line 325 "The maximum seasonal HS was during [..]" - Is there something missing (sensitivity?)

Yes, we have added: "… maximum seasonal HS **sensitivity to temperature and precipitation**".

Once these rather minor revisions have been addressed, I will do my best that the manuscript can be published very soon.

Best regards,

Nora Helbig

**Response to third review of "Snow sensitivity to temperature and precipitation change during compound cold-hot and wet-dry seasons in the Pyrenees"**

**by Josep Bonsoms[1], Juan Ignacio López-Moreno[2] and Esteban Alonso[3]**

[1] Department of Geography, University of Barcelona, Barcelona, Spain.

[2] Instituto Pirenaico de Ecología (IPE-CSIC), Campus de Aula Dei, Zaragoza, Spain.

[3] Centre d'Etudes Spatiales de la Biosphère (CESBIO), Université de Toulouse, CNES/CNRS/IRD/UPS, Toulouse, France.

Corresponding author: J.I López-Moreno (nlopez@ipe.csic.es)

Response to Reviewer 2.

Reviewer comments are in black and responses in blue.

Dear Editor, Dear Authors,

I would like to acknowledge the work done on the manuscript. All my concerns have been satisfactorily addressed (either by incorporating my recommendations or by clear responses).

The additional text and figures improve the manuscript. I have only a few remaining minor points, which are listed below.

Best regards,

Adrien Michel

Thank you very much for your recommendations and suggestions.

I fully agree with the extended argumentation on the contradiction between nonlinearity and the use of a linear indicator (%/°C) to facilitate interpretation and comparison between massifs and with other studies. I appreciate the more detailed analysis between massifs, which 1) confirms the main results and 2) strengthens the analysis. I would only recommend adding some details on the PCA. It is not clear to me on which sets of variables the PCA is performed. The two

references given did not allow me to fully understand. Just one or two sentences explaining exactly on which data set the PCA is performed would be very helpful.

We have changed the description for:

Following previous studies, massifs were grouped into four sectors by applying a Principal Component Analysis (PCA) (i.e., López-Moreno et al., 2020b; Matiu et al., 2020, among others). We applied a PCA over HS data for each month, year, massif, and elevation. Massifs were grouped into fours sectors depending on the maximum correlation to PC1 and PC2 scores (see Figures S2).

Finally, in the conclusion (l.628) you say: "[…] in this region, a 10% increase of precipitation, as suggested by many climate projections over the eastern regions of this range, could compensate for temperature increases on the order of about < 1ºC. " However, the only references I can find in the text are on line 96: "an increase (decrease) of precipitation by about 10% for the eastern (western) regions during winter and spring (Amblar-Francés et al., 2020)" and on line 479: "Snow sensitivity in the easternmost areas could decline during the winter because of a trend for an increase of about 10% in precipitation in this area (Amblar-Francés et al., 2020)". A single reference is not enough to state: "as suggested by many climate projections". You could add some references or rephrase this sentence.

Thank you for your suggestion. This work is the reference study about climate projections in the Pyrenees. It was published under the framework of the last climate change research project in the mountain range (CLIMPY). There are not more works about precipitation projections at high resolution and using CMIP6.

We have moderated our statement and we changed "as suggested by many climate projections" to "as suggested by the Spanish Meteorological Agency (AEMET) climate projections."